# ON THE COMPLETENESS OF INVARIANT GEOMETRIC DEEP LEARNING MODELS

**Zian Li**[1,2], **Xiyuan Wang**[1,2], **Shijia Kang**[1], **Muhan Zhang**[1,*]
[1]Institute for Artificial Intelligence, Peking University
[2]School of Intelligence Science and Technology, Peking University

## ABSTRACT

Invariant models, one important class of geometric deep learning models, are capable of generating meaningful geometric representations by leveraging informative geometric features in point clouds. These models are characterized by their simplicity, good experimental results and computational efficiency. However, their theoretical expressive power still remains unclear, restricting a deeper understanding of the potential of such models. In this work, we concentrate on characterizing the theoretical expressiveness of a wide range of invariant models under *fully-connected* conditions. We first rigorously characterize the expressiveness of the most classic invariant model, message-passing neural networks incorporating distance (DisGNN), restricting its unidentifiable cases to be only highly symmetric point clouds. We then prove that GeoNGNN, the geometric counterpart of one of the simplest subgraph graph neural networks, can effectively break these corner cases' symmetry and thus achieve E(3)-completeness. By leveraging GeoNGNN as a theoretical tool, we further prove that: 1) most subgraph GNNs developed in traditional graph learning can be seamlessly extended to geometric scenarios with E(3)-completeness; 2) DimeNet, GemNet and SphereNet, three well-established invariant models, are also all capable of achieving E(3)-completeness. Our theoretical results fill the gap in the expressive power of invariant models, contributing to a rigorous and comprehensive understanding of their capabilities.

## 1 INTRODUCTION

Learning geometric structural information from 3D point clouds constitutes a fundamental requirement for various real-world applications, including molecular property prediction, physical simulation, and point cloud classification/segmentation (Schmitz et al., 2019; Sanchez-Gonzalez et al., 2020; Jumper et al., 2021; Guo et al., 2020).

In the context of designing geometric models for point clouds, a pivotal consideration involves ensuring that the model respects both permutation symmetry and Euclidean symmetry (i.e., symmetry of SE(3) or E(3) group). This requirement necessitates the incorporation of appropriate inductive biases into the architecture, ensuring that the model's output remains invariant or equivariant to point reordering and Euclidean transformation.

However, these restrictions on symmetry can impose limitations on the ability of models to approximate a broad range of functions. For example, Message Passing Neural Networks (MPNNs) (Gilmer et al., 2017), a representative class of Graph Neural Networks (GNNs), learn permutation symmetric functions over graphs. However, it has been shown that such functions are not universal (Xu et al., 2018a; Morris et al., 2019), leading to the development of more expressive GNN frameworks (Maron et al., 2018; Morris et al., 2019; Zhang et al., 2023). Similar challenges arise in the geometric setting, where models must additionally respect the Euclidean symmetry, and thus achieving geometric universality (E(3)-completeness) becomes a non-trivial task as well.

In this work, we systematically investigate the expressiveness of a wide range of invariant models under fully-connected conditions , where interactions occur among all points. We begin by revisiting DisGNN (Li et al., 2024), the simplest invariant model augmenting MPNNs with Euclidean distances

---

*Corresponding author: Muhan Zhang (muhan@pku.edu.cn).

between nodes as additional edge features. We seek to answer the question: *how close is DisGNN to completeness?* Through theoretical analysis, we show that **DisGNN is nearly-E(3)-complete**, whose unidentifiable cases can be restricted to a 0-measure subset containing well-defined highly-symmetric point clouds only. Our results extend the findings of previous works that solely concentrated on individual hand-crafted counterexamples illustrating the incompleteness of DisGNN (Li et al., 2024; Pozdnyakov & Ceriotti, 2022; Pozdnyakov et al., 2020), and a contemporary work that provided a coarser characterization of DisGNN's expressiveness (Hordan et al., 2024b).

We then show that **GeoNGNN**, the geometric counterpart of NGNN (Zhang & Li, 2021) (one simple subgraph GNN), can effectively break all DisGNN's unidentifiable cases' symmetry through node marking, thereby achieving **E(3)-completeness**. We then leverage GeoNGNN as a theoretical tool, proceeding to establish the E(3)-completeness of a wide range of invariant models by showcasing their ability to implement GeoNGNN. Specifically, we systematically define the geometric counterparts of subgraph GNNs within Zhang et al. (2023)'s framework by generalizing their local operations to incorporate geometric information, and then prove that **all of these geometric subgraph GNNs are E(3)-complete**. Furthermore, we establish that three well-established invariant geometric models, **DimeNet** (Gasteiger et al., 2019), **SphereNet** (Liu et al., 2021) and **GemNet** (Gasteiger et al., 2021), are also all **capable of achieving E(3)-completeness**.

These established E(3)-complete models, except for GemNet (Gasteiger et al., 2021), are either known to be strictly weaker than 2-FWL (Cai et al., 1992) in traditional graph learning setting, or utilize less informative aggregation schemes than 2-FWL-like geometric models (Li et al., 2024; Delle Rose et al., 2023; Hordan et al., 2024b;a) (See Section 5 for details). However, our investigation reveals that in the geometric setting, these models are equally E(3)-complete. These surprising findings can deepen our understanding of the power of invariant models, while also suggesting that the bottleneck of invariant models may lie in generalization instead of expressiveness. As a side effect, by generalizing subgraph GNNs to the geometric setting, we greatly enlarge the design space of geometric deep learning models, which we hope could inspire future, more powerful model designs.

## 2 RELATED WORKS

Designing complete invariant descriptors for point clouds plays a crucial role for scientific problems (Nigam et al., 2024), since complete invariants directly allow universal function approximations when coupled with MLPs (Hordan et al., 2024b; Li et al., 2024). Previous works, such as Widdowson & Kurlin (2022; 2023); Kurlin (2023), have proposed polynomial-time algorithms to compute and compare complete invariants that uniquely determine point clouds. However, these invariants are often highly structured, such as sets of matrices (Kurlin, 2023), and the absence of corresponding neural-network formulations limits their practical applicability.

In the neural-network context, most prior works propose powerful models but typically with "weaker completeness"(Wang et al., 2022; Puny et al., 2021; Duval et al., 2023; Du et al., 2024; Villar et al., 2021). For example, ComENet (Wang et al., 2022) uses nearest neighbors as references for computing higher-order geometric information, which is infeasible for symmetric point clouds where nodes can have multiple nearest neighbors, thereby limiting its completeness over a asymmetric subset of point clouds [1]. Frame-based methods, such as FAENet (Duval et al., 2023), LEFTNet (Du et al., 2024), and others (Wang & Zhang, 2022; Puny et al., 2021), project coordinates or vector features onto local or global equivariant frames and employ powerful structures like MLPs (Hornik et al., 1989) to ensure expressiveness. However, these frames can degenerate in symmetric structures (e.g., structures with non-distinct eigenvectors (Puny et al., 2021; Duval et al., 2023)), a limitation that these methods do not carefully address and theoretically characterize. Although symmetric point clouds often represent zero-measure cases (Hordan et al., 2024b), they can significantly impact downstream continuous function learning, as demonstrated in (Pozdnyakov & Ceriotti, 2022; Hordan et al., 2024b).

Several previous works (Dym & Maron, 2020; Joshi et al., 2023; Hordan et al., 2024b; Li et al., 2024; Delle Rose et al., 2023; Hordan et al., 2024a) have characterized geometric models' expressiveness in a more rigorous sense. The seminal work (Dym & Maron, 2020) proved the universality of structures like TFN (Thomas et al., 2018), while such universality requires arbitrarily high-order tensors that are computationally expensive in practice. Another related work, GWL (Joshi et al., 2023), offers many

---

[1]Arbitrarily selecting one when multiple nearest neighbors exist could break permutation invariance, a fundamental principle ensuring stable outputs. We focus only on invariant models.

insightful theoretical conclusions, including that GWL provides upper bounds on the expressiveness of many equivariant models (Batatia et al., 2022; Satorras et al., 2021). However, GWL relies on injective multiset functions involving equivariant features, which do not exhibit practical polynomial-time neural forms like scalar multiset functions proposed in (Dym & Gortler, 2024; Amir et al., 2024), thus serving more as conceptual upper bounds. Joshi et al. (2023) also does not directly address completeness of models as we do. A recent work (Cen et al., 2024) characterizes the symmetry of point clouds from point group perspective, and investigate the expressiveness through degeneration of equivariant functions. Notably, recent works (Li et al., 2024; Delle Rose et al., 2023; Hordan et al., 2024b;a) have proposed provably complete geometric models that can produce complete invariant features. Nevertheless, these methods are predominantly built upon the 2-FWL framework (Cai et al., 1992), and the completeness of a broader class of invariant models—many of which employ weaker aggregation schemes (Gasteiger et al., 2019)—remains an open question. In this work, we focus on a significantly broader range of invariant methods (Schütt et al., 2018; Gasteiger et al., 2019; 2021; Liu et al., 2021; Zhang & Li, 2021; Zhang et al., 2023), some of which are newly proposed based on subgraph GNNs in traditional graph learning (Zhang & Li, 2021; Zhang et al., 2023), with the aim to fully characterize their completeness rigorously.

# 3 PRELIMINARY

## 3.1 NOTATIONS AND DEFINITIONS

We investigate the expressiveness of invariant models for unlabeled point clouds consisting of $n$ nodes[2], denoted by $P \in \mathbb{R}^{n \times 3}$. The coordinates of the $i$-th node in $P$ are represented by $p_i$. We use to denote sets and $\{\!\{\}\!\}$ to denote multisets, with the set $1, 2, \ldots, n$ represented as $[n]$. The Euclidean distance between nodes $i$ and $j$ is denoted as $d_{ij}$.

Two point clouds $P_1 \in \mathbb{R}^{n_1 \times 3}$ and $P_2 \in \mathbb{R}^{n_2 \times 3}$ are *isomorphic* if $n_1 = n_2 = n$, and there exists a rotation matrix $\mathbf{R} \in O(3)$, a translation vector $\mathbf{t} \in \mathbb{R}^3$, and a permutation matrix $\mathbf{P} \in \mathbb{R}^{n \times n}$ such that $(\mathbf{P}P_1)\mathbf{R} + \mathbf{t} = P_2$. Invariant models considered in this study are all *permutation- and E(3)-invariant*, meaning they embed isomorphic point clouds into the same $k$-dimensional representation $s \in \mathbb{R}^k$.

**Definition 3.1** (Distinguish, Identify, and E(3)-Completeness). Let $f : \bigcup_{n=1}^{\infty} \mathbb{R}^{n \times 3} \to \mathbb{R}^k$ be a permutation- and E(3)-invariant function that maps point clouds to a $k$-dimensional representation. We define the following concepts:

- **Distinguish:** Given two non-isomorphic point clouds $P_1$ and $P_2$, if $f(P_1) \neq f(P_2)$, we say $f$ can distinguish $P_1$ and $P_2$.
- **Identify:** For a point cloud $P_1$, if for any non-isomorphic $P_2$, we always have $f(P_1) \neq f(P_2)$, we say $f$ can identify $P_1$.
- **E(3)-Completeness:** If for any pair of non-isomorphic point clouds $P_1$ and $P_2$, we always have $f(P_1) \neq f(P_2)$, we say $f$ is E(3)-complete.

Notably, we introduce the novel concept of "Identify", which allows for a *finer-grained* analysis of model expressiveness than commonly adopted "Distinguish": By definition, each "identify" operation encompasses an infinite number of "distinguish" pairs. It is evident that identifying all point clouds implies E(3)-completeness. Furthermore, since distinguishing point clouds with different node counts is straightforward, we focus solely on the case where point clouds have the same finite size.

For a parametric model $f_\theta$, we investigate its maximal expressive form where all intermediate functions are injective, following typical ways (Delle Rose et al., 2023; Li et al., 2024). Notably, recent work by Dym & Gortler (2024); Amir et al. (2024) proves the existence of neural function forms and parametrizations that ensure injectivity for multiset functions commonly used in geometric models (Gasteiger et al., 2019; 2021; Li et al., 2024), allowing us to achieve such completeness in polynomial time and under (for Lebesgue) almost every parametrization.

---

[2] Unlabeled point clouds represent one of the most complex and general cases in geometric expressiveness (Kurlin, 2023; Widdowson & Kurlin, 2023), surpassing the complexity of point clouds with initial features (e.g., molecules). We consider cases where $P$ contains more than one point, as trivial cases arise otherwise.

## 3.2 DisGNN

To leverage the rich geometric information present in point clouds, a straightforward approach is to model the point cloud as a *distance graph*, where edges are present between nodes if their Euclidean distance is below a certain cutoff $r_{\text{cutoff}}$, and edge weights are Euclidean distance $d_{ij}$. Then, an MPNN is applied to it. We call models with such framework as DisGNN, and SchNet (Schütt et al., 2018) is a representative of such works.

To be specific, DisGNN models first embed nodes based on their initial node features $X \in \mathbb{R}^{n \times d}$ (if there exists), such as atomic numbers, obtaining initial node embedding $h_i^{(0)}$ for node $i$. They then perform message passing according to the following equation iteratively:

$$h_i^{(l+1)} = f_{\text{update}}^{(l)} \left( h_i^{(l)}, f_{\text{aggr}}^{(l)} \left( \{\!\{ (h_j^{(l)}, d_{ij}) \mid j \in \mathcal{N}(i) \}\!\} \right) \right), \tag{1}$$

where $h_i^{(l)}$ represents the node embedding for node $i$ at layer $l$, $\mathcal{N}(i)$ denotes the neighbor set of node $i$. The global graph embedding is then obtained by aggregating the representations of all nodes using a permutation-invariant function $f_{\text{out}}(\{\!\{ h_i^{(L)} \mid i \in [n] \}\!\})$, where $L$ is the total number of layers and $n$ is the total number of nodes.

In the theoretical analysis for DisGNN (Section 4), we assume *fully-connected* distance graph modeling for point clouds, i.e., $\mathcal{N}(i)$ is $[n]$ by default, and sufficient iterations until convergence. This ensures maximal geometric expressiveness of DisGNN and aligns with existing works (Pozdnyakov & Ceriotti, 2022; Hordan et al., 2024b; Li et al., 2024).

## 4 HOW POWERFUL IS DISGNN?

In Pozdnyakov & Ceriotti (2022); Li et al. (2024), researchers have carefully constructed pairs of symmetric point clouds that DisGNN cannot distinguish to illustrate its E(3)-incompleteness. However, rather than relying on hand-crafted finite pairs of counterexamples and intuitive illustration, a more significant question arises: *what properties do the point clouds that DisGNN cannot **identify** share in common*? Only after theoretically characterizing these corner cases can we describe and bound the expressiveness of DisGNN in a rigorous way, and then design more powerful geometric models accordingly by "solving" these corner cases.

Naively, to determine whether a point cloud $P$ can be identified by DisGNN requires iterating through all other point clouds $P'$ within the entire cloud set $\mathbb{R}^{n \times 3}$ according to its definition. Here, we propose a simple sufficient condition for a point cloud to be identifiable by DisGNN, which focuses *solely on $P$*. To achieve this, we first rigorously define the concept of $\mathcal{A}$-symmetry.

**Definition 4.1** ($\mathcal{A}$-symmetry). Given a point cloud $P \in \mathbb{R}^{n \times 3}$, let $\mathcal{A}$ be a permutation-equivariant and E(3)-invariant algorithm taking $P$ as input and produces node-level features $X^{\mathcal{A}} = \mathcal{A}(P) \in \mathbb{R}^{n \times d}$ (we use $x_i^{\mathcal{A}} \in \mathbb{R}^d$ to denote the $i$-th nodes' feature). We define $\mathcal{A}$ center set of $P$ as $\mathcal{A}^{\text{set}}(P) = \{ \frac{\sum_{i \in [n]} m(x_i^{\mathcal{A}}) p_i}{\sum_{i \in [n]} m(x_i^{\mathcal{A}})} \mid m : \mathbb{R}^d \to \mathbb{R}, \sum_{i \in [n]} m(x_i^{\mathcal{A}}) \neq 0 \}$. If $|\mathcal{A}^{\text{set}}(P)| = 1$, then we say $P$ is $\mathcal{A}$-symmetric; otherwise, we say $P$ is $\mathcal{A}$-asymmetric.

Intuitively, $\mathcal{A}$ assigns features to nodes of the point cloud based on their spatial properties, while $m$ represents a "mass" function that maps these features to corresponding "masses".[3] Thus, $\mathcal{A}^{\text{set}}(P)$ denotes the collection of "barycenters" associated with different "mass" assignments. For example, we can consider $\mathcal{A}$ as the distance encoding function, denoted as $\mathcal{C}$, which computes the distance from each node to the geometric center as the node feature. A $\mathcal{C}$-asymmetric case is shown in Figure 1(a).

Obviously, the cardinality of $\mathcal{A}^{\text{set}}(P)$ is solely dependent on the *partition* of node features $\mathcal{A}(P)$ and is independent of the specific feature values. We also propose a more powerful algorithm, DisGNN encoding, denoted as $\mathcal{D}$, which applies DisGNN to point clouds iteratively until the partition of node features stabilizes. Notably, earlier studies by Delle Rose et al. (2023); Li et al. (2024) have shown that $\mathcal{D}$ guarantees a node feature partition that is no coarser than that of $\mathcal{C}$. Consequently, $\mathcal{D}$-symmetry always implies $\mathcal{C}$-symmetry, representing a stricter form of symmetry.

Based on these definitions, we now present the primary theorem demonstrating that all such asymmetric point clouds can be identified by DisGNN.

---

[3]Note that the geometric center of $P$ is always included in $\mathcal{A}^{\text{set}}(P)$ by defining $m$ as a constant function.

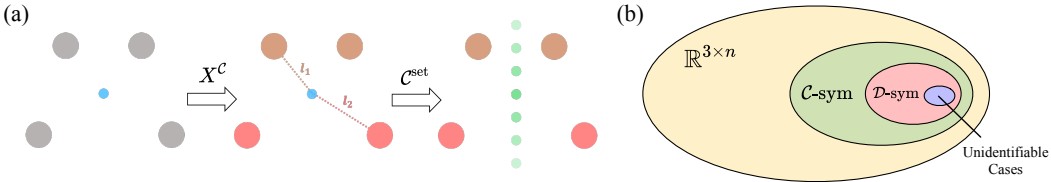

Figure 1: (a) A $\mathcal{C}$-asymmetric point cloud. The blue small node represents the geometric center, and each green node represents an element in $\mathcal{C}^{\mathrm{set}}(P)$. The large nodes constitute the point cloud, with different node colors denoting different node features. (b) Illustration of the relations among different subsets of the point cloud. $\mathcal{A}$-sym parts encompass all $\mathcal{A}$-symmetric graphs. Each subset has a measure of 0 and strictly contains their sub-parts (proper subset relation).

**Theorem 4.2** (Asymmetric point clouds are identifiable). *Let $\mathcal{C}$ denote the center distance encoding and $\mathcal{D}$ the DisGNN encoding. Then, given an arbitrary point cloud $P \in \mathbb{R}^{n \times 3}$, $P$ is $\mathcal{C}$-asymmetric $\Rightarrow P$ is $\mathcal{D}$-asymmetric $\Rightarrow P$ can be identified by DisGNN.*

**Key idea.** Given a point cloud $P$, when it has two global-level *distinct anchors*, such as geometric center or other centers that can be determined in a permutation-invariant and E(3)-equivariant way, DisGNN has the potential to learn *triangular distance encoding* for each node $i$ w.r.t. $i$ and these two global anchors, and produces complete representation for $P$ by leveraging such encoding information (Lemma B.6). For $\mathcal{C}$- or $\mathcal{D}$-asymmetric point clouds, their center sets $\mathcal{C}^{\mathrm{set}}(P)$ or $\mathcal{D}^{\mathrm{set}}(P)$ have more than 1 element, within which any pair of distinct centers can be utilized by DisGNN for such encoding (Lemma B.4), thereby facilitating DisGNN to fully represent and identify $P$.

We provide rigorous proof in Appendix B.3 and *encourage readers to refer to it*, where we show the superior global geometric learning ability of DisGNN and can inspire the development of new complete model designs in the triangular distance encoding perspective, as exemplified by GeoNGNN proposed in the following section.

Theorem 4.2 shows that to test if a point cloud is identifiable by DisGNN, one can first check its $\mathcal{C}$-symmetry, which is quick and intuitive. If it is $\mathcal{C}$-asymmetric, it is identifiable; otherwise, one can test for the more powerful but costlier $\mathcal{D}$-symmetry. Notably, although the latter test still requires running DisGNN, it demonstrates that a single execution of DisGNN on $P$ can provide an indication of identifiability, offering insights without necessitating an exhaustive search over all $P' \in \mathbb{R}^{n \times 3}$.

Importantly, Theorem 4.2 restricts the unidentifiable set of DisGNN to a constrained, highly symmetric subset, as illustrated in Figure 1(b). In Theorem 4.3, we provide a rigorous analysis demonstrating that the unidentifiable set of DisGNN has zero measure. These results sufficiently demonstrate that, despite its extremely simple and naive design, DisGNN is **nearly E(3)-complete**.

**Theorem 4.3** (Unidentifiable set of DisGNN has measure zero). *The Lebesgue measure on $\mathbb{R}^{n \times 3}$ of the $\mathcal{C}$-symmetric, $\mathcal{D}$-symmetric, and unidentifiable point cloud sets is zero.*

Finally, we note that a recent study (Hordan et al., 2024b) also indicates the near-completeness of DisGNN, by showing that it can distinguish arbitrary pairs of point clouds from an asymmetric point cloud $\mathbb{R}^{n \times 3}_{\mathrm{distinct}}$. Nevertheless, our findings represent a significant advancement by introducing a strictly larger asymmetric subset and employing the concept of "identify" instead of "pair-wise distinguish". This provides a finer characterization of the near completeness of DisGNN. Please refer to Appendix C.2 for a detailed comparison.

## 5 ON THE COMPLETENESS OF INVARIANT GEOMETRIC MODELS

In this section, we move on to demonstrate and prove the E(3)-completeness of a broad range of invariant models that leverage more geometric features or more complicated aggregation schemes than DisGNN under fully-connected conditions. These models can effectively identify all those highly-symmetric point clouds that DisGNN can not. We first introduce a new simple invariant model design, which can precisely break such symmetry and achieve completeness. Consequently, we show how this new design serves as a theoretical tool for proving a wide range of invariant models' expressiveness. Proof for this section is provided in Appendix B.

## 5.1 GeoNGNN: Breaking Symmetry Through Node Marking

According to Theorem 4.2 (along with the lemmas emphasized in its key idea), the essential factor in identifying a point cloud by DisGNN is the existence of two distinct anchors. Although every point cloud inherently possesses one, i.e., the geometric center, the other may not exist in some highly symmetric cases, such as a sphere. This hinders DisGNN from achieving completeness. In this subsection, we show that GeoNGNN, the geometric counterpart of Nested GNN (NGNN) (Zhang & Li, 2021) , can fill in this last piece of the puzzle by applying DisGNN on point clouds with an additionally marked node, which breaks the symmetry and exactly acts as the other anchor.

GeoNGNN employs a two-level hierarchical framework by *nesting* DisGNN. Specifically, it processes a given point cloud as follows: **1)** The first level, referred to as the inner GNN, operates independently on each point's $r_{\text{sub}}$-sized sub-point cloud and aggregates the local sub-point cloud information into an initial embedding for the corresponding point in the original point cloud. **2)** The second level, referred to as the outer GNN, processes the original point cloud based on the embeddings generated by the inner GNN, thereby producing final point-level and cloud-level representations.

Formally, for the inner GNN, the representation of the node $j$ in node $i$'s sub-point cloud at the $l$-th layer, denoted as $h_{ij}^{(l)}$, is updated as:

$$h_{ij}^{(l+1)} = f_{\text{update, inner}}^{(l)}\left(h_{ij}^{(l)}, f_{\text{aggr, inner}}^{(l)}\left(\{\!\!\{(h_{ik}^{(l)}, d_{kj}) \mid k \in \mathcal{N}(j), d_{ik} \leq r_{\text{sub}}\}\!\!\}\right)\right), \qquad (2)$$

for all $h_{ij}$ satisfying $d_{ij} \leq r_{\text{sub}}$, and $\mathcal{N}(j)$ represents all nodes $k$ satisfying $d_{kj} \leq r_{\text{cutoff}}$. Here, $r_{\text{sub}}$ and $r_{\text{cutoff}}$ are hyperparameters representing the subgraph size and interaction cutoff, respectively. $h_{ij}^{(0)}$ is initialized as:

$$h_{ij}^{(0)} = f_{\text{init, inner}}(x_j, d_{ij}, \mathbb{1}_{i=j}), \qquad (3)$$

where $x_j$ is the raw feature of node $j$, and $d_{ij}$ and $\mathbb{1}_{i=j}$ denote the position encoding and marking of the center point, respectively. After $N_{\text{in}}$ iterations, the inner GNN summarizes the sub-point cloud as:

$$h_i^{(0)} = f_{\text{output, inner}}(\{\!\!\{h_{ij}^{(N_{\text{in}})} \mid j \in [n], d_{ij} \leq r_{\text{sub}}\}\!\!\}), \qquad (4)$$

producing the initial point representations for the outer GNN. The outer GNN then updates these representations over $N_{\text{out}}$ iterations following the framework in Equation (1).

By representing point clouds as distance graphs, GeoNGNN captures *subgraph* patterns instead of *subtree* patterns, which have been shown to be more expressive in traditional graph learning (Zhang & Li, 2021). We now show that GeoNGNN achieves geometric completeness under fully-connected conditions, thereby effectively addressing expressiveness limitations of DisGNN.

**Theorem 5.1** (E(3)-Completeness of GeoNGNN). *When the following conditions are met, GeoNGNN is E(3)-complete:*

- *$N_{in} >= 5$ and $N_{out} >= 0$ (where $0$ indicates that the outer GNN only performs final pooling).*
- *The distance graph is fully-connected ($r_{cutoff} = +\infty$).*
- *All subgraphs are the original graph ($r_{sub} = +\infty$).*

**Key idea.** Given an arbitrary point cloud $P$, consider a specific node $i$ within it that differs from the geometric center. As node $i$ is explicitly marked within its own subgraph (Equation (3)), there are now two anchors in this subgraph, namely node $i$ and the geometric center, that facilitate triangular distance encoding by DisGNN. Consequently, DisGNN can fully represent and identify $P$ through the representation of node $i$'s subgraph, as underscored in the key idea of Theorem 4.2. Notice that in a point cloud with more than two nodes, a node that is distinct from the geometric center always exists, and through the outer pooling, overall completeness and permutation invariance are guaranteed.

While Theorem 5.1 establishes that infinite subgraph radius and distance cutoff guarantee completeness over all point clouds, in Appendix D.2 we show that finite values also lead to boosted expressiveness compared to DisGNN. And we notice that the conditions specified in Theorem 5.1 ultimately result in *polynomial-time* complexity w.r.t. the size of the point cloud even when considering the complexity involved in obtaining intermediate injective functions, which aligns with prior studies (Kurlin, 2023; Widdowson & Kurlin, 2023; Hordan et al., 2024b; Li et al., 2024; Delle Rose et al., 2023). The detailed complexity analysis is provided in Appendix D.1.

## 5.2 COMPLETENESS OF GEOMETRIC SUBGRAPH GNNS

GeoNGNN provides a simple approach to extend traditional subgraph GNNs to geometric scenarios with remarkable geometric expressiveness. Indeed, the realm of traditional graph learning literature contains a multitude of subgraph GNNs such as DSS-GNN (Bevilacqua et al., 2021), GNN-AK (Zhao et al., 2021), OSAN (Qian et al., 2022) and so on (You et al., 2021; Frasca et al., 2022). Extending these models to geometric settings can significantly *enlarge the design space* of geometric models, and potentially introduce valuable *inductive biases*. In this section, we take a pioneering step to do this, subsequently establishing the geometric completeness of all these models.

We first define the *general geometric subgraph GNN*, a broad family of geometric models, by slightly adapting the unweighted graph models from Zhang et al. (2023) to handle geometric scenarios. For simplicity, we provide an intuitive overview and outline the modifications here, while self-contained and formal definitions can be found in Appendix E.

**Definition 5.2** (General geometric subgraph GNN, *informal*). A general geometric subgraph GNN takes point clouds $P \in \mathbb{R}^{n \times 3}$ (potentially with node features $X \in \mathbb{R}^{n \times d}$, such as atomic numbers) as input, treats it as a distance graph with interaction cutoff $r_{\text{cutoff}}$, and:

- Utilizes node marking with $r_{\text{sub}}$-size ego subgraph as the subgraph generation policy.
- Stacks multiple geometric subgraph GNN layers, each following *general subgraph GNN layer* defined in Zhang et al. (2023), which could arbitrarily include single-point, global, and local operations for aggregating graph information at different levels. The only modification is to the local operations: when updating $h_{uv}$ (the representation of node $v$ in $u$'s subgraph), distance information is additionally integrated, thus the local operations aggregate information $\{\!\!\{ \big( h_{uw}, d_{vw} \big) \mid w \in \mathcal{N}(v) \}\!\!\}$ and $\{\!\!\{ \big( h_{wv}, d_{uw} \big) \mid w \in \mathcal{N}(u) \}\!\!\}$ respectively.
- It adopts vertex-subgraph or subgraph-vertex pooling schemes (Zhang et al., 2023) to summarize node and subgraph features and produce the final graph representation.

It is noteworthy that GeoNGNN (without outer layers) represents a particular instance of general geometric subgraph GNNs, focusing solely on *intra-subgraph* message propagation. Beyond this scope, general geometric subgraph GNNs have the capability to engage in *inter-subgraph* message passing in diverse manners and adopt different pooling schemes. Now, we show that the whole family, *with at least one local aggregation*, is complete under exactly the same conditions as GeoNGNN.

**Theorem 5.3** (Completeness for general geometric subgraph GNNs). *When the general geometric subgraph layer number is larger than some constant $C$ (irrelevant to the node number), and the last two conditions specified in Theorem 5.1 are met, all general geometric subgraph GNNs in Definition 5.2 with at least one local aggregation are E(3)-complete.*

**Key idea.** It can be shown that any type of general geometric subgraph GNNs, which could employ any combination of aggregation and pooling schemes, can implement (Frasca et al., 2022) GeoNGNN, which is established as an E(3)-complete model in Theorem 5.1. Consequently, any general geometric subgraph GNN can also achieve E(3)-completeness under the same conditions as GeoNGNN.

Concerning prior research on traditional subgraph GNNs (Bevilacqua et al., 2021; Zhao et al., 2021; Qian et al., 2022; You et al., 2021; Frasca et al., 2022), their geometric counterparts can be defined correspondingly, by substituting all local aggregation schemes with *distance-aware* counterparts. We denote their geometric counterpart by prefixing *Geo* to their original names, for instance, GeoSUN Frasca et al. (2022). Consequently, it can be established that some of them precisely fall into the general definition 5.2, such as GeoOSAN (Qian et al., 2022), and for the others such as GeoGNN-AK (Zhao et al., 2021) and GeoSUN Frasca et al. (2022), while even though they do not exactly match the general definition, they can still implement GeoNGNN, thereby exhibiting E(3)-completeness (Theorem E.2). Please refer to Appendix E.2 for details about all of these.

We note that all subgraph GNNs in this section have been proven to be strictly weaker than 2-FWL (Cai et al., 1992) *in traditional graph learning context* (Zhang et al., 2023). Specifically, there exist pairs of unweighted graphs that can be distinguished by 2-FWL but remain indistinguishable by these subgraph GNNs. Moreover, these subgraph GNNs themselves establish a strict expressiveness hierarchy (Zhang et al., 2023). Interestingly, our findings reveal that *all of these discrepancies diminish* when these models are extended to geometric scenarios with point clouds by leveraging distance graphs. This phenomenon could be attributed to the low-rank nature of distance graphs,

whose intrinsic dimension is less than the point cloud space dimension $3n$ rather than equals to their ambient dimension $n^2$.

## 5.3 COMPLETENESS OF WELL-ESTABLISHED INVARIANT MODELS

Based on the completeness of GeoNGNN, we move on to establish the E(3)-completeness of several well-established invariant models, including DimeNet (Gasteiger et al., 2019), GemNet (Gasteiger et al., 2021) and SphereNet (Liu et al., 2021). These models do not exactly learn on subgraphs, however, can still be mathematically aligned with GeoNGNN. Formal descriptions of these models can be found in Appendix B.6.

**Theorem 5.4** (E(3)-Completeness of DimeNet, SphereNet, GemNet). *When the following conditions are met, DimeNet, SphereNet[4] and GemNet are E(3)-complete.*

- *The aggregation layer number is larger than some constant $C$ (irrelevant to the node number).*
- *They initialize and update all edge representations, i.e., $r_{embed} = +\infty$.*
- *They interact with all neighbors, i.e., $r_{int} = +\infty$.[5]*

**Key idea.** The key insight underlying is that all these models track edge representations ($h_{ij}^{\text{edge}}$ for edge $(i, j)$), which can be mathematically aligned with the node-subgraph representations tracked in GeoNGNN ($h_{ij}^{\text{subg}}$ for node $j$ in subgraph $i$). Moreover, the additionally incorporated features, such as angles, can all be equivalently expressed by multiple distances. Consequently, these three models can all implement GeoNGNN, and thereby achieving completeness.

Note that DimeNet is a relatively simpler invariant model that aggregates neighbor information in a *weaker manner* compared to existing 2-FWL-like complete geometric models such as 2F-DisGNN (Li et al., 2024). Specifically, when updating $h_{ij}$, DimeNet can be equivalently considered as aggregating $(h_{ki}, d_{kj})$ for a specific neighbor $k$, while 2F-DisGNN aggregates $(h_{ik}, h_{kj})$. The latter incorporates a more informative hidden representation $h_{kj}$, which is essential in the proof of 2F-DisGNN's completeness (Li et al., 2024; Delle Rose et al., 2023), instead of $d_{kj}$ in the former. Nevertheless, our findings first show that these models are equally E(3)-complete under fully-connected conditions. Consequently, future developments like GemNet that integrate higher-order geometry could be unnecessary in terms of boosting theoretical expressiveness under such conditions.

## 5.4 SUMMARIZATION AND DISCUSSION

**Fully-Connected Condition.** Thus far, we have characterized a broad collection of E(3)-complete invariant models under similar conditions. Among these conditions, the condition of *fully-connected* geometric graphs (or equivalently, *infinite* cutoffs) is required. Notably, this condition is consistent with all prior works that rigorously characterize invariant models/descriptors in the same sense, including (Kurlin, 2023; Widdowson & Kurlin, 2022; 2023; Li et al., 2024; Delle Rose et al., 2023; Hordan et al., 2024b;a). And due to significant local information loss during invariant message passing (Joshi et al., 2023; Du et al., 2024), even under this condition the characterization remains highly nontrivial. Indeed, removing the fully-connected condition would require conditions about specific forms of sparsity, which would be considerably more demanding than the overall connectivity typically required by equivariant models (Joshi et al., 2023; Du et al., 2024; Wang et al., 2024; Sverdlov & Dym, 2024) that can maintain local information through equivariant features. Such characterization is left for future work.

**Theoretical Characterization vs. Practical Use.** In practical scenarios, *efficiency* and *generalization* are often prioritized, and local/sparse connectivity can typically offer empirical advantages in these aspects (Musaelian et al., 2023). Thus, strictly adhering to the complete condition which requires full connectivity is unnecessary, and finding a balance is crucial, as demonstrated in our additional experiments in Appendix F (Figure 6). However, this does not diminish the importance of theoretical characterization, since theoretical completeness provides an *upper bound* for the parametric model's

---

[4]Here in SphereNet we do not consider the relative azimuthal angle $\varphi$, since SphereNet with $\varphi$ is not E(3)-invariant, while E(3)-completeness is defined on E(3)-invariant models. Note that the exclusion of $\varphi$ only results in weaker expressiveness.

[5]In DimeNet and GemNet, nodes $i$ or $j$ are excluded when aggregating neighbors for edge $ij$. The condition requires the inclusion of these end nodes.

potential–when combined with MLPs (Hornik et al., 1989), complete models can achieve universal approximation over continuous invariant functions (Hordan et al., 2024b; Li et al., 2024). This is analogous to the universal approximation property for MLPs (Hornik et al., 1989) and the Turing completeness for RNNs (Siegelmann & Sontag, 1992), where practical implementations do not (and typically cannot) satisfy all theoretical conditions, yet these results indicate their superiority over weaker structures that cannot achieve certain approximations even with unlimited resources.

**SE(3)-complete Counterpart.** Finally, we present a provable SE(3)-complete variant of GeoNGNN, capable of distinguishing chiral molecules, by making a minor modification to the distance features through the addition of a orientation sign. This is further detailed in Appendix H.

## 6    EXPERIMENTS

In this section, we conduct additional assessments to validate our theoretical claims. The first experiment aims to verify the conclusion that DisGNN is nearly complete by assessing the proportion of its unidentifiable cases in real-world point clouds. The second experiment is designed to evaluate whether the complete models consistently demonstrate separation power for challenging pairs of point clouds, where numerical precision may influence the outcomes. We provide more evaluations of GeoNGNN on practical molecular-relevant tasks in Appendix F, which could offer further insights.

### 6.1    ASSESSMENT OF UNIDENTIFIABLE CASES OF DISGNN

In Theorem 4.3, we have rigorously shown the rarity of symmetric point cloud sets, which are supersets of the unidentifiable set of DisGNN. However, since real-world point clouds are typically subject to *slight noise*, requiring exact symmetry would be overly restrictive. Therefore, we address a more challenging setting by explicitly accounting for noise and evaluating the rarity of **relaxed symmetric** point clouds in practical scenarios. To this end, we first define two noise tolerances that allow us to relax the exact $\mathcal{C}$- and $\mathcal{D}$-symmetry.

**Noise Tolerances.** 1) The *rounding number* $r$ for distance-related calculations. When applying algorithms $\mathcal{C}$ and $\mathcal{D}$, we round the distance values to $r$ decimal places for robustness to noise. 2) The *deviation error* $\epsilon$. In Definition 4.1, we define symmetry by the set of extended "mass" centers. We provide an equivalent definition in Proposition C.4, which shows that a point cloud is $\mathcal{A}$-symmetric if and only if the geometric centers of all sub-point clouds, partitioned by distinct node features of $\mathcal{A}$, coincide. This alternative definition allows us to define another noise tolerance: we now consider a point cloud as symmetric if all sub-point clouds' centers described above lie within a ball of radius $\epsilon$. Together, $r$ and $\epsilon$ define a relaxed symmetry that accounts for noise in real-world point clouds. When $r \to +\infty$ and $\epsilon \to 0$, the relaxed symmetry becomes the exact theoretical symmetry.

We are now ready to evaluate the proportion of relaxed symmetric point clouds in real-world datasets. We select two representative datasets, namely QM9 (Ramakrishnan et al., 2014; Wu et al., 2018) and ModelNet40 (Wu et al., 2015), for this assessment. The QM9 dataset comprises approximately 130K molecules, while the ModelNet40 dataset consists of roughly 12K real-world point clouds categorized into 40 classes, including objects such as chairs. We first rescale all point clouds, and then fix the rounding number to small values and evaluate the proportion of symmetric point clouds with respect to different values of $\epsilon$. Please refer to Appendix C.3 for detailed settings.

Results are shown in Figure 2. Here are several key observations: 1) For the QM9 dataset, even with the largest deviation error ($10^{-1}$), the proportion of $\mathcal{C}$- and $\mathcal{D}$-symmetric point clouds is less than $\sim$0.15% of the entire dataset. Since $\mathcal{C}$- and $\mathcal{D}$-symmetric point cloud sets are supersets of the unidentifiable set of DisGNN, the unidentifiable proportion is therefore no more than $\sim$0.15%. With the strictest deviation error, only 0.0046% of the graphs (6 out of $\sim$130K) exhibit $\mathcal{D}$-symmetry. 2) For the ModelNet40 dataset, when the deviation error is less than $10^{-2}$, only 1 point cloud exhibit $\mathcal{C}$-symmetry. Moreover, no $\mathcal{D}$-symmetric point clouds are found across all deviation errors.

To summarize, the statistical results suggest that the occurrence of unidentifiable graphs in DisGNN is practically negligible in real-world scenarios, even when the criterions are significantly relaxed. This supports the conclusion that DisGNN is almost E(3)-complete.

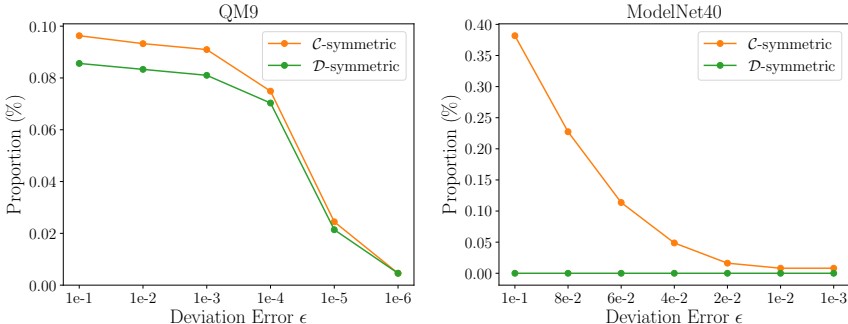

Figure 2: Assessment of symmetric point clouds in real-world datasets. (a) Proportion of symmetric point clouds in QM9 ($r = 2$). (b) Proportion of symmetric point clouds in ModelNet40. The proportion of $\mathcal{D}$-symmetric point clouds in ModelNet40 is zero across all deviation errors ($r = 1$).

## 6.2 SEPARATION POWER ON SYNTHETIC POINT CLOUD PAIRS

To construct hard-to-distinguish counterexamples, we develop a geometric expressiveness dataset based on the counterexamples proposed by Li et al. (2024). The synthetic dataset consists of 10 isolated and 7 combinatorial counterexamples. Each counterexample is composed of a pair of highly symmetric point clouds (all of which are $\mathcal{D}$-symmetric, as described in Section 4), which are non-isomorphic yet *indistinguishable by DisGNN*. We provide some examples in Figure 7, and please refer to Appendix G for further settings.

Table 1: Separation results on the constructed geometric expressiveness dataset. Models for which we have theoretically established completeness are highlighted in gray.

|  | Invaraint | | | | | | Equivaraint | |
| --- | --- | --- | --- | --- | --- | --- | --- | --- |
|  | SchNet | DisGNN | DimeNet | SphereNet | GemNet | GeoNGNN | PaiNN | MACE |
| Isolated (10 cases) | 0% | 0% | 100% | 100% | 100% | 100% | 100% | 100% |
| Combined (7 cases) | 0% | 0% | 100% | 100% | 100% | 100% | 100% | 100% |

Results are presented in Table 1. As shown, our established complete models can all distinguish these challenging pairs effectively, whereas DisGNN and SchNet cannot. This supports our theory and indicates that numerical precision is not impacting these complete models' separation ability here. Interestingly, two equivariant models, PaiNN (Schütt et al., 2021) and MACE (Batatia et al., 2022) that leverages vectors and high-order tensors respectively, also effectively distinguish all the pairs. This prompts further question of whether they are also complete, especially under sparse connections or *finite* tensor orders, and requires further investigation.

## 7 CONCLUSION AND LIMITATION

**Conclusion.** In this study, we thoroughly analyze a wide range of invariant models' theoretical expressiveness under fully-connected condition. Specifically, we rigorously characterize the expressiveness of DisGNN, showcasing that all its unidentifiable cases exhibit $\mathcal{D}$-symmetry and have a measure of 0. We then establish a large family of E(3)-complete models, which encompasses GeoNGNN, geometric subgraph GNNs, as well as three established models - DimeNet, GemNet, and SphereNet. This contributes significantly to a comprehensive understanding of invariant geometric models. Moreover, the newly introduced geometric subgraph GNNs notably enlarge the design space of expressive geometric models. Experiments further validate our theoretical findings.

**Limitation.** The rigorous E(3)-completeness characterization for invariant models is under the conditions of fully connected graphs, which can be limited and have been throughout discussed in Section 5.4. The extent to which invariant models can exhibit high expressiveness on general *sparse* graphs remains an open question that needs further investigation. Additionally, future research is needed to characterize the expressiveness of vector models that rely solely on 1-order equivariant representations and adopt atom-level representations, such as PaiNN (Schütt et al., 2021), which show promising experimental results but still lack theoretical expressiveness characterization.

ACKNOWLEDGMENT

This work is supported by the National Natural Science Foundation of China (62276003).

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

## A EXTENDED DISCUSSION OF RELATED WORK

**Invaraint geometric models** To leverage the rich 3D geometric information contained in point clouds in permutation- and E(3)-invariant manner, early work Schütt et al. (2018) integrated 3D Euclidean distance into the MPNN framework (Gilmer et al., 2017) and aggregate geometric information iteratively. Nevertheless, this model exhibits a restricted capacity to distinguish non-isomorphic point clouds even on fully-connected graphs as substantiated by Li et al. (2024); Pozdnyakov & Ceriotti (2022); Pozdnyakov et al. (2020); Hordan et al. (2024b). Subsequent invariant models (Gasteiger et al., 2019; 2020; 2021; 2022; Liu et al., 2021; Wang et al., 2022; Li et al., 2024) have endeavored to enhance their geometric expressiveness through the incorporation of carefully designed high-order geometric features (Gasteiger et al., 2019; 2020; 2021; 2022), adopting local spherical representations (Wang et al., 2022) or higher-order distance features (Li et al., 2024). These designs have greatly improved the models' performance on downstream tasks, however, most of them lack theoretical guarantees of geometric completeness over the whole point cloud spaces.

## B PROOF OF MAIN CONCLUSIONS IN MAIN BODY

### B.1 MEASURE OF SYMMETRIC AND UNIDENTIFIABLE POINT CLOUDS

In this subsection, we aim to prove that the Lebesgue measure on $\mathbb{R}^{n \times 3}$ of the $\mathcal{C}$-symmetric point cloud set, the $\mathcal{D}$-symmetric point cloud set, and the unidentifiable point cloud set are all **zero**.

**Theorem 4.3** *(Unidentifiable set of DisGNN has measure zero) The Lebesgue measure on $\mathbb{R}^{n \times 3}$ of the $\mathcal{C}$-symmetric, $\mathcal{D}$-symmetric, and unidentifiable point cloud sets is zero.*

We first show that the $\mathcal{C}$-symmetric point cloud set has measure zero. We denote the set containing all $\mathcal{C}$-symmetric point clouds as $\mathbb{R}_{\mathcal{C}}^{n \times 3}$. Note that

$$\mathbb{R}_{\mathcal{C}}^{n \times 3} \subseteq \mathbb{R}_{\text{super } \mathcal{C}}^{n \times 3} := \{P \in \mathbb{R}^{n \times 3} \mid \exists i \neq j \in [n] \text{ such that } \text{cond}_{i,j} \text{ holds}\},$$

where

$$\text{cond}_{i,j} := \|p_i - \text{mean}_k(p_k)\|^2 - \|p_j - \text{mean}_k(p_k)\|^2 = 0,$$

which is essentially a non-trivial polynomial equality constraint. Here, $\text{mean}_k(p_k)$ calculates the geometric center of the point cloud, and $\|p_i - \text{mean}_k(p_k)\|$ is essentially the distance between $i$ and the geometric center. The set $\mathbb{R}_{\text{super } \mathcal{C}}^{n \times 3}$ is a superset of $\mathbb{R}_{\mathcal{C}}^{n \times 3}$, which can be validated as follows:

Assume some element $P$ belongs to $\mathbb{R}_{\mathcal{C}}^{n \times 3}$ but not to $\mathbb{R}_{\text{super } \mathcal{C}}^{n \times 3}$. Then, we have

$$\forall i \neq j \in [n], \quad \text{cond}_{i,j} \text{ for } P \text{ does not hold}.$$

This essentially means that all the nodes are embedded with different features after applying the $\mathcal{C}$ algorithm. Then, it is obvious that $P$ is $\mathcal{C}$-asymmetric, which contradicts the assumption that $P \in \mathbb{R}_{\mathcal{C}}^{n \times 3}$. Hence, we conclude that

$$\mathbb{R}_{\mathcal{C}}^{n \times 3} \subseteq \mathbb{R}_{\text{super } \mathcal{C}}^{n \times 3}.$$

The set $\mathbb{R}_{\text{super } \mathcal{C}}^{n \times 3}$ defines an algebraic manifold with a non-trivial polynomial equality constraint, thus having dimension at most $3n - 1$. Therefore, according to (Mityagin, 2015; Hordan et al., 2024b), the set $\mathbb{R}_{\text{super } \mathcal{C}}^{n \times 3}$ has measure zero, which implies that $\mathbb{R}_{\mathcal{C}}^{n \times 3}$ also has measure zero.

According to Proposition C.1, since $\mathbb{R}_{\mathcal{C}}^{n \times 3}$ is the largest set under consideration, it follows that both the $\mathcal{D}$-symmetric point cloud set and the unidentifiable point cloud set also have measure zero.

### B.2 PREPARATION FOR RECONSTRUCTION PROOF

In the following proof, we particularly focus on models' ability to encode input representations without significant information loss. We provide a formal definition of an important concept *derive* below:

**Definition B.1.** (**Derive**) Given input representations $P \in \mathbb{P}$, let $f : \mathbb{P} \to \mathbb{O}_1$ and $g : \mathbb{P} \to \mathbb{O}_2$ be two property encoders defined on $\mathbb{P}$. If there exists a function $h : \mathbb{O}_1 \to \mathbb{O}_2$ such that for all $P \in \mathbb{P}$, $h(f(P)) = g(P)$, we say that $f$ can derive $g$. With a slight abuse of notation, we say that $f(P)$ can derive $g(P)$, denoted as $f(P) \to g(P)$.

For example, consider the space $\mathbb{P} = \mathbb{R}^{n \times 3}$ comprising all point clouds of size $n$. Let $f$ denote an encoder that computes $f(P) = (\sum_{i=1}^{n} p_i, n)$, which represents the sum of the nodes' coordinates and the number of nodes, while $g$ being an encoder calculating the geometric center of $P$, it follows that $f(P) \to g(P)$, since we can calculate the geometric center from the sum of the nodes' coordinates and the number of nodes. In the following analysis, we consider $\mathbb{P} = \mathbb{R}^{n \times 3}$ by default. Notice that derivation exhibits transitivity: if $f_1 \to f_2$ and $f_2 \to f_3$, it follows that $f_1 \to f_3$.

Intuitively, "derive" is a concept describing the relation of two properties, i.e., whether property $f(P)$ contains all the information needed to calculate $g(P)$. Obviously, if an encoder $f$ can embed all the information needed for reconstructing a given point cloud, then it is E(3)-complete:

**Proposition B.2.** *Consider $\mathbb{P} = \mathbb{R}^{n \times 3}$ being the space of all point clouds of size $n$, and $f$ satisfying permutation- and E(3)-invariance, then: $f(P) \to P$ up to permutation and Euclidean isometry $\iff$ $f$ is E(3)-complete.*

*Proof.* We first prove the reverse direction. Since $f$ is E(3)-complete, it will give distinct graph embeddings/features $s \in \mathbb{R}^d$ for non-isomorphic finite-size point clouds. We can thus *let $h : \mathbb{R}^d \to \mathbb{R}^{n \times 3}$* be a mapping which exactly maps the graph embeddings to the original point clouds up to permutation and Euclidean isometry. The existence of such $h$ implies that $f(P) \to P$ up to permutation and Euclidean isometry.

For the forward direction, we prove by contradiction. Assume that $f$ is not E(3)-complete, i.e., there exists two non-isomorphic point clouds $P_1, P_2$ such that $f(P_1) = f(P_2)$. Since $f(P) \to P$ up to permutation and Euclidean isometry, we have that there exists a function $h$ that can reconstruct the point cloud from the embedding. However, since $f(P_1) = f(P_2)$, their reconstruction $h(f(P_1)), h(f(P_2))$ will be the same up to permutation and Euclidean isometry, which contradicts the assumption that $P_1$ and $P_2$ are non-isomorphic. Therefore, $f$ is E(3)-complete. $\square$

## B.3 PROOF OF THEOREM 4.2

As indicated in Figure 1(b), all $\mathcal{D}$-symmetric point clouds are also $\mathcal{C}$-symmetric, which is a direct conclusion from Delle Rose et al. (2023); Li et al. (2024). We formally prove it here.

**Proposition B.3.** *($\mathcal{D}$-symmetry implies $\mathcal{C}$-symmetry) For any point cloud $P \in \mathbb{R}^{n \times 3}$, if $P$ is $\mathcal{D}$-symmetric, then $P$ is $\mathcal{C}$-symmetric.*

*Proof.* As indicated by the conclusions of Delle Rose et al. (2023); Li et al. (2024), with the "derive" notations we introduced in the previous subsection, we have that: $x_i^{\mathcal{D}} \to x_i^{\mathcal{C}}$, where $P \in \mathbb{R}^{n \times 3}$. This essentially implies that DisGNN's node-level output contains the information of the distance between nodes to the geometric center. As a direct consequence, for any point cloud $P$, the node partition based on node features $X^{\mathcal{D}}$ will be no coarser than that based on $\mathcal{C}(P)$.

If $P$ is $\mathcal{D}$-symmetric, the center set $\mathcal{D}^{\text{set}}(P)$ will obtain only one element. By definition, the cardinality of the center sets only depends on the node features partitions, and no coarser partition will lead to no smaller center set. Therefore, $\mathcal{C}^{\text{set}}(P)$ will also contain only one element, and thus $P$ is $\mathcal{C}$-symmetric. $\square$

Based on Proposition B.3, to prove Theorem 4.2, it suffices to prove that point clouds that are not $\mathcal{D}$-symmetric are identifiable. To prove this, we first show the ability of DisGNN to learn global geometric information:

**Lemma B.4.** *(Locate $\mathcal{A}$ centers) Given a point cloud $P \in \mathbb{R}^{n \times 3}$ with node features $X^{\mathcal{A}}$ calculated by algorithm $\mathcal{A}$. Denote the $\mathcal{D}$-center calculated by mass function $m$ as $c^m \in \mathbb{R}^{n \times 3}$. With a bit notation abuse, we use $d_{ic^m}$ to represent the distance between node $i$ and $c^m$.*

*We now run a DisGNN on $(P, X^{\mathcal{A}})$, and denote the node $i$'s representations at layer $l$ as $h_i^{(l)}$ (we let $h_i^{(0)} = x_i^{\mathcal{A}}$). Then we have:*

- *(Node-center distance) Given an arbitrary mass function $m$, $h_i^{(2)}$ can derive $d_{i,c^m}$.*

- *(Center-center distance) Given two arbitrary mass functions $m_1$ and $m_2$, $\{\!\!\{ h_i^{(2)} \mid i \in [n] \}\!\!\}$ can derive $d_{c^{m_1}, c^{m_2}}$.*

- *(Counting centers) $\{\!\!\{ h_i^{(2)} \mid i \in [n] \}\!\!\}$ can derive $\mathbb{1}_{|\mathcal{A}^{set}(P)|=1}$.*

This proposition essentially extends the Barycenter Lemma in Delle Rose et al. (2023); Li et al. (2024) to the general case of centers defined by $\mathcal{D}$ features. Such information can enable DisGNN to obtain many global geometric features, facilitating it to distinguish non-isomorphic point clouds. Moreover, the ability to count centers is crucial for DisGNN to distinguish any point cloud that is not $\mathcal{A}$-symmetric from a $\mathcal{A}$-symmetric one, thereby achieving "identifiability". Now we give the formal proof.

*Proof.*

**Node-center distance**  We first denote the weighted distance profile of node $i$ as $f(m, i)$, which is defined as $f(m, i) = \sum_{j \in [n]} m(x_j^{\mathcal{A}}) d_{ij}^2$. According to the aggregation scheme of DisGNN, $f(m, i)$ can be derived from $h_i^{(1)}$, since $h_i^{(1)} \to (h_i^{(0)}, \{\!\!\{ (h_j^{(0)}, d_{ij}) \mid j \in [n] \}\!\!\}) \to \{\!\!\{ (x_j^{\mathcal{A}}, d_{ij}) \mid j \in [n] \}\!\!\} \to \sum_{j \in [n]} m(x_j^{\mathcal{A}}) d_{ij}^2$ when $m$ is given. All the $\to$ here hold because we assume that DisGNN uses injective function forms and parameterizations (thereby resulting in no information loss), and $j \in [n]$ since DisGNN treats point clouds as fully connected distance graphs, as mentioned at the beginning of the main body.

$f(m, i)$ can be further decomposed as:

$$
\begin{aligned}
f(m, i) &= \sum_{j \in [n]} m(x_j^{\mathcal{A}}) d_{ij}^2 \\
&= \sum_{j \in [n]} m(x_j^{\mathcal{A}}) \|p_i - p_j\|^2 \\
&= \sum_{j \in [n]} m(x_j^{\mathcal{A}}) \|p_i - c^m + c^m - p_j\|^2 \\
&= \sum_{j \in [n]} m(x_j^{\mathcal{A}}) \left( \|p_i - c^m\|^2 + \|p_j - c^m\|^2 - 2\langle p_i - c^m, p_j - c^m \rangle \right) \\
&= M\|p_i - c^m\|^2 + \sum_{j \in [n]} m(x_j^{\mathcal{A}}) \|p_j - c^m\|^2 - 2\langle p_i - c^m, \sum_{j \in [n]} m(x_j^{\mathcal{A}})(p_j - c^m) \rangle
\end{aligned}
$$

where $M = \sum_{j \in [n]} m(x_j^{\mathcal{A}})$ and $\langle \rangle$ denotes the inner product.

By definition of $c^m$, $\sum_{j \in [n]} m(x_j^{\mathcal{A}})(p_j - c^m) = 0$. Therefore, we have:

$$
f(m, i) = M\|p_i - c^m\|^2 + \sum_{j \in [n]} m(x_j^{\mathcal{A}}) \|p_j - c^m\|^2
$$

Therefore, the distance from $i$ to $c^m$ can be calculated as:

$$
\frac{1}{M} \left( f(m, i) - \frac{1}{2M} \sum_{j \in [n]} m(x_j^{\mathcal{A}}) f(m, j) \right) = \|p_i - c^m\|^2 = d_{i,c^m}^2 \tag{5}
$$

And since

$$
\begin{aligned}
h_i^{(2)} &\to (h_i^{(1)}, \{\!\!\{ (h_j^{(1)}, d_{ij}) \mid j \in [n] \}\!\!\}) \\
&\to (f(m, i), \{\!\!\{ (m(x_j^{\mathcal{A}}), f(m, j)) \mid j \in [n] \}\!\!\}), \tag{6}
\end{aligned}
$$

we finally have: $h_i^{(2)} \to d_{i,c^m}$.

**Center-center distance** Based on the conclusion of Node-center distance, at round 2, each node is aware of its distance to the two centers defined by $m_1$ and $m_2$:

$$h_i^{(2)} \to (d_{i,c^{m_1}}, d_{i,c^{m_2}}). \tag{7}$$

Therefore,

$$\begin{aligned} h_i^{(2)} &\to \|p_i - c^{m_1}\|^2 - \|p_i - c^{m_2}\|^2 \\ &= \langle c^{m_2} - c^{m_1}, 2p_i - c^{m_1} - c^{m_2} \rangle \end{aligned}$$

.

Therefore, the multiset of all node representations at round 2 can derive the distance between the two centers:

$$\begin{aligned} \{\!\{ h_i^{(2)} \mid i \in [n] \}\!\} &\to \{\!\{ (h_i^{(2)}, m_1(x_i^{\mathcal{A}})) \mid i \in [n] \}\!\} \\ &\to \frac{1}{M_1} \sum_{i \in [n]} m_1(x_i^{\mathcal{A}}) \langle -c^{m_1} + c^{m_2}, 2p_i - c^{m_1} - c^{m_2} \rangle \\ &= \langle -c^{m_1} + c^{m_2}, \big( \frac{2}{M_1} \sum_{i \in [n]} m_1(x_i^{\mathcal{A}}) p_i \big) - c^{m_1} - c^{m_2} \rangle \\ &= \langle -c^{m_1} + c^{m_2}, 2c^{m_1} - c^{m_1} - c^{m_2} \rangle \\ &= \langle -c^{m_1} + c^{m_2}, c^{m_1} - c^{m_2} \rangle \\ &= -\|c^{m_1} - c^{m_2}\|^2, \end{aligned} \tag{8}$$

where $M_1 = \sum_{i \in [n]} m_1(x_i^{\mathcal{A}})$.

**Counting centers** In this case, we are not given any specific mass function $m$. It seems like we need to iterate all possible mass functions to determine the number of centers, which is infeasible due to the infinite number of mass functions. However, thanks to Proposition C.4, we can determine whether $\mathcal{A}^{\text{set}}(P)$'s cardinality is 1 (i.e., whether $P$ is $\mathcal{A}$-symmetric) by checking whether the geometric centers of all corresponding sub point clouds coincide with each other.

Now, assume that there are $K$ types of node features in $X^{\mathcal{A}}$, and we use $m_k$ to represent the mass function that maps all $x_i^{\mathcal{A}}$ to 1 if $x_i^{\mathcal{A}}$ is the $k$-th type of features and 0 otherwise, and use $c$ solely to represent the coordinate of geometric center (unweighted average coordinates of all points).

Based on the conclusion of center-center distance part, we know that $\{\!\{ h_i^{(2)} \mid i \in [n] \}\!\}$ can derive the distance between $c$ to all $c^{m_k}$, $k \in [K]$, i.e., $\{\!\{ h_i^{(2)} \mid i \in [n] \}\!\} \to \{ d_{c,c^{m_k}} \mid k \in [K] \}$. We can easily check the cardinality of the set $\{ d_{c,c^{m_k}} \mid k \in [K] \}$ to determine whether the geometric centers of all these $K$ sub point clouds coincide, and according to Proposition C.4, this is equivalent to determining whether $|\mathcal{A}^{\text{set}}(P)| = 1$.

Therefore, $\{\!\{ h_i^{(2)} \mid i \in [n] \}\!\} \to \mathbb{1}_{|\mathcal{A}^{\text{set}}(P)|=1}$. $\qquad\square$

Notice that DisGNN can itself calculate $\mathcal{C}$ and $\mathcal{D}$ feature, therefore, it can simply take **unlabeled point clouds as input**, and eventually is able to learn distance information related to all centers in set $\mathcal{D}^{\text{set}}(P)$. This is concluded in the following corollary:

**Corollary B.5.** *Given an unlabelled point cloud $P \in \mathbb{R}^{n \times 3}$. DisGNN can derive the "node-center", "center-center" and "center count" information defined in Lemma B.4 w.r.t. center sets $\mathcal{A}^{set}(P)$ after $k$ rounds, where $\mathcal{A}$ and $k$ can be:*

- *$\mathcal{A} = NULL$ (i.e., $\mathcal{A}$ assigns the same features to all nodes), $k = 2$. In such case, $\mathcal{A}^{set}(P)$ contains only the geometric center of $P$.*

- *$\mathcal{A} = \mathcal{C}$, $k = 4$.*

- *$\mathcal{A} = \mathcal{D}$, $k$'s upper bound is $(n + 2)$ ($n$ means that we need to first ensure that DisGNN stabilizes).*

In the case of where $\mathcal{A}$ is NULL, the conclusion degenerates to the Barycenter Lemma in Delle Rose et al. (2023); Li et al. (2024).

We now give an essential lemma. In previous work (Pozdnyakov & Ceriotti, 2022; Li et al., 2024), it has been shown that DisGNN is E(3)-incomplete, i.e., there exist pairs of non-isomorphic point clouds that DisGNN cannot distinguish. This essentially means that we cannot prove that DisGNN's output $s$ can derive the point cloud $P$ up to permutation and Euclidean isometry. However, in the following lemma and corollary, we will show that, if we restrict the **underlying space $\mathbb{P}$ to be all point clouds that are not $\mathcal{A}$-symmetric** for arbitrary $\mathcal{A}$ algorithm, we can prove that DisGNN's output can derive the input point cloud up to permutation and Euclidean isometry. This implies that DisGNN is complete on such subsets.

**Lemma B.6.** *Given a point cloud $P \in \mathbb{R}_{not\ \mathcal{A}}^{n \times 3}$ with node features $X^{\mathcal{A}}$ calculated by algorithm $\mathcal{A}$, where $\mathbb{R}_{not\ \mathcal{A}}^{n \times 3}$ contains all $\mathcal{A}$-asymmetric point clouds in $\mathbb{R}^{n \times 3}$. Then, 4-round DisGNN's output $s$ can derive $P$ up to permutation and Euclidean isometry.*

*Proof.* By definition of "derive", we describe the reconstruction function that takes DisGNNs' output $s$ as input and produces the point cloud $P$ up to permutation and Euclidean isometry. We initially assume that the point cloud *does not degenerate into 2D*, which is a trivial case and will be discussed at the end of the proof.

Since $P$ is $\mathcal{A}$-asymmetric, *there are at least two centers* in $\mathcal{A}^{set}(P)$. We now take two arbitrary such centers and denote them as $c_1$ and $c_2$. According to Corollary B.5, 2-round DisGNN's node-level features can derive the distance information related to these two centers, namely, $h_i^{(2)} \to (d_{i,c_1}, d_{i,c_2}, d_{c_1,c_2})$. In the following, we show how **DisGNN can reconstruct the whole geometry based on triangular distance encoding**.

By definition of DisGNN and the injectivity assumption, its output $s$ can derive the multiset $\{\!\!\{ h_i^{(4)} \mid i \in [n] \}\!\!\}$. And since $h_i^{(4)} \to h_i^{(3)} \to h_i^{(2)} \to (d_{i,c_1}, d_{i,c_2}, d_{c_1,c_2})$, we know that from $h_i^{(4)}$ we can reconstruct the triangle formed by $i, c_1, c_2$. Consequently, it is feasible to determine the following dihedral angle list from $h_i^{(4)}$:

$$
\begin{aligned}
h_i^{(4)} = \text{HASH}(h_i^{(3)}, &\{\!\!\{ (h_j^{(3)}, d_{ij}) \mid j \in [n] \}\!\!\}) \\
&\to \{\!\!\{ (h_i^{(3)}, h_j^{(3)}, d_{ij}) \mid j \in [n] \}\!\!\} \\
&\to \{\!\!\{ (h_i^{(2)}, h_j^{(2)}, d_{ij}) \mid j \in [n] \}\!\!\} \\
&\to \{\!\!\{ (d_{ij}, d_{i,c_1}, d_{i,c_2}, d_{j,c_1}, d_{j,c_2}, d_{c_1,c_2}) \mid j \in [n] \}\!\!\} \\
&\to \{\!\!\{ \theta(ic_1c_2, jc_1c_2) \mid j \in [n] \}\!\!\}.
\end{aligned}
$$

Here, $\theta(ic_1c_2, jc_1c_2)$ represents the dihedral angle formed by plane $ic_1c_2$ and plane $jc_1c_2$. If $i$ or $j$ lies on the line $c_1c_2$, or the angle $\theta$ is 0, we define/overwrite $\theta$ as $+\infty$.

Given this observation, we now conduct a search across the multiset $\{\!\!\{ h_i^{(4)} \mid i \in [n] \}\!\!\}$ to find the node $x$ which can derive the smallest $\theta(xc_1c_2, yc_1c_2)$ relative to some other node $y$ (In case of multiple minimal angles, an arbitrary $x$ is chosen). Denote the minimal angle as $\alpha$, and we record the corresponding $h_x^{(4)}$. And note that since $\theta(xc_1c_2, yc_1c_2)$ is derived from $(h_x^{(4)}, h_y^{(3)}, d_{xy})$, the corresponding $(h_y^{(3)}, d_{xy})$ in the calculation of $h_x^{(4)}$ can also be recorded. We now aim to prove that the entire geometry can be reconstructed using $h_x^{(4)}$ and the corresponding $(h_y^{(3)}, d_{xy})$.

First of all, having found the node $y$ in the multiset, we can calculate the exact 3D coordinates of nodes $x, y, c_1, c_2$ *up to Euclidean isometry* given $(h_x^{(4)}, h_y^{(3)}, d_{xy})$. This is because $(h_x^{(4)}, h_y^{(3)}, d_{xy}) \to (d_{xy}, d_{x,c_1}, d_{x,c_2}, d_{y,c_1}, d_{y,c_2}, d_{c_1,c_2})$. At this stage, the coordinates of $n-2$ nodes remain undetermined.

We proceed to traverse all nodes in the multiset $\{\!\!\{ (h_j^{(1)}, d_{xj}) \mid j \in [n] \}\!\!\}$ that can be derived from $h_x^{(4)}$ by definition. For each node $j$, with information $(h_x^{(1)}, h_j^{(1)}, d_{xj})$, we can derive $(d_{jx}, d_{j,c_1}, d_{j,c_2})$. Consequently, we can determine the 3D coordinates of $j$ to at most two positions, which are symmetrically mirroring w.r.t. the plane $xc_1c_2$. Notably, $j$'s position is unique if and only if it lies on

the plane $xc_1c_2$ (denoted as $x$-plane). We first identify all nodes on the $x$-plane, since their calculated positions are unique.

Additionally, from $h_y^{(3)}$, we can also derive $(d_{jy}, d_{j,c_1}, d_{j,c_2})$ for each node $j$, allowing us to identify all nodes on the $y$-plane.

The coordinates of these nodes are added to the "known" set of nodes $K$, while the remaining nodes are labeled as "unknown."

It is established that there are no "unknown" nodes in the interior (as $\theta_{xc_1c_2, yc_1c_2}$ is minimal) and border (since we have identified these points) of the $x$-plane and $y$-plane. We denote the interior and border of the two planes as $P_0$.

Now, we reflect the $y$-plane mirrorly w.r.t. the $x$-plane which yields the $p_{-1}$-plane. The interior and border formed by $p_{-1}$-plane and $x$-plane are denoted as $P_{-1}$. Since there are no "unknown" nodes in $P_0$, the remaining undetermined nodes in $P_{-1}$ can be determined from $h_x^{(4)}$, as their another possible position calculated by $h_x^{(4)}$ lie in the "dead area" $P_0$, where it is ensured to be empty (no undetermined nodes left). Similarly, reflecting the $x$-plane w.r.t. the $y$-plane produces the $p_1$-plane, and the corresponding area by $p_1$-plane and $y$-plane is denoted as $P_1$. Nodes in $P_1$ are determined likewise from $h_y^{(3)}$. This process continues by reflecting the $p_1$-plane w.r.t. the $x$-plane to obtain the $p_{-2}$-plane, determining the nodes in $P_{-2}$, and so forth, until all $P$ areas collectively cover the entire 3D space. Consequently, all nodes can be determined, and the geometric structure can be reconstructed.

When the point cloud degenerates into 2D, we can initially derive a node $i$'s representation $h_i^{(4)}$ from the output $s$ of DisGNN that does not lie on the line $c_1c_2$. By leveraging $h_i^{(4)}$, we can extract the necessary distance information to compute the coordinates of $i$, $c_1$, and $c_2$ up to Euclidean isometry. Given that $h_i^{(4)} \to \{\!\!\{ (h_j^{(3)}, d_{ij}) \mid j \in [n] \}\!\!\}$, we can derive $(d_{ij}, d_{jA_1}, d_{jA_2})$ from each $(h_j^{(3)}, d_{ij})$. Consequently, the coordinates of node $j$ can be uniquely determined (2D setting). And therefore, the complete geometry can be reconstructed up to Euclidean isometry.

$\square$

As an insightful takeaway from Lemma B.6, we notice that the key in reconstruction is the initial **triangular distance encoding** that can be captured by DisGNN. The triangular distance encoding enhances each node $i$'s feature with distance information $(d_{ic_1}, d_{ic_2}, d_{c_1c_2})$, where $c_1$ and $c_2$ are two global *anchors*. In Lemma B.6, the two anchors are from $\mathcal{A}$ center sets, and DisGNN can capture the corresponding distance encoding according to Lemma B.4. In Section 5.1, we show that we can mark a node as an additional anchor, which can also be captured by DisGNN due to the distinct mark. *Other designs may also be proposed by considering this insightful idea.*

Similarly to Corollary B.5, DisGNN can take **unlabelled point clouds** as input and finally reconstruct them when the unlabelled point clouds are $\mathcal{C}$-asymmetric or $\mathcal{D}$-asymmetric.

**Corollary B.7.** *Given a point cloud $P \in \mathbb{R}^{n \times 3}$, assume that $P$ is $\mathcal{A}$-asymmetric. Then, $k$-round DisGNN's output $s$ can derive $P$ up to permutation and Euclidean isometry, where $\mathcal{A}$ and $k$ can be:*

- $\mathcal{A} = \mathcal{C}$, $k = 6$.

- $\mathcal{A} = \mathcal{D}$, $k$'s upper bound is $(n + 4)$.

Now, we are ready to prove Theorem 4.2 based on all the above propositions and theorems.

**Theorem 4.2.** *(Asymmetric point clouds are identifiable) Let $\mathcal{C}$ denote the center distance encoding and $\mathcal{D}$ the DisGNN encoding. Then, given an arbitrary point cloud $P \in \mathbb{R}^{n \times 3}$, $P$ is $\mathcal{C}$-asymmetric $\Rightarrow P$ is $\mathcal{D}$-asymmetric $\Rightarrow P$ can be identified by DisGNN.*

*Proof.* The first $\Rightarrow$ is proved in Proposition B.3.

It suffices to show that $P$ is $\mathcal{D}$-asymmetric $\Rightarrow P$ can be identified by DisGNN.

Now, given another $P' \in \mathbb{R}^{n \times 3}$, there are two cases:

- $P'$ is $\mathcal{D}$-asymmetric. As a direct consequence of Lemma B.6, Corollary B.7 and Proposition B.2, DisGNN can distinguish $P$ and $P'$.

- $P'$ is $\mathcal{D}$-symmetric. According to Lemma B.4 and Corollary B.5, DisGNN's output for $P$ and $P'$ can derive $\mathbb{1}_{|\mathcal{D}^{\text{set}}(P)|=1}$ and $\mathbb{1}_{|\mathcal{D}^{\text{set}}(P')|=1}$ respectively. Since $P$ is $\mathcal{D}$-asymmetric while $P'$ is, by definition, we know that $\mathbb{1}_{|\mathcal{D}^{\text{set}}(P)|=1} \neq \mathbb{1}_{|\mathcal{D}^{\text{set}}(P')|=1}$, and thus DisGNN can distinguish $P$ and $P'$.

$\square$

### B.4 PROOF OF THEOREM 5.1

**Theorem 5.1** *(E(3)-Completeness of GeoNGNN) When the following conditions are met, GeoNGNN is E(3)-complete:*

- $N_{in} >= 5$ *and* $N_{out} >= 0$ *(where* $0$ *indicates that the outer GNN only performs final pooling).*
- *The distance graph is fully-connected (*$r_{cutoff} = +\infty$*).*
- *All subgraphs are the original graph (*$r_{sub} = +\infty$*).*

*Proof.* Based on the condition that $r_{\text{sub}} = \infty$, all subgraphs are exactly the original distance graph, with the exception that the central node (the subgraph around which is generated) is explicitly marked.

We consider the GeoNGNN with exactly 5 inner layers and 0 outer layers, and since we assume the injectiveness of intermediate functions, more layers will only lead to no-worse expressiveness.

Now consider node $i$'s subgraph. We first assume that node $i$ is distinct to the geometric center $c$. We denote node $j$'s representation in subgraph $i$ using $h_j$ instead of $h_{ij}$ for brevity when the context is clear. We can now prove the following:

1. After 1 round of DisGNN, for all $j \in [n]$, $h_j^{(1)} \to d_{ij}$ (since node $i$ is explictly marked in its subgraph).

2. After 2 rounds of DisGNN, according to Lemma B.4, for all $j \in [n]$, $h_j^{(2)} \to d_{jc}$.

3. After 3 rounds of DisGNN, for all $j \in [n]$, $h_j^{(3)} \to h_i^{(2)} \to d_{ic}$.

At this point, for all nodes $j$, $h_j^{(3)}$ can derive $d_{ji}, d_{jc}$, and $d_{ic}$, thus completing the *triangular distance encoding* necessary for reconstruction in Lemma B.6. Therefore, similar to the proof of Lemma B.6, with another two more rounds of DisGNN, the output can derive the point cloud up to permutation and Euclidean isometry.

4. At round 5, the output of DisGNN w.r.t. the subgraph of node $i$, $s_i = f_{\text{output}}(\{\!\!\{ h_j^{(5)} \mid j \in [n] \}\!\!\})$, can derive the point cloud up to permutation and Euclidean isometry.

There is still a potential problem with the above proof: node $i$ may coincide with the geometric center. In such case, we can not anymore obtain triangle distance information formed by each node $j$ and $i, c$, which is essential in Lemma B.6 for reconstruction. However, since $P$ has more than 2 nodes, there is at least one node that satisfies the assumption.

Notice that GeoNGNN injectively pools all the subgraph representations in the outer GNN and obtains the final output $s = f_{\text{outer}}(\{\!\!\{ s_i \mid i \in [n] \}\!\!\})$. Each subgraph represnetation $s_i$ can derive the distance between node $i$ and the geometric center $c$ as following: $s_i \to h_i^{(5)} \to h_i^{(4)} \to ... \to h_i^{(2)} \to d_{ic}$, according to Lemma B.4. This can be leveraged as an indicator, telling us whether node $i$ coincides with the $c$. As a consequence, we can derive the point cloud from $s$ as follows: first, search across all subgraph representations $s_i$ to find the one that does not coincide with the geometric center, then reconstruct the point cloud based on this subgraph representation according to Lemma B.6. This finishes the proof.

$\square$

### B.5    PROOF OF THEOREM 5.3

**Theorem 5.3** *(Completeness for general geometric subgraph GNNs) Under the conditions specified in Theorem 5.1, all general geometric subgraph GNNs in Definition 5.2 with at least one local aggregation are E(3)-complete.*

*Proof.* We establish by demonstrating that all general geometric subgraph GNNs $\mathsf{GeoA}(\mathcal{A}, \mathsf{Pool}, L, r_{\mathrm{sub}} = +\infty, r_{\mathrm{cutoff}} = +\infty)$ (Please see Appendix E for formal definitions), abbreviated as $\mathsf{GeoA}$ hereafter, are capable of *implementing* (Frasca et al., 2022) GeoNGNN (with $N_{\mathrm{inner}}$ inner layers, $r_{\mathrm{sub}} = +\infty, r_{\mathrm{cutoff}} = +\infty$ and without outer GNN) – denoted as complete GeoNGNN – with a fixed number of layers $L$. Since complete GeoNGNN is E(3)-complete when $N_{\mathrm{inner}} \geq 5$, $\mathsf{GeoA}$ can also achieve E(3)-completeness when $L$ is larger than some constant.

Two scenarios are considered: when $\mathsf{Geoagg}_u^L \in \mathcal{A}$ and when $\mathsf{Geoagg}_u^L \notin \mathcal{A}$.

In the scenario where $\mathsf{Geoagg}_u^L \in \mathcal{A}$, a single aggregation layer in $\mathsf{GeoA}$ can implement a complete GeoNGNN layer by learning an aggregation function $f^{(l)}$, defined in Section E, that only maintains $\mathsf{Geoagg}_{u,v}^P$ and $\mathsf{Geoagg}_u^L$ while disregarding other aggregation operations.

Notably, complete GeoNGNN utilizes VS pooling, while Pool in $\mathsf{GeoA}$ encompasses the options of VS or SV pooling. If Pool is VS, then consequently $\mathsf{GeoA}$ with $L = N_{\mathrm{inner}}$ has the capability to implement complete GeoNGNN.

On the other hand, should Pool be SV, $\mathsf{GeoA}$ can deploy an additional aggregation layer in conjunction with SV pooling to implement VS pooling. In the following, we mainly elaborate on this simulation process.

First, it is crucial to note that SV pooling learns the global representation $h_G$ through the function $f^{\mathrm{SV}}$ as follows:
$$h_G = f^{\mathrm{SV}}(\{\!\{\{\!\{h_{uv}^{(L_{\mathrm{SV}})} \mid u \in \mathcal{V}_G\}\!\} \mid v \in \mathcal{V}_G\}\!\}),$$
and VS pooling learns through $f^{\mathrm{VS}}$ in the subsequent manner:
$$h_G = f^{\mathrm{VS}}(\{\!\{\{\!\{h_{uv}^{(L_{\mathrm{VS}})} \mid v \in \mathcal{V}_G\}\!\} \mid u \in \mathcal{V}_G\}\!\}).$$
With $L_{\mathrm{SV}} = L_{\mathrm{VS}} + 1$, the last aggregation layer can be utilized to accumulate all subgraph information to the central node representation, i.e., $h^{(L_{\mathrm{SV}})}(u, u) = h^{(L_{\mathrm{VS}}+1)}(u, u) \to \{\!\{h_{uv}^{(L_{\mathrm{VS}})} \mid v \in \mathcal{V}_G\}\!\}$, given that $\mathsf{Geoagg}_u^L \in \mathcal{A}$. Notice that this is feasible since each subgraph is the original graph and is fully connected. Consequently:
$$\{\!\{\{\!\{h_{uv}^{(L_{\mathrm{SV}})} \mid u \in \mathcal{V}_G\}\!\} \mid v \in \mathcal{V}_G\}\!\}$$
$$\to \{\!\{h_{vv}^{(L_{\mathrm{SV}})} \mid v \in \mathcal{V}_G\}\!\}$$
$$\to \{\!\{\{\!\{h_{vu}^{(L_{\mathrm{VS}})} \mid u \in \mathcal{V}_G\}\!\} \mid v \in \mathcal{V}_G\}\!\}$$
$$= \{\!\{\{\!\{h_{uv}^{(L_{\mathrm{VS}})} \mid v \in \mathcal{V}_G\}\!\} \mid u \in \mathcal{V}_G\}\!\},$$
indicating that $f^{\mathrm{SV}}$ can acquire a function that initially converts $\{\!\{\{\!\{h_{uv}^{(L_{\mathrm{SV}})} \mid u \in \mathcal{V}_G\}\!\} \mid v \in \mathcal{V}_G\}\!\}$ to $\{\!\{\{\!\{h_{uv}^{(L_{\mathrm{VS}})} \mid v \in \mathcal{V}_G\}\!\} \mid u \in \mathcal{V}_G\}\!\}$, and subsequently emulates $f^{\mathrm{VS}}$. Thus, $\mathsf{GeoA}$ with $L = N_{\mathrm{inner}} + 1$ has the capability to implement complete GeoNGNN.

Finally, consider the case where $\mathsf{Geoagg}_u^L \notin \mathcal{A}$. And since we assume the existence of at least one local operation, it follows that $\mathsf{Geoagg}_v^L \in \mathcal{A}$. By presenting a proposition that essentially underscores symmetry, we can affirm that this case is equivalent to the first case, and therefore the conclusion still holds.

**Proposition B.8.** *Let* $\mathsf{GeoA}$ *be any general geometric subgraph GNN defined in Section E. Denote* $\mathcal{A}^{u \leftrightarrow v}$ *as the aggregation scheme obtained from* $\mathcal{A}$ *by exchanging the element* $\mathsf{agg}_{uu}^P$ *with* $\mathsf{agg}_{vv}^P$*, exchanging* $\mathsf{Geoagg}_u^L$ *with* $\mathsf{Geoagg}_v^L$*, and exchanging* $\mathsf{agg}_u^G$ *with* $\mathsf{agg}_v^G$*.    Then,* $\mathsf{GeoA}(\mathcal{A}, \mathsf{VS}, L, r_{sub}, r_{cutoff})$ *and* $\mathsf{GeoA}(\mathcal{A}^{u \leftrightarrow v}, \mathsf{SV}, L, r_{sub}, r_{cutoff})$ *can implement each other.*

*Proof.* Similar to the original proof of Proposition 4.5. in Zhang et al. (2023), the proof is almost trivial by symmetry: All functions within $\mathsf{GeoA}(\mathcal{A}, \mathsf{VS})$ can inherently learn to ensure that $h_{uv}^{(l)}$

in $\mathsf{GeoA}(\mathcal{A}, \mathsf{VS})$ precisely corresponds to $h_{vu}^{(l)}$ in $\mathsf{GeoA}(\mathcal{A}^{\mathsf{u}\leftrightarrow\mathsf{v}}, \mathsf{SV})$ for any arbitrary $u, v, l$, with the reverse equivalence also holding. Additionally, given the symmetry of the pooling method, consistency in the output can be guaranteed. □

As a direct consequence of Proposition B.8, all of the general subgraph GNN with $\mathsf{Geoagg}_{\mathsf{v}}^{\mathsf{L}}$ can implement another equivalent one with $\mathsf{Geoagg}_{\mathsf{u}}^{\mathsf{L}}$, and according to the previous proof, they are also E(3)-complete under the conditions specified in Theorem. This ends the proof.

□

## B.6 PROOF OF THEOREM 5.4

**Theorem 5.4** *(E(3)-Completeness of DimeNet, SphereNet, GemNet) When the following conditions are met, DimeNet, SphereNet and GemNet are E(3)-complete.*

- *The aggregation layer number is larger than some constant $C$ (irrelevant to the node number).*
- *They initialize and update all edge representations, i.e., $r_{embed} = +\infty$.*
- *They interact with all neighbors, i.e., $r_{int} = +\infty$..*

The main idea to prove this theorem is to also show that DimeNet, SphereNet and GemNet can implement (Frasca et al., 2022) GeoNGNN, which is E(3)-complete, and thus they are E(3)-complete as well. We elaborate on these models in separate subsections.

### B.6.1 E(3)-COMPLETENESS OF DIMENET

To begin, let us abstract the functions utilized in DimeNet. Fundamentally, DimeNet employs edge representations, iteratively updating these representations based on the neighbors of edges within a specified neighbor cutoff. During aggregation, DimeNet incorporates angle information formed by the center edge and its neighbor edges. Finally, all these edge representations are aggregated to produce a graph-level output.

We formally state these procedures:

**Initialization (DimeNet)** In DimeNet, the initial representation $h_{ij}$ of the edge $ij$ is initialized based on the tuple $(x_i, x_j, d_{ij})$, where $x_i$ and $x_j$ are the features of nodes $i$ and $j$, and $d_{ij}$ is the distance between these nodes:

$$h_{ij} = f_{\text{init}}^{\text{DimeNet}}(x_i, x_j, d_{ij}) \tag{9}$$

**Message Passing (DimeNet)** The message passing in DimeNet updates the edge representation $h_{ij}$ based on the features of neighboring edges and their respective geometric information. It aggregates information from all neighbors $k$ of node $i$:

$$h_{ij} = f_{\text{update}}^{\text{DimeNet}}\left(h_{ij}, \{\!\{(\theta_{kij}, h_{ki}, d_{ij}) \mid k \in \mathcal{N}(i)\}\!\}\right)$$

Note that $h_{ij}$ is initialized by fusing $d_{ij}$ and atomic information, therefore we have: $h_{ij} \to d_{ij}$. And clearly $d_{ij}, d_{ik}, d_{kj} \to \theta_{kij}$. Therefore, the above function is expressively equivalent to:

$$h_{ij} = f_{\text{update}}^{\text{DimeNet}}\left(h_{ij}, \{\!\{(d_{kj}, h_{ki}) \mid k \in \mathcal{N}(i)\}\!\}\right)$$

When the interaction cutoff of DimeNet is infinite, $\mathcal{N}(i) = [n]$, giving the final update function:

$$h_{ij} = f_{\text{update}}^{\text{DimeNet}}\left(h_{ij}, \{\!\{(h_{ki}, d_{kj}) \mid k \in [n]\}\!\}\right) \tag{10}$$

**Output Pooling (DimeNet)** Finally, DimeNet pools over all node pairs to obtain the final representation $t$:

$$t = f_{\text{output}}^{\text{DimeNet}}(\{\!\{\{\!\{h_{ij} \mid i \in [n]\}\!\} \mid j \in [n]\}\!\}) \tag{11}$$

GeoNGNN, as a subtype of subgraph GNN, initializes and updates the node $j$'s represnetation in node $i$'s subgraph based on its atomic number and its distance to node $i$. Representing the node $j$'s representation in node $i$'s subgraph as $h_{ij}$, then we can samely abstract its function forms.

**Initialization (GeoNGNN)** GeoNGNN initializes $h_{ij}$ based on node $j$'s initial node feature and distance encoding w.r.t. $i$:

$$h_{ij} = f_{\text{init}}^{\text{GeoNGNN}}(x_j, d_{ij}) \tag{12}$$

**Message Passing (GeoNGNN)** GeoNGNN's inner GNN iteratively updates $h_{ij}$ based on the following procedure:

$$h_{ij} = f_{\text{update}}^{\text{GeoNGNN}}(h_{ij}, \{\!\{(h_{ik}, d_{kj}) \mid k \in [n]\}\!\}) \tag{13}$$

Note that $k$ iterates all nodes, because according to Theorem 5.1, the complete version GeoNGNN has infinite subgraph size and message passing cutoff.

**Output Pooling (GeoNGNN)** In the absence of an outer GNN, GeoNGNN produces the scalar output by respectively aggregating all in-subgraph nodes' representations as the subgraph representation, and then aggregating all subgraphs' representations:

$$t = f_{\text{output}}^{\text{GeoNGNN}}(\{\!\{\{\!\{h_{ij} \mid j \in [n]\}\!\} \mid i \in [n]\}\!\}) \tag{14}$$

**Implementing GeoNGNN with DimeNet** Now we show how to use DimeNet to implement GeoNGNN. Since GeoNGNN is E(3)-complete, it can be observed that if DimeNet can successfully implement GeoNGNN, then DimeNet is also E(3)-complete.

Let us start by examining the initialization step. Note that $f_{\text{init}}^{\text{DimeNet}}$ (Eq. 9) takes more information than $f_{\text{init}}^{\text{GeoNGNN}}$ (Eq. 12) as input. Therefore, by learning a function $f_{\text{init}}^{\text{DimeNet}}$ that simply disregards the $x_i$ term, $f_{\text{init}}^{\text{DimeNet}}$ can achieve the same function form as $f_{\text{init}}^{\text{GeoNGNN}}$.

Next, we move on to the update (message passing) step. The only difference between $f_{\text{update}}^{\text{DimeNet}}$ (Eq. 10) and $f_{\text{update}}^{\text{GeoNGNN}}$ (Eq. 13) lies in their input arrangements. Specifically, $f_{\text{update}}^{\text{DimeNet}}$ takes $h_{ki}, k \in [n]$ as part of input, while $f_{\text{update}}^{\text{GeoNGNN}}$ takes $h_{ik}, k \in [n]$ as the corresponding part of input. This index swap can be mathematically aligned: We can stack two DimeNet update layers to implement one GeoNGNN update layer. The first DimeNet layer, starting from $h_{ij}^{(l)}$, calculates $h_{ij}^{(l+\frac{1}{2})}$ to store the information of $h_{ji}^{(l)}$, i.e., $h_{ij}^{(l+\frac{1}{2})} \rightarrow h_{ji}^{(l)}$. This is feasible due to the property that in Eq.10, $(d_{jj}, h_{ji})$ can be selected uniquely from the multiset by $f_{\text{update}}^{\text{DimeNet}}$, as $d_{kj} = 0$ if and only if $k = j$. Subsequently, the second DimeNet layer swaps the indices within the multiset, transforming $\left(h_{ij}^{(l+\frac{1}{2})}, \{\!\{(d_{kj}, h_{ki}^{(l+\frac{1}{2})}) \mid k \in [n]\}\!\}\right)$ into $\left(h_{ij}^{(l)}, \{\!\{(d_{kj}, h_{ik}^{(l)}) \mid k \in [n]\}\!\}\right)$. At this stage, the function form aligns with that of $f_{\text{update}}^{\text{GeoNGNN}}$ (Eq.13).

Finally, we consider the output step. Similarly, the only distinction between $f_{\text{output}}^{\text{DimeNet}}$ (Eq.11) and $f_{\text{update}}^{\text{GeoNGNN}}$ (Eq.14) lies in the swapping of input indices. By stacking one DimeNet update layer and one DimeNet output layer, we can similarly implement the GeoNGNN output layer, as discussed in the prior paragraph.

In conclusion, we have shown that DimeNet's layers can be utilized to implement GeoNGNN's layers. By doing so, we establish that DimeNet is also E(3)-complete with a constant number of layers.

### B.6.2 E(3)-COMPLETENESS OF SPHERENET

We aim to demonstrate the E(3)-completeness of SphereNet by illustrating that SphereNet can implement DimeNet, which has already been established as E(3)-complete in the preceding subsection. We begin by abstracting the layers of SphereNet.

**Initialization (SphereNet)** Similar to DimeNet, SphereNet also initializes edge representations by integrating atom properties and distance information:

$$h_{ij} = f_{\text{init}}^{\text{SphereNet}}(x_i, x_j, d_{ij}) \tag{15}$$

**Message Passing (SphereNet)** During message passing, SphereNet aggregates neighbor edge information into the center edge, while additionally considering end nodes' embeddings and spherical

coordinates. Since we consider SphereNet*, which drops the relative azimuthal angle $\varphi_k$ of end node $k$, the function form is highly similar to that of DimeNet:

$$h_{ij} = f_{\text{update}}^{\text{SphereNet}}(h_{ij}, v_i, v_j, \{\!\!\{(h_{ki}, d_{ki}, \theta_{kij}) \mid k \in [n]\}\!\!\})$$

Note that the original SphereNet can only be more powerful than SphereNet*.

Since $h_{ij} \to d_{ij}, (d_{ij}, d_{ki}, \theta_{kij}) \to d_{kj}$, the above function is equivalently expressive as

$$h_{ij} = f_{\text{update}}^{\text{SphereNet}}(h_{ij}, v_i, v_j, \{\!\!\{(h_{ki}, d_{kj}) \mid k \in [n]\}\!\!\}) \tag{16}$$

**Output (SphereNet)**   The output block of SphereNet first aggregates edge representations into node representations as follows:

$$v_j = f_{\text{node}}^{\text{SphereNet}}(\{\!\!\{h_{ij} \mid i \in [n]\}\!\!\})$$

Then, in order to calculate graph/global embedding, it further aggregates node representations as follows:

$$t = f_{\text{graph}}^{\text{SphereNet}}(\{\!\!\{v_j \mid j \in [n]\}\!\!\})$$

Therefore, the overall output function can be abstracted as:

$$t = f_{\text{output}}^{\text{SphereNet}}(\{\!\!\{\{\!\!\{h_{ij} \mid i \in [n]\}\!\!\} \mid j \in [n]\}\!\!\}) \tag{17}$$

**Implementing DimeNet with SphereNet**   Now we show how to use SphereNet to implement DimeNet.

At the initialization step, $f_{\text{init}}^{\text{SphereNet}}$ (Eq. 15) and $f_{\text{init}}^{\text{DimeNet}}$ (Eq. 9) exhibit exactly the same function form.

At the update step, $f_{\text{update}}^{\text{SphereNet}}$ (Eq. 16) takes strictly more information, i.e., the node representations, than $f_{\text{update}}^{\text{DimeNet}}$ (Eq. 10). Therefore, by learning a function $f_{\text{update}}^{\text{SphereNet}}$ that simply ignores node representations $v_i, v_j$, $f_{\text{update}}^{\text{SphereNet}}$ and $f_{\text{update}}^{\text{DimeNet}}$ can share the same function form.

At the output step, $f_{\text{output}}^{\text{SphereNet}}$ (Eq. 17) and $f_{\text{output}}^{\text{DimeNet}}$ (Eq. 11) share exactly the same function form.

In conclusion, we have shown that SphereNet's layers can be utilized to implement DimeNet's layers. By doing so, we establish that SphereNet is also E(3)-complete with a constant number of layers.

### B.6.3   E(3)-completeness of GemNet

GemNet(-Q) and DimeNet share the same initialization and output block. The difference between them lies in the update block. Specifically, GemNet adopts a three-hop message passing to incorporate higher-order geometric information, dihedral angle. The update function can be abstracted as:

$$h_{ij} = f_{\text{update}}^{\text{GemNet}}\big(h_{ij}, \{\!\!\{(h_{ab}, d_{ij}, d_{bj}, d_{ab}, \theta_{ijb}, \theta_{jba}, \theta_{ijba}) \mid b \in [n], a \in [n]\}\!\!\}\big)$$

Here, $\theta_{ijba}$ that has 4 subscripts represents the dihedral angle of plane $ijb$ and $jba$. By learning a function that simply selects all $b = i$ from the multiset and ignores some terms (Note that this is feasible, since $(h_{ab}, d_{ij}, \theta_{ijb}, \theta_{ijba}, d_{bj}, \theta_{jba}, d_{ab}) \to (d_{ij}, \theta_{ijb}, d_{bj}) \to d_{bi}$, while only $b = i$ results in $d_{bi} = 0$), $f_{\text{update}}^{\text{GemNet}}$ can be simplified as follows:

$$h_{ij} = f_{\text{update}}^{\text{GemNet}}(h_{ij}, \{\!\!\{(\theta_{aij}, h_{ai}, d_{ij}) \mid a \in [n]\}\!\!\}) \tag{18}$$

Now, the simplified $f_{\text{update}}^{\text{GemNet}}$ has the same form as $f_{\text{update}}^{\text{DimeNet}}$ (Eq. 10), indicating that GemNet can implement DimeNet, and therefore is also E(3)-complete.

## C   Extended Analysis of DisGNN

### C.1   The Proper Subset Relation in Figure 1(b)

**Proposition C.1.** *(Proper Subset)*

- $\mathcal{D}$-*symmetric point cloud set is a proper subset of* $\mathcal{C}$-*symmetric point cloud set.*

- *The unidentifiable point cloud set is a proper subset of* $\mathcal{D}$-*symmetric point cloud set.*

*Proof.* The first $\subseteq$ relation is a direct consequence of results in Delle Rose et al. (2023); Li et al. (2024), and reformulated in Proposition B.3. The second $\subseteq$ relation is a direct consequence of Theorem 4.2. Therefore, we only need to prove the $\subset$ relation for the two cases. To see this, we construct a point cloud $P$ for each case, showing that $P$ is in the second set but not in the first set.

The first constructed $P$ is shown in Figure 3. This point cloud $P$ is $\mathcal{C}$-symmetric but $\mathcal{D}$-asymmetric.

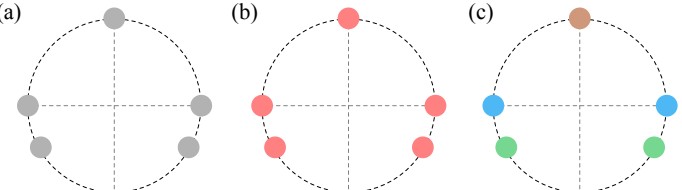

Figure 3: A point cloud $P$ that exhibits $\mathcal{C}$-symmetry but not $\mathcal{D}$-symmetry. (a) The point cloud $P$, which consists of an equilateral triangle and two additional nodes. (b) The labeled point cloud $(P, X^{\mathcal{C}})$ after calculating $\mathcal{C}$ features. Note that each node has the same distance to the geometric center, therefore the nodes still remain undivided. (c) The labeled point cloud $(P, X^{\mathcal{VD}})$ after calculating $\mathcal{D}$ features.

We describe the second case as follows. Consider an equilateral triangle $P$ of arbitrary side length. For any other point cloud $P'$ consisting of three points and non-isomorphic to $P$, there are only two possible scenarios: (1) $P'$ consists of three nodes forming an equilateral triangle with a side length distinct from that of $P$; (2) $P'$ consists of 3 nodes arranged in a manner that does not form an equilateral triangle. Importantly, DisGNN possesses the capability to distinguish between $P$ and $P'$ for both cases, due to its ability to embed node numbers and all distance lengths. Consequently, DisGNN successfully identifies $P$ (i.e., $P$ is not in the unidentifiable set). However, it's easy to check that $P$ is $\mathcal{D}$-symmetric.

$\square$

## C.2 COMPARISON TO HORDAN ET AL. (2024B)

We first show that our established symmetric point cloud set, specifically the $\mathcal{D}$-symmetric point cloud set, denoted as $\mathbb{R}_{\mathcal{D}}^{n\times3}$, is strictly *smaller* than the symmetric point cloud set $\mathbb{R}^{n\times3} \setminus \mathbb{R}_{\text{distinct}}^{n\times3}$ defined in Hordan et al. (2024b) (and restated in Definition C.3), as illustrated in Figure 4.

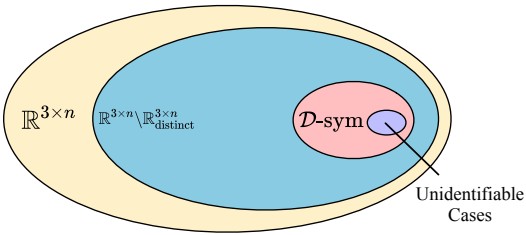

Figure 4: The relationship between our proposed symmetric point cloud sets and the symmetric set $\mathbb{R}^{n\times3} \setminus \mathbb{R}_{\text{distinct}}^{n\times3}$ characterized in prior work (Hordan et al., 2024b). The $\mathcal{D}$-symmetric point cloud set is a proper subset of $\mathbb{R}^{n\times3} \setminus \mathbb{R}_{\text{distinct}}^{n\times3}$, as stated in Proposition C.2.

More precisely, if we denote the complement of the $\mathcal{D}$-symmetric point cloud set as $\mathbb{R}_{\text{not }\mathcal{D}}^{n\times3}$, we arrive at the conclusion stated in Proposition C.2.

**Proposition C.2.** $\mathbb{R}_{distinct}^{n\times3} \subset \mathbb{R}_{not\ \mathcal{D}}^{n\times3}$.

*Proof.* To establish this result, we first revisit the findings of Hordan et al. (2024b), which define an asymmetric point cloud subset $\mathbb{R}^{n\times 3}_{\text{distinct}}$ as follows:

**Definition C.3** ($\mathbb{R}^{n\times 3}_{\text{distinct}}$ set (Hordan et al., 2024b))**.** We define $\mathbb{R}^{n\times 3}_{\text{distinct}} \subset \mathbb{R}^{n\times 3}$ as

$$\mathbb{R}^{n\times 3}_{\text{distinct}} := \{P \in \mathbb{R}^{n\times 3} \mid d(i,P) \neq d(j,P) \ \forall i,j \in [n], i \neq j\},$$

where the geometric degree of $i$ is defined as

$$d(i,P) = \{\|p_1 - p_i\|, \ldots, \|p_n - p_i\|\}.$$

It is evident that $\mathbb{R}^{n\times 3}_{\text{distinct}} \subseteq \mathbb{R}^{n\times 3}_{\text{not } \mathcal{D}}$, since DisGNN embeds nodes based on their geometric degrees. According to Definition C.3, all nodes' geometric degrees are distinct, leading to unique embedded node features. Thus, these point clouds are $\mathcal{D}$-asymmetric, meaning that $\mathbb{R}^{n\times 3}_{\text{distinct}} \subseteq \mathbb{R}^{n\times 3}_{\text{not } \mathcal{D}}$.

Furthermore, there exist many cases in $\mathbb{R}^{n\times 3}_{\text{not } \mathcal{D}}$ that do not belong to $\mathbb{R}^{n\times 3}_{\text{distinct}}$. For example, see the case in Figure 3. Other examples can be easily constructed, such as isosceles triangles where the leg length differs from the base length.

$\square$

The key reason why $\mathbb{R}^{n\times 3}_{\text{not } \mathcal{D}}$ contains significantly more point clouds than $\mathbb{R}^{n\times 3}_{\text{distinct}}$ is that the latter imposes overly strict constraints on asymmetry, requiring **all pairs** of nodes to be distinct. In contrast, $\mathbb{R}^{n\times 3}_{\text{not } \mathcal{D}}$ emphasizes the **global** asymmetry of the entire point cloud.

In Theorem 2 of Hordan et al. (2024b), the authors showed that given any two point clouds $P_1, P_2 \in \mathbb{R}^{n\times 3}_{\text{distinct}}$, DisGNN can distinguish them. Our result is strictly stronger than that of Hordan et al. (2024b) in the sense that our Theorem 4.2 identifies many more distinguishable pairs of point clouds (by DisGNN) than their result, in the following sense:

- Theorem 4.2 shows that arbitrary pairs of point clouds $P_1, P_2$ from $\mathbb{R}^{n\times 3}_{\text{not } \mathcal{D}}$ (which is strictly larger than $\mathbb{R}^{n\times 3}_{\text{distinct}}$ as shown in Proposition C.2) can be distinguished by DisGNN.

- Theorem 4.2 further establishes **identifiability**, meaning that for any $P_1 \in \mathbb{R}^{n\times 3}_{\text{not } \mathcal{D}}$ and any $P_2$ from the **entire** point cloud set $\mathbb{R}^{n\times 3}$, DisGNN can still distinguish them.

### C.3 ASSESSMENT OF UNIDENTIFIABLE CASES OF DISGNN

Here, we provide further details about the experiment in Section 6.1.

We first give a simple proposition, which can determine whether a given point cloud is $\mathcal{A}$-symmetric without the need to consider "mass" functions $m$, and subsequentially facilitate the noise tolerance setting.

**Proposition C.4.** *(Equivalent definition for $\mathcal{A}$-symmetry) Given an arbitrary point cloud $P \in \mathbb{R}^{n\times 3}$ and a $E(3)$-invaraint and permutation-equivariant algorithm $\mathcal{A}$, let $K$ denote the number of distinct node features in $\mathcal{A}(P)$, and we consider $K$ sub-point clouds each only contain nodes from $P$ with the same node feature. Then we have: $P$ is $\mathcal{A}$-symmetric $\iff$ all these $K$ sub-point clouds' geometric centers coincide.*

*Proof.* We first prove that $P$ is $\mathcal{A}$-symmetric $\implies$ all these $K$ sub-point clouds' geometric centers coincide. This is actually a direct consequence of the original definition of $\mathcal{A}$-symmetry: each geometric center of the $K$ point clouds is in $\mathcal{A}^{\text{set}}(P)$, and $P$ is $\mathcal{A}$-symmetric means that $\mathcal{A}^{\text{set}}(P)$ contains only one element, therefore all $K$ sub-point clouds' geometric centers coincide.

We then prove the reverse direction. We denote $I_k$ as the index set of the nodes with the $k$-th kind of node features in $\mathcal{A}(P)$. For an arbitrary element $\mathcal{A}(P)^m$ from collections $\mathcal{A}^{\text{set}}(P)$ calculated by

function $m$, we have $\mathcal{A}(P)^m = \frac{\sum_{i \in [n]} m(x_i^{\mathcal{A}}) p_i}{\sum_{i \in [n]} m(x_i^{\mathcal{A}})}$ by definition, which can be decomposed as follows:

$$
\begin{aligned}
\mathcal{A}(P)^m &= \frac{\sum_{i \in [n]} m(x_i^{\mathcal{A}}) p_i}{\sum_{i \in [n]} m(x_i^{\mathcal{A}})} \\
&= \frac{\sum_{k \in [K]} \sum_{i \in I_k} m(x_i^{\mathcal{A}}) p_i}{\sum_{k \in [K]} \sum_{i \in I_k} m(x_i^{\mathcal{A}})} \\
&= \frac{\sum_{k \in [K]} M_k c_k}{\sum_{k \in [K]} M_k},
\end{aligned}
\tag{19}
$$

where $c_k \in \mathbb{R}^3$ denotes the geometric center of the $k$-th point cloud, $M_k = \sum_{i \in I_k} m(x_i^{\mathcal{A}})$ denotes the sum of "masses" associated with the $k$-th sub-point cloud. Since all $c_k$ coincide, i.e., $c_1 = c_2 = \ldots = c_K$, we have: $\mathcal{A}(P)^m = \frac{c_1 \sum_{k \in [K]} M_k}{\sum_{k \in [K]} M_k} = c_1$. Thus, all the possible elements from $\mathcal{A}^{\text{set}}(P)$ coincide with $c_1$. Obviously, since the geometric center $c$ of the whole point cloud is also in $\mathcal{A}^{\text{set}}(P)$, $c_1 = c$. Therefore, $P$ is $\mathcal{A}$-symmetric. $\qquad\square$

**Rescaling and criteria settings**   Since the two datasets under consideration exhibit different scales, the use of fixed tolerance errors is inapplicable. To address this, a preprocessing step is performed on both datasets by rescaling all point clouds to ensure that the distance between the geometric center and the farthest node is 1. For ModelNet40, we preprocess the data using farthest point sampling with 256 nodes. We then fix the rounding number $r$ to 2 for QM9 and 1 for ModelNet40 (which is thus quite robust against noise in this scale), and conduct the assessment with the deviation error ranging from 1e-6 to 1e-1, as shown in Figure 2. Note that a deviation error of 1e-1 is quite large for a point cloud located within a unit sphere. As a result, many asymmetric clouds may still be determined as symmetric under such criteria. We use this as a rough upper bound only for reference.

## D   EXTENDED ANALYSIS OF GEONGNN

### D.1   COMPLEXITY ANALYSIS

For a $n$-sized point cloud, without considering the complexity of achieving injective intermediate functions, GeoNGNN achieves theoretical completeness with an asymptotic time complexity of $O(n^3)$. This complexity arises from the fact that there are $n$ subgraphs, each of which undergoes the complete-version DisGNN operation, resulting in an overall complexity of $O(n^2)$ per subgraph. Importantly, this time complexity is consistent with that of 2-F-DisGNN (Li et al., 2024), DimeNet (Gasteiger et al., 2019), and SphereNet (Liu et al., 2021), when their conditions for achieving E(3)-completeness are met.

We leverage the findings introduced by Amir et al. (2024) to analyze the complexity involving the realization of *injective* neural functions.[6] Within each aggregation layer, GeoNGNN embeds the multiset $\{\!\{(h_{ij}^{(l)}, d_{ij}) \mid j \in [n]\}\!\}$ to update the representation of $h_{ij}^{(l)}$. As per the study conducted by Amir et al. (2024), an embedding dimension of $O(kn)$ is sufficient for injectively embedding such a multiset, with $k$ representing the embedding dimension of $(h_{ij}^{(l)}, d_{ij})$. In the initial layer, $(h_{ij}^{(0)}, d_{ij})$ possesses a constant dimension independent of $n$. Therefore, the sufficient embedding dimension for $h_{ij}^{(1)}$ is $O(n)$. Though it seems that the sufficient embedding dimension will grow exponentially with respect to the layer number $l$, Hordan et al. (2024b) has demonstrated that the crucial dimension is the *intrinsic dimension* of the multiset, which maintains $O(n)$ throughout all layers, rather than $kn$, the ambient dimension. Consequently, the sufficient embedding dimension for any given layer is $O(n)$. By following the neural function form proposed by Amir et al. (2024), we apply a shallow MLP individually to each element within the multiset and sum them up to obtain the multiset embedding. This leads to a complexity of $O(n^2) \times n = O(n^3)$, where $O(n^2)$ represents the complexity of forwardness of MLP whose input and output dimension are both $O(n)$, and $n$ represents the $n$

---

[6]We note that for tractability, "E(3)-completeness" now should refer to the model's ability to distinguish between any pairs of non-isomorphic point clouds of size *less than or equal to* $n$.

elements in the multiset. Considering the updating of all $h_{ij}^{(l)}, i, j \in [n]$, the complexity in each layer becomes $O(n^3) \times n^2 = O(n^5)$. During the final pooling stage, the nodes are initially pooled into subgraph representations, resulting in $n \times O(n^2) \times n = O(n^4)$, where the first $n$ denotes $n$ subgraphs, and the last $n$ denotes $n$ nodes within each subgraph. Subsequently, all subgraph representations are further pooled into a graph-level representation, culminating in a complexity of $O(n^2) \times n = O(n^3)$, where $n$ represents $n$ nodes. Consequently, the overall complexity amounts to $O(n^5)$.

### D.2 THEORETICAL EXPRESSIVENESS WITH FINITE SUBGRAPH RADIUS AND DISTANCE CUTOFF

While Theorem 5.1 establishes that infinite subgraph radius and distance cutoff guarantees completeness over all point clouds, this section explores how finite values can also enhance geometric expressiveness compared to DisGNN. We demonstrate this through an example.

Take the left pair of point clouds in Figure 7 for example, and note that this pair of point clouds cannot be distinguished by DisGNN even when taking fully-connected point cloud as input. Assuming that the subgraph radius and distance cutoff for each node are finite, only covering nodes' one-hop neighbors, as illustrated in Figure 5. The representation of the green node produced by inner DisGNN will differ between the two point clouds, due to the presence of long-distance information on the left point cloud, whereas it is absent on the right point cloud. Therefore, GeoNGNN with such a small subgraph radius and distance cutoff can easily distinguish the two point clouds.

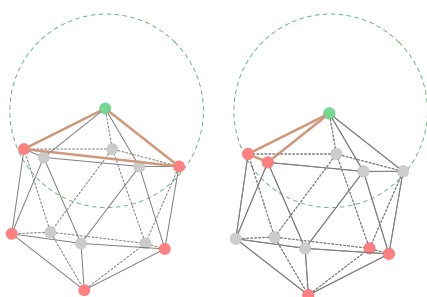

Figure 5: An example that illustrates the separation power of finite-subgraph-size GeoNGNN. The green node represents the central node, while the green *sphere* depicts the subgraph environment surrounding the central node. The brown line signifies the distance information that will be aggregated during the message passing.

## E GEOMETRIC SUBGRAPH GRAPH NEURAL NETWORKS

In this section, we provide self-contained definitions of general geometric subgraph GNNs as delineated in Definition 5.2 as well as definitions of the geometric counterparts of well-established traditional subgraph GNNs, all of which are proven to be E(3)-complete under specific conditions.

### E.1 BASIC DEFINITIONS

We first notice that the main definition of general geometric subgraph GNN is based on the general definitions of subgraph GNNs in traditional graph settings from Zhang et al. (2023), where the input graphs are unweighted graphs. The difference is that the geometric subgraph GNN is applied to point clouds, treating it as distance graphs with distance cutoff $r_{\text{cutoff}}$, and applies geometric aggregations.

We separately introduce the main components in Definition 5.2. From now on, we follow the notation style of (Zhang et al., 2023), but for geometric settings.

We denote the original distance graph as $G = (\mathcal{V}_G, \mathcal{E}_G)$, where $\mathcal{V}_G = [n]$ contains all nodes in the distance graph, and $\mathcal{E}_G = \{(u, v) \mid u, v \in \mathcal{V}_G, d_{uv} \leq r_{\text{cutoff}}\}$ consists of all weighted edges with weight (distance) less than $r_{\text{cutoff}}$, by definition of the distance graph. We denote the subgraph of node $u$ as $G^u = (\mathcal{V}_{G^u}, \mathcal{E}_{G^u})$. We use $\mathcal{N}_{G^v}(u)$ to denote the set of neighbors of $u$ in $v$'s subgraph,

i.e., $\mathcal{N}_{G^v}(u) = \{i \mid i \in \mathcal{V}_{G^v}, (i, u) \in \mathcal{E}_{G^v}\}$. We use $h_{uv}^{(l)}$ to denote the node embedding for node $v$ in node $u$'s subgraph at the $l$-th layer of the subgraph GNN.

**Subgraph generation**   As defined in Definition 5.2, general geometric subgraph GNNs adopt node marking with $r_{\text{sub}}$-size ego subgraph as the subgraph generation policy. This essentially means that node $u$'s subgraph, $G^u = (\mathcal{V}_{G^u}, \mathcal{E}_{G^u})$, contains all the nodes and edges within Euclidean distance $r_{\text{sub}}$, i.e., $\mathcal{V}_{G^u} = \{i \mid i \in \mathcal{V}_G, d_{iu} \leq r_{\text{sub}}\}$, $\mathcal{E}_{G^u} = \{(i, j) \mid (i, j) \in \mathcal{E}_G, i, j \in V_{G^u}\}$. And node $v$ in $u$'s subgraph's initial representation is $h_{uv}^{(0)} = f^{\text{init}}(\mathbb{1}_{u=v}, x_u, x_v)$, where $\mathbb{1}_{u=v}$ represents indicator function that equals 1 when $u = v$ and 0 otherwise, and $x_u, x_v$ are the potential node features.

**Geometric aggregation schemes**   As defined in Definition 5.2, geometric subgraph GNN adopts *general geometric subgraph GNN layers* as the basic aggregation layer, which is formally defined as follows:

**Definition E.1.** A general geometric subgraph GNN layer has the form

$$h_{uv}^{(l+1)} = f^{(l+1)}(h_{uv}^{(l)}, \mathsf{op}_1(u, v, h^{(l)}, G), \cdots, \mathsf{op}_r(u, v, h^{(l)}, G)),$$

where $f^{(l+1)}$ is an arbitrary parameterized continuous function, and each atomic operation $\mathsf{op}_i(u, v, h, G)$ can take any of the following expressions:

- Single-point: $h_{vu}, h_{uu}$, or $h_{vv}$;
- Global: $\{\!\{h_{uw} \mid w \in \mathcal{V}_G\}\!\}$ or $\{\!\{h_{wv} \mid w \in \mathcal{V}_G\}\!\}$;
- Local: $\{\!\{(h_{uw}, d_{vw}) \mid w \in \mathcal{N}_{G^u}(v)\}\!\}$ or $\{\!\{(h_{wv}, d_{uw}) \mid w \in \mathcal{N}_{G^v}(u)\}\!\}$

Notice that $h_{uv}^{(l)}$ is always in the input to ensure that geometric subgraph GNN can always *refine* the node feature partition.   These operations are denoted as $\mathsf{agg}_{uu}^{\mathsf{P}}, \mathsf{agg}_{vv}^{\mathsf{P}}, \mathsf{agg}_{vu}^{\mathsf{P}}, \mathsf{agg}_{u}^{\mathsf{G}}, \mathsf{agg}_{v}^{\mathsf{G}}, \mathsf{Geoagg}_{u}^{\mathsf{L}}, \mathsf{Geoagg}_{v}^{\mathsf{L}}$, respectively.   Here, we color the local operations with blue and name it with an additional "Geo" prefix since all the aggregation schemes do not incorporate geometric information and are the same as those in Zhang et al. (2023), except for the local operations, which additionally incorporates distance information.

**Pooling layer**   After $L$ geometric aggregation layers, geometric subgraph GNN outputs a graph-level representation $f(G)$ through a two-level pooling of the collected features $\{\!\{h_{uv}^{(L)} : u, v \in \mathcal{V}_G\}\!\}$ the same as Zhang et al. (2023) do. Specifically, there are two approaches, named *VS* (vertex-subgraph pooling) and *SV* (subgraph-vertex pooling). The first approach first pools all node features in each subgraph $G^u$ to obtain the subgraph representation, i.e., $h_u = f^{\mathsf{S}}\left(\{\!\{h_{uv}^{(L)} \mid v \in \mathcal{V}_G\}\!\}\right)$, and then pools all subgraph representations to obtain the final output $h_G = f^{\mathsf{G}}(\{\!\{h_u \mid u \in \mathcal{V}_G\}\!\})$. Here, $f^{\mathsf{S}}$ and $f^{\mathsf{G}}$ can be any parameterized function. The second approach first generates node representations $h_v = f^{\mathsf{V}}\left(\{\!\{h_{uv}^{(L)} \mid u \in \mathcal{V}_G\}\!\}\right)$, and then pools all these node representations to obtain the graph representation, i.e., $h_G = f^{\mathsf{G}}(\{\!\{h_v \mid v \in \mathcal{V}_G\}\!\})$.

We denote a general geometric subgraph GNN with $L$ layers, each with aggregation scheme $\mathcal{A} = \mathcal{B} \cup \mathsf{agg}_{uv}^{\mathsf{P}}$, and with parameters $r_{\text{sub}}, r_{\text{cutoff}}$ as $\mathsf{GeoA}(\mathcal{A}, \mathsf{Pool}, L, r_{\text{sub}}, r_{\text{cutoff}})$, where $\mathsf{Pool} \in \{\mathsf{VS}, \mathsf{SV}\}$, and

$$\mathcal{B} \subset \{\mathsf{agg}_{uu}^{\mathsf{P}}, \mathsf{agg}_{vv}^{\mathsf{P}}, \mathsf{agg}_{vu}^{\mathsf{P}}, \mathsf{agg}_{u}^{\mathsf{G}}, \mathsf{agg}_{v}^{\mathsf{G}}, \mathsf{Geoagg}_{u}^{\mathsf{L}}, \mathsf{Geoagg}_{v}^{\mathsf{L}}\}.$$

It is obvious that GeoNGNN without outer GNN, is a specific kind of general subgraph GNN, which can be denoted as $\mathsf{GeoA}(\{\mathsf{agg}_{uv}^{\mathsf{P}}, \mathsf{Geoagg}_{u}^{\mathsf{L}}\}, VS, N_{\text{inner}}, r_{\text{sub}}, r_{\text{cutoff}})$.

### E.2   GEOMETRIC COUNTERPARTS OF WELL-KNOWN TRADITIONAL SUBGRAPH GNNS

In the main body, we have extended one of the simplest subgraph GNNs, NGNN (Zhang & Li, 2021), to geometric scenarios, and shown that GeoNGNN is already E(3)-complete. Similar efforts can be made to other well known subgraph GNNs, including DS-GNN (Bevilacqua et al., 2021), DSS-GNN (Bevilacqua et al., 2021), GNN-AK (Zhao et al., 2021), OSAN (Qian et al., 2022) and so on. We now give a general definition of the geometric counterparts of these subgraph GNNs, and

show that they, even though does not exactly match the general forms in Definition 5.2, are also E(3)-complete under exactly the same conditions as Theorem 5.3.

Similar to Zhang et al. (2023), we consider the following well-known subgraph GNNs: IDGNN (You et al., 2021), DS-GNN (Bevilacqua et al., 2021), OSAN (Qian et al., 2022), GNN-AK (Zhao et al., 2021), DSS-GNN (ESAN) (Bevilacqua et al., 2021), GNN-AK-ctx (Zhao et al., 2021), SUN (Frasca et al., 2022). Generally, all these traditional subgraph GNNs involve local aggregations. In a manner akin to the modification detailed in Definition 5.2, to get the geometric counterpart of these models, we replace the node representations aggregated by local operations with *node representations that integrate the distances between the aggregated node and the base node* (i.e., the node performing the aggregation).

Notably, GNN-AK, GNN-AK-ctx, IDGNN, OSAN, DS-GNN, and DSS-GNN all employ base GNNs, which can operate on graphs, subgraphs, or a combination of subgraphs (as in DSS-GNN). In these cases, the proposed adjustment simply involves replacing these base GNNs with DisGNN. In certain architectures like SUN, they may have more complicated operations, such as aggregating $\{\!\{h_{ww'} \mid w' \in \mathcal{N}(v), w \in \mathcal{V}_G\}\!\}$ to update $h_{uv}$, which involve a combination of global ($w$) and local ($w'$) aggregation. Consistent with the modification rule, we selectively incorporate distance information solely for these local operations. Therefore, the featured geometric counterpart aggregates $\{\!\{(h_{w,w'}, d_{w'v}) \mid w' \in \mathcal{N}(v), w \in \mathcal{V}_G\}\!\}$ instead. One can also incorporate distance information into those global operations. However, they are unnecessary in terms of boosting expressiveness, as current modifications are enough for them to achieve completeness, as shown in the following. We name all the geometric counterparts with a *Geo* prefix, such as GeoSun.

**Theorem E.2.** *(Completeness for geometric counterparts of well-known subgraph GNNs) Under the node marking with $r_{sub}$-size ego subgraph policy, when the conditions described in Theorem 5.3 are met, GeoIDGNN, GeoDS-GNN, GeoOSAN, GeoGNN-AK, GeoDSS-GNN (ESAN), GeoGNN-AK-ctx, GeoSUN are all E(3)-complete.*

*Proof.*
- Among these models, GeoIDGNN, GeoDS-GNN, GeoOSAN fall under the general definition of geometric subgraph GNNs (see Definition 5.2), and they all incorporate at least one local operation, therefore they are inherently complete given these conditions according to Theorem 5.3 (See proof in Appendix B.5).

- Under node marking with original graph ($r_{sub} = \infty$) policy, the other models can all be well abstracted and summarized, which have been done by Zhang et al. (2023). We list here for convenience (non-blue parts represent the geometric modification):

GeoGNN-AK (Zhao et al., 2021).

$$h_{uv}^{(l+1)} = \begin{cases} f^{(l)}(h_{uv}^{(l)}, \\ \qquad h_{vv}^{(l)}, \qquad\qquad\qquad\qquad \text{if } u \neq v, \\ \qquad \{\!\{(h_{uw}^{(l)}, d_{vw}) : w \in \mathcal{N}_G(v)\}\!\}) \\ f^{(l)}(h_{vv}^{(l)}, \\ \qquad \{\!\{(h_{uw}^{(l)}, d_{vw}) : w \in \mathcal{N}_G(v)\}\!\}, \quad \text{if } u = v. \\ \qquad \{\!\{h_{uw}^{(l)} : w \in \mathcal{V}_G\}\!\}) \end{cases}$$

It adopts VS pooling.

GeoGNN-AK-ctx (Zhao et al., 2021). The GNN aggregation scheme can be written as

$$h_{uv}^{(l+1)} = \begin{cases} f^{(l)}(h_{uv}^{(l)}, \\ \qquad h_{vv}^{(l)}, \qquad\qquad\qquad\qquad \text{if } u \neq v, \\ \qquad \{\!\{(h_{uw}^{(l)}, d_{vw}) : w \in \mathcal{N}_G(v)\}\!\}) \\ f^{(l)}(h_{vv}^{(l)}, \\ \qquad \{\!\{(h_{uw}^{(l)}, d_{vw}) : w \in \mathcal{N}_G(v)\}\!\}, \\ \qquad \{\!\{h_{uw}^{(l)} : w \in \mathcal{V}_G\}\!\}, \qquad \text{if } u = v. \\ \qquad \{\!\{h_{wv}^{(l)} : w \in \mathcal{V}_G\}\!\}) \end{cases}$$

It adopts VS pooling.

GeoDSS-GNN (Bevilacqua et al., 2021). The aggregation scheme of DSS-GNN can be written as

$$
\begin{aligned}
h_{uv}^{(l+1)} = f^{(l)}(h_{uv}^{(l)}, \\
&\{\!\{(h_{uw}^{(l)}, d_{vw}) : w \in \mathcal{N}_G(v)\}\!\}, \\
&\{\!\{h_{wv}^{(l)} : w \in \mathcal{V}_G\}\!\}, \\
&\{\!\{(h_{w,w'}^{(l)}, d_{w'v}) : w \in \mathcal{V}_G, w' \in \mathcal{N}_G(v)\}\!\}).
\end{aligned}
$$

It adopts VS pooling.

GeoSUN (Frasca et al., 2022). The WL aggregation scheme can be written as

$$
\begin{aligned}
h_{uv}^{(l+1)} = f^{(l)}(h_{uv}^{(l)}, h_{uu}^{(l)}, h_{vv}^{(l)}, \\
&\{\!\{(h_{uw}^{(l)}, d_{vw}) : w \in \mathcal{N}_G(v)\}\!\}, \\
&\{\!\{h_{uw}^{(l)} : w \in \mathcal{V}_G\}\!\}, \\
&\{\!\{h_{wv}^{(l)} : w \in \mathcal{V}_G\}\!\}, \\
&\{\!\{(h_{w,w'}^{(l)}, d_{w'v}) : w \in \mathcal{V}_G, w' \in \mathcal{N}_G(v)\}\!\}).
\end{aligned}
$$

It can adopt VS or SV pooling.

Regarding these models, it is observed that they all exhibit the local operation $\mathsf{Geoagg}_u^L$, thus enabling them to implement the GeoNGNN inner layer with a single aggregation layer, as demonstrated in the proof of Theorem 5.3. Moreover, irrespective of whether they employ SV or VS pooling, they are all capable of implementing VS pooling, which is adopted by GeoNGNN, as evidenced in the proof of Theorem 5.3. Hence, under the given conditions, these models can also implement GeoNGNN under the given conditions and consequently achieve completeness, as GeoNGNN itself is complete.

$\square$

## F  EXTENDED EVALUATION OF GEONGNN

In this section, we further evaluate GeoNGNN on various datasets, including rMD17 (Chmiela et al., 2017), which assesses the model's ability to predict high-precision energy and forces based on molecular conformations; MD22 (Chmiela et al., 2023), which requires models to efficiently handle large point clouds; 3BPA (Kovács et al., 2021), which assesses the model's ability to generalize well on out-of-domain datasets; and QM9 (Ramakrishnan et al., 2014; Wu et al., 2018), which contains diverse properties for evaluating the models' universal learning capabilities. Please refer to Appendix G for further model architecture and experimental settings.

Since GeoNGNN is built upon DisGNN, a direct comparison is made to DisGNN, which we implement as an enhanced SchNet (Schütt et al., 2018) with additional residual layers. Additionally, we include two other complete models: DimeNet (Gasteiger et al., 2019) and GemNet (Gasteiger et al., 2021), as well as the recent advanced invariant model 2-F-DisGNN (Li et al., 2024). Furthermore, we incorporate equivariant models that leverage vectors or higher-order tensors, including PaiNN (Schütt et al., 2021), NequIP (Batzner et al., 2022), and MACE (Batatia et al., 2022).

Overall, GeoNGNN demonstrates good experimental results, and in some cases, performs comparably to advanced models that incorporate equivariant high-order tensors, such as Allegro (Musaelian et al., 2023), MACE (Batatia et al., 2022), and PaiNN (Schütt et al., 2021), on specific tasks. This validates the effectiveness of capturing *subgraph representations* in molecular-relevant tasks, which may provide even better inductive bias or generalization ability in this scenario than the design like directional message passing in DimeNet (Gasteiger et al., 2019), which is also complete in our proof yet did not catch up with GeoNGNN's downstream performance.

However, it does not consistently achieve the best performance, which may be attributed to its simple design, relying solely on distance features and the weak base GNN it employs. Therefore, future

Table 2: MAE loss on revised MD17. Energy (E) is in kcal/mol, and force (F) is in kcal/mol/Å. The best and second-best results are shown in **bold** and underline. Results for GeoNGNN are highlighted in green if they outperform their base model, DisGNN, while models are grayed if they are our characterized powerful models. The average rank is computed as the mean rank across all rows.

| Molecule | Target | Equivariant models | | | Invariant models | | | | |
|---|---|---|---|---|---|---|---|---|---|
| | | PaiNN | NequIP | MACE | DisGNN | 2F-Dis. | DimeNet | GemNet | GeoNGNN |
| Aspirin | E | 0.1591 | 0.0530 | 0.0507 | 0.1565 | **0.0465** | 0.1321 | - | 0.0502 |
| | F | 0.3713 | 0.1891 | 0.1522 | 0.4855 | **0.1515** | 0.3549 | 0.2191 | 0.1720 |
| Azobenzene | E | - | **0.0161** | 0.0277 | 0.2312 | 0.0315 | 0.1063 | - | 0.0315 |
| | F | - | **0.0669** | 0.0692 | 0.5050 | 0.1121 | 0.2174 | - | 0.1157 |
| Benzene | E | - | **0.0009** | 0.0092 | 0.0308 | 0.0013 | 0.0061 | - | 0.0014 |
| | F | - | 0.0069 | 0.0069 | 0.2209 | 0.0085 | 0.0170 | 0.0115 | **0.0065** |
| Ethanol | E | 0.0623 | 0.0092 | 0.0092 | 0.0117 | **0.0065** | 0.0345 | - | 0.0074 |
| | F | 0.2306 | 0.0646 | 0.0484 | 0.0774 | **0.0379** | 0.1859 | 0.0830 | 0.0482 |
| Malonaldehyde | E | 0.0899 | 0.0184 | 0.0184 | 0.2814 | **0.0129** | 0.0507 | - | 0.0143 |
| | F | 0.3182 | 0.1176 | 0.0945 | 0.1661 | **0.0782** | 0.2743 | 0.1522 | 0.0875 |
| Naphthalene | E | 0.1176 | 0.0208 | 0.0115 | 0.1269 | 0.0103 | 0.0445 | - | **0.0069** |
| | F | 0.0830 | **0.0300** | 0.0369 | 0.4144 | 0.0478 | 0.1105 | 0.0438 | 0.0377 |
| Paracetamol | E | - | 0.0323 | **0.0300** | 0.1534 | 0.0310 | 0.1176 | - | 0.0352 |
| | F | - | 0.1361 | **0.1107** | 0.4698 | 0.1178 | 0.3028 | - | 0.1385 |
| Salicylic acid | E | 0.1130 | **0.0161** | 0.0208 | 0.0791 | 0.0174 | 0.0590 | - | 0.0168 |
| | F | 0.2099 | 0.0922 | **0.0715** | 0.3481 | 0.0860 | 0.2428 | 0.1222 | 0.0881 |
| Toluene | E | 0.0969 | 0.0069 | 0.0115 | 0.0918 | **0.0051** | 0.0228 | - | 0.0069 |
| | F | 0.1015 | 0.0369 | 0.0346 | 0.3070 | **0.0284** | 0.1085 | 0.0507 | 0.0375 |
| Uracil | E | 0.1038 | **0.0092** | 0.0115 | 0.0363 | 0.0139 | 0.0338 | - | 0.0096 |
| | F | 0.1407 | 0.0715 | **0.0484** | 0.1973 | 0.0828 | 0.1634 | 0.0876 | 0.0577 |
| AVG RANK | | 6.71 | 2.80 | 2.50 | 6.55 | 2.25 | 5.70 | 5.00 | 2.55 |

improvements could focus on integrating vectors (Schütt et al., 2021; Thölke & De Fabritiis, 2021) or higher-order tensors (Thomas et al., 2018) into GeoNGNN's base model, as these have been shown to enhance model generalization (Du et al., 2024; Musaelian et al., 2023).

Nevertheless, GeoNGNN shows promise, leaving open opportunities for future research by incorporating more powerful base architectures within its nested framework to improve empirical performance, or developing theoretically more powerful models analogous to its underlying principles.

## F.1 MOLECULE STRUCTURE LEARNING: REVISED MD17

We first evaluate on the rMD17 dataset (Chmiela et al., 2017), which poses a substantial challenge to models' geometric learning ability. This dataset encompasses trajectories from molecular dynamics simulations of several small molecules, and the objective is to predict the energy and atomic forces of a given molecule conformation, which contains all atoms' positions and atomic numbers.

The comprehensive results and comparison is shown in Table 2. There are several important observations: 1) GeoNGNN exhibits substantial improvements over DisGNN, demonstrating the effectiveness of higher expressiveness and the significance of capturing subgraph representations on practical tasks. 2) Though equally being E(3)-complete, with practical implements, DimeNet and GemNet show worse performance than GeoNGNN. This finding suggests that different inductive biases of different model designs can impact significantly on *generalization ability*, while learning more refined *subgraph representations* like GeoNGNN may be superior to directional message passing in certain molecule-relevant tasks. 3) GeoNGNN shows competitive performance in comparison to other well-designed advanced models. However, it does not consistently achieve the best performance.

## F.2 SCALING TO LARGE GRAPHS: MD22

Geometric models face challenges in scaling to larger point clouds. Here, we evaluate GeoNGNN on MD22 (Chmiela et al., 2023), a dataset of molecules with up to 370 atoms. To enhance efficiency, we apply GeoNGNN with a *finite* subgraph cutoff $r_{sub}$ (5 Å) and message passing cutoff $r_{cutoff}$ (5 Å), and assess its practical performance.

In this table, we further include a recent SOTA model VisNet-LSRM (Li et al., 2023). As shown in Table 3, GeoNGNN consistently outperforms DisGNN across all targets. It also delivers *competitive*

Table 3: MAE loss on MD22. Energy (E) in meV/atom, force (F) in meV/Å. The best and the second best results are shown in **bold** and underline. We color the cell if GeoNGNN outperforms DisGNN. The average rank is the average of the rank of each row.

| Mol | # atoms | Target | sGDML | PaiNN | TorchMD-NET | Allegro | Equiformer | MACE | VisNet-LSRM | DisGNN | GeoNGNN |
|---|---|---|---|---|---|---|---|---|---|---|---|
| | | | | | | *Equivariant models* | | | | *Invariant models* | |
| Ac-Ala3-NHMe | 42 | E | 0.4 | 0.121 | 0.116 | 0.105 | 0.106 | **0.064** | 0.070 | 0.153 | 0.093 |
| | | F | 34 | 10.0 | 8.1 | 4.6 | 3.9 | 3.8 | 3.9 | 9.2 | **3.6** |
| Docosahexaenoic acid | 56 | E | 1 | 0.089 | 0.093 | 0.089 | 0.214 | 0.102 | **0.070** | 0.281 | 0.072 |
| | | F | 33 | 5.9 | 5.2 | 3.2 | **2.5** | 2.8 | 2.6 | 11.3 | 2.5 |
| Stachyose | 87 | E | 2 | 0.076 | 0.069 | 0.124 | 0.078 | 0.062 | **0.050** | 0.077 | 0.057 |
| | | F | 29 | 10.1 | 8.3 | 4.2 | 3.0 | 3.8 | 3.3 | 4.1 | **2.4** |
| AT-AT | 60 | E | 0.52 | 0.121 | 0.139 | 0.103 | 0.109 | 0.079 | **0.060** | 0.109 | 0.078 |
| | | F | 30 | 10.3 | 8.8 | 4.1 | 4.3 | 4.3 | **3.4** | 8.221 | 5.0 |
| AT-AT-CG-CG | 118 | E | 0.52 | 0.097 | 0.193 | 0.145 | 0.055 | 0.058 | **0.040** | 0.098 | 0.081 |
| | | F | 31 | 16.0 | 20.4 | 5.6 | 5.4 | 5 | **4.6** | 9.8 | 6.4 |
| Buckyball catcher | 148 | E | 0.34 | - | - | - | - | 0.141 | - | 0.157 | **0.112** |
| | | F | 29 | - | - | - | - | **3.7** | - | 17.6 | 4.3 |
| Double-walled nanotube | 370 | E | 0.47 | - | - | - | - | **0.194** | - | 0.387 | 0.219 |
| | | F | 23 | - | - | - | - | 12 | - | 12.3 | **10.6** |
| AVG RANK | | | 7.57 | 6.50 | 6.50 | 4.80 | 3.80 | 2.64 | **1.80** | 5.50 | 2.29 |

*results* compared to other SOTA models, including MACE and VisNet-LSRM, ranking 2nd on average. These results demonstrate the *effectiveness of subgraph enhancement* in practical GeoNGNN with finite configurations.

To explore how subgraph size affects experimental performance, we selected three molecules of different representative scales and observed the effects, as depicted in Figure 6. As anticipated, performance generally improves with increased subgraph size, roughly matching the improved theoretical expressiveness. However, the rule does not always hold: there are cases where increasing the subgraph size leads to degraded performance, which can be attributed to the potential noise and redundant information. It implies that practically, there could exist a conflict between theoretical expressiveness and experimental performance, and striking a balance between them is important.

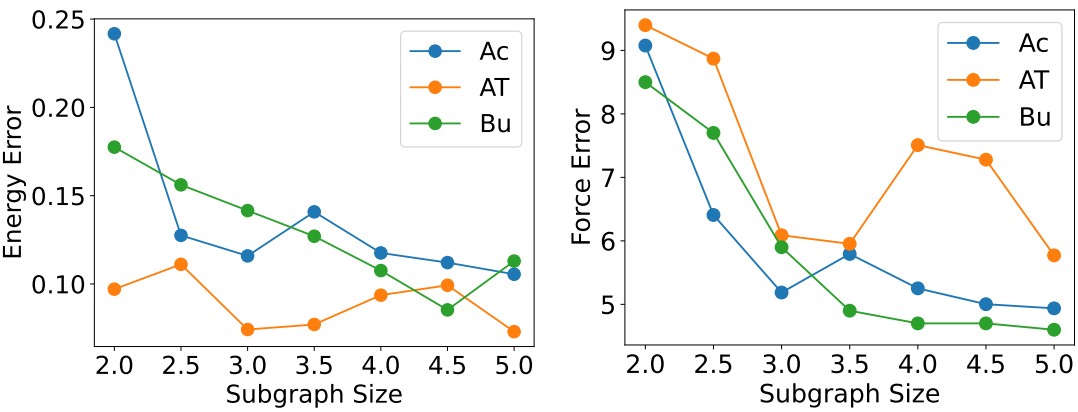

Figure 6: Effect of subgraph size for energy and force prediction on Ac-Ala3-NHMe, AT-AT and Buckyball catcher in MD22. Subgraph size in Å, energy (E) in meV/atom, force (F) in meV/Å.

We further evaluated the inference time and GPU memory consumption of GeoNGNN and several high-performance models. As shown in Table 4, GeoNGNN is more than 2x *faster* than MACE and more than 8x faster than Equiformer, which both utilize high-order equivariant representations and perform tensor products. Additionally, GeoNGNN demonstrates *efficient memory usage*, particularly when compared to Equiformer. This could be attributed to the invariant representation and the simple architecture GeoNGNN adopts.

## F.3 GENERALIZATION ON OUT-OF-DOMAIN DATA: 3BPA

GeoNGNN has demonstrated remarkable expressiveness. However, higher expressiveness only implies a better capability to fit data, and whether this leads to improved inductive bias and generalization power remains uncertain. Therefore, we conducted further evaluations of GeoNGNN on the 3BPA dataset (Kovács et al., 2021), which includes lots of *out-of-domain* data in its test set, providing

Table 4: Efficiency Analysis of Models. The batch size for all molecules is set to 16, except for Bu and Do where it is 4. Models marked with an asterisk (*) utilize a batch size of 8, except for Bu and Do, which use a batch size of 2. The evaluation is conducted on Nvidia A100.

| Molecule | # atoms | MACE | Equiformer* | GeoNGNN |
|---|---|---|---|---|
| Ac-Ala3-NHMe | 42 | 0.166 | 0.305 | **0.070** |
| | | 13400 | 38895 | **12997** |
| Docosahexaenoic acid | 56 | 0.209 | 0.397 | **0.088** |
| | | 17875 | 51565 | **16980** |
| Stachyose | 87 | 0.388 | OOM | **0.161** |
| | | 33146 | OOM | **32430** |
| AT-AT | 60 | 0.195 | 0.361 | **0.083** |
| | | 17310 | 50032 | **15864** |
| AT-AT-CG-CG | 118 | 0.437 | OOM | **0.175** |
| | | 38090 | OOM | **34951** |
| Buckyball catcher | 148 | 0.183 | 0.382 | **0.074** |
| | | 14466 | 47702 | **13891** |
| Double-walled nanotube | 370 | 0.545 | OOM | **0.180** |
| | | 44923 | OOM | **36849** |

a good assessment of the generalization power of geometric models. The dataset comprises molecular dynamic simulations of the molecule 3-(benzyloxy)pyridin-2-amine, with the training set consisting of samples from the trajectory at 300K, and the test sets containing samples from trajectories at 300K, 600K, and 1200K.

Table 5: Root-mean-square errors (RMSE) on 3BPA. Energy (E) in meV, force (F) in meV/Å. Standard deviations are computed over three runs.

| | | NequIP | Allegro | MACE | GeoNGNN |
|---|---|---|---|---|---|
| 300K | E | $3.1\pm0.1$ | $3.84\pm0.08$ | $3\pm0.2$ | $3.76\pm0.29$ |
| | F | $11.3\pm0.2$ | $12.98\pm0.7$ | $8.8\pm0.3$ | $11.77\pm0.34$ |
| 600K | E | $11.3\pm0.31$ | $12.07\pm0.45$ | $9.7\pm0.5$ | $12.49\pm0.38$ |
| | F | $27.3\pm0.3$ | $29.17\pm0.22$ | $21.8\pm0.6$ | $28.91\pm0.71$ |
| 1200K | E | $40.8\pm1.3$ | $42.57\pm1.46$ | $29.8\pm1.0$ | $44.82\pm0.88$ |
| | F | $86.4\pm1.5$ | $82.96\pm1.77$ | $62\pm0.7$ | $89.62\pm1.58$ |

Consistent with prior studies, we present the Root Mean Square Error for energy and force predictions, comparing GeoNGNN with MACE (Batatia et al., 2022), NequIP (Batzner et al., 2022), and Allegro (Musaelian et al., 2023). The results are detailed in Table 5. GeoNGNN demonstrates competitiveness with Allegro, an equivariant model equipped with high-order tensors for learning local equivariant representations. However, when compared to the SOTA method MACE, GeoNGNN exhibits relatively worse performance, especially under higher temperatures, which signifies a greater distribution shift between the testing and training sets. This underscores that despite GeoNGNN's strong theoretical expressiveness, its generalization power may lag behind that of those well-designed equivariant models. As stated in the main text Section 6, this discrepancy could potentially be attributed to the fact that GeoNGNN only adopts invariant representations, whereas equivariant representations could offer superior generalization abilities, particularly in sparse graph cases, as demonstrated in Du et al. (2024).

## F.4 HANDLING VARIOUS PROPERTY PREDICTIONS: QM9

Finally, we evaluate GeoNGNN on QM9 (Ramakrishnan et al., 2014; Wu et al., 2018), which contains approximately 13k well-annotated small molecules, each paired with 12 properties to predict. The results, shown in Table 6, indicate that GeoNGNN performs particularly well in energy-related predictions, such as $U_0$ and $H$.

Table 6: MAE loss on QM9. The best and the second best results are shown in **bold** and underline. We color the cell if GeoNGNN outperforms DisGNN. GeoNGNN demonstrates competitive performance across all models and ranks first in terms of average ranking.

| Target | Unit | SchNet | PhysNet | ComENet | SphereNet | 2F-Dis. | PaiNN | Torchmd | DisGNN | GeoNGNN |
|--------|------|--------|---------|---------|-----------|---------|-------|---------|--------|---------|
| $\mu$ | D | 0.033 | 0.053 | 0.025 | 0.025 | 0.010 | 0.012 | **0.002** | 0.015 | 0.012 |
| $\alpha$ | $a_0^3$ | 0.235 | 0.062 | 0.045 | 0.045 | 0.043 | 0.045 | **0.010** | 0.057 | 0.044 |
| $\epsilon_{HOMO}$ | meV | 41.0 | 32.9 | 23.1 | 22.8 | 21.8 | 27.6 | **21.2** | 34.0 | 23.8 |
| $\epsilon_{LUMO}$ | meV | 34.0 | 24.7 | 19.8 | 18.9 | 21.2 | 20.4 | **17.8** | 28.3 | 21.5 |
| $\Delta\epsilon$ | meV | 63.0 | 42.5 | 32.4 | **31.1** | 31.3 | 45.7 | 38.0 | 45.2 | 33.3 |
| $\langle R^2 \rangle$ | $a_0^2$ | 0.073 | 0.765 | 0.259 | 0.268 | 0.030 | 0.066 | **0.015** | 0.160 | 0.046 |
| ZPVE | meV | 1.7 | 1.39 | 1.20 | **1.12** | 1.26 | 1.28 | 2.12 | 1.88 | 1.4 |
| $U_0$ | meV | 14 | 8.15 | 6.59 | 6.26 | 7.33 | 5.85 | 6.24 | 9.60 | **5.29** |
| $U$ | meV | 19 | 8.34 | 6.82 | 6.36 | 7.37 | 5.83 | 6.30 | 9.88 | **5.35** |
| $H$ | meV | 14 | 8.42 | 6.86 | 6.33 | 7.36 | 5.98 | 6.48 | 9.82 | **5.35** |
| $G$ | meV | 14 | 9.40 | 7.98 | 7.78 | 8.56 | 7.35 | 7.64 | 11.06 | **7.04** |
| $c_v$ | cal/mol/K | 0.033 | 0.028 | 0.024 | **0.022** | 0.023 | 0.024 | 0.026 | 0.031 | 0.023 |
| AVG RANK | | 8.42 | 7.25 | 4.58 | 3.42 | 3.67 | 3.92 | 3.17 | 7.42 | 2.92 |

## G EXPERIMENT SETTINGS

### G.1 DATASET SETTINGS

**Synthetic dataset** We generate the synthetic dataset by implementing the counterexamples proposed by Li et al. (2024).

The dataset includes 10 isolated cases and 7 combinatorial cases, where each case consists of a pair of symmetric point clouds that cannot be distinguished by DisGNN. Figure 7 shows two pairs of selected counterexamples, and more counterexamples are provided in Figure 8.

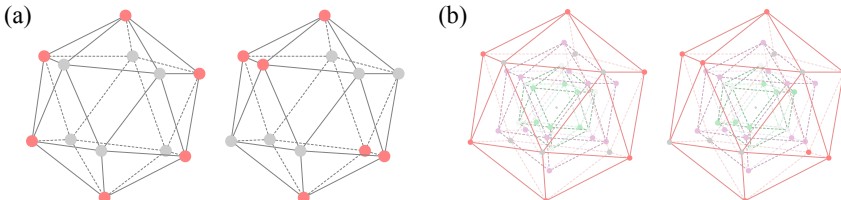

Figure 7: Selected counterexamples in the synthetic dataset. Only colored nodes belong to the point clouds. The grey nodes and "edges" are for visualization purposes only. (a) One isolated case. (b) One combinatorial case.

The 10 isolated cases comprise the following: 6 pairs of point clouds sampled from regular dodecahedrons, 1 pair from icosahedrons, 2 pairs from a combination of one cube and one regular octahedron, and 1 pair from a combination of two cubes. In the combination cases, the relative size is set to 1/2.

The 7 combinatorial cases are obtained through the augmentation of the 6 pairs of point clouds sampled from regular dodecahedrons and 1 pair from icosahedrons, using the method outlined in Theorem A.1 of Li et al. (2024). When augmenting the base point cloud, we consider the *original*, *complementary*, and *all* types of the base graph (Li et al., 2024), respectively.

Given that the counterexamples are designed to test the maximal expressiveness of geometric models, we configure all point clouds in the dataset to be fully connected.

**rMD17 and MD22** The data split (training/validation/testing) in rMD17 is 950/50/the rest, following related works such as Batatia et al. (2022); Gasteiger et al. (2021). For MD22, we adopt the data split specified in Chmiela et al. (2023), which is also consistent with the other works. We train the models using a batch size of 4.

To optimize the parameters $\theta$ of the model $f_\theta$, we use a weighted loss function:

$$\mathcal{L}(\boldsymbol{X}, \boldsymbol{z}) = (1-\rho)|f_\theta(\boldsymbol{X}, \boldsymbol{z}) - \hat{t}(\boldsymbol{X}, \boldsymbol{z})| + \frac{\rho}{m}\sum_{i=1}^{n}\sqrt{\sum_{\alpha=1}^{3}(-\frac{\partial f_\theta(\boldsymbol{X}, \boldsymbol{z})}{\partial \boldsymbol{x}_{i\alpha}} - \hat{F}_{i\alpha}(\boldsymbol{X}, \boldsymbol{z}))^2},$$

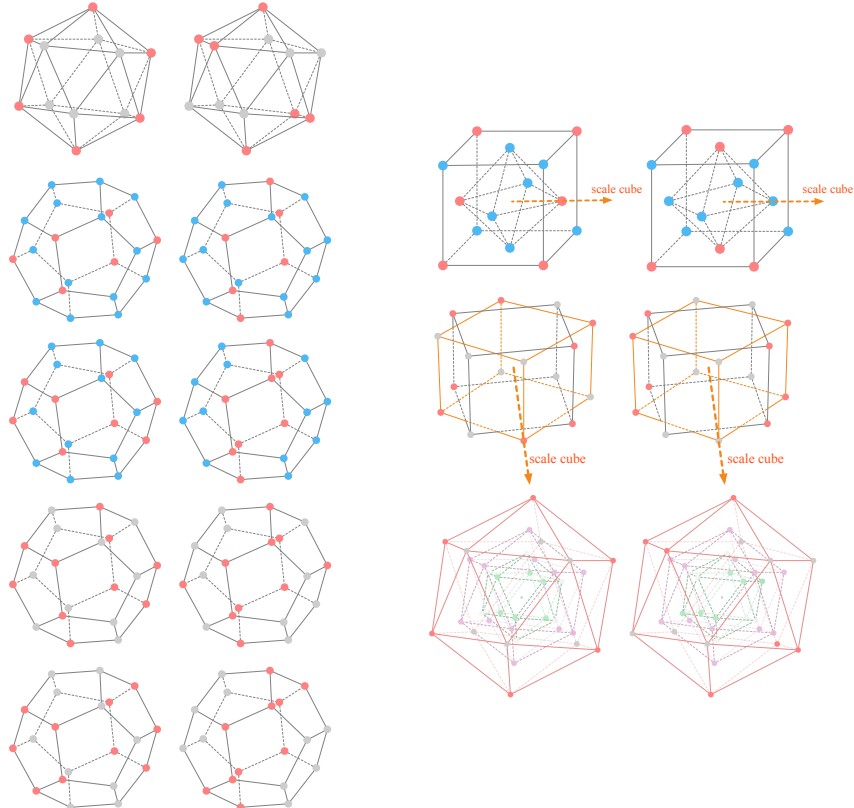

Figure 8: Pairs of point clouds that cannot be distinguished by DisGNN, taken from Li et al. (2024). The nodes are arranged on the surfaces of regular polyhedra, only red or blue nodes are part of point clouds; the grey nodes and the "edges" are included solely for vi- sualization purposes. For those subfigures with two node colors, nodes of the same color in the left and right point clouds represent one specific pair of indistinguishable point clouds by GeoNGNN. The label "Scale cube" signifies that the relative sizes of the two regular polyhedra can vary arbitrarily. Each subfigure, except the bottom-rightmost one, represents an isolated counterexample. The bottom-rightmost case illustrates a combinatorial counterexample, constructed by combining multiple instances of the top-left pair in an "origin-all-complementary" pattern. Comprehensive explanations of these counterexamples are provided in Li et al. (2024).

where $X \in \mathbb{R}^{n \times 3}$ and $z \in \mathbb{R}$ represents the $n$ atoms' coordinates and atomic number respectively, $\hat{t}$ and $\hat{F}$ represents the molecule's energy and forces acting on atoms respectively. The force ratio $\rho$ is chosen as 0.99 or 0.999.

In rMD17, the experimental results of NequIP (Batzner et al., 2022), GemNet (Gasteiger et al., 2021), are sourced from Musaelian et al. (2023). The result of PaiNN (Schütt et al., 2021) is sourced from Batatia et al. (2022). The results of 2-F-DisGNN (Li et al., 2024) and MACE (Batatia et al., 2022) are reported by their original papers. We trained and evaluated DimeNet (Gasteiger et al., 2019) using its implementation in Pytorch Geometric (Fey & Lenssen, 2019) since the original paper did not report results on this dataset, and we use the default parameters (We find that the default parameters produce the best performance).

In MD22, the results of sGDML and MACE (Kovacs et al., 2023) are sourced from Kovacs et al. (2023). The results of PaiNN (Schütt et al., 2021), TorchMD-Net (Thölke & De Fabritiis, 2021), Allegro (Musaelian et al., 2023), and Equiformer (Liao & Smidt, 2022) are obtained from the work presented by Li et al. (2023).

**3BPA**   Following Batatia et al. (2022), we split the training set and validation set at a ratio of 450/50, utilizing pre-split test sets for evaluation. We use a batch size of 4, an identical objective function to that in rMD17 and MD22, and a force ratio of 0.999 were employed. The results of all other models are sourced from Batatia et al. (2022).

**QM9**   Following Li et al. (2024), we split the training set and validation set at a ratio of 110K/10K, utilizing pre-split test sets for evaluation. We use a batch size of 32. See the following for details of model hyper-parameters.

### G.2   MODEL SETTINGS

**RBF**   Following previous work, we use radial basis functions (RBF) $f_{\mathrm{e}}^{\mathrm{rbf}} : \mathbb{R} \to \mathbb{R}^{H_{\mathrm{rbf}}}$ to expand the euclidean distance between two nodes into a vector, which is shown to be beneficial for inductive bias Gasteiger et al. (2019); Li et al. (2024). We use the same expnorm RBF function as Li et al. (2024), defined as

$$f_e^{\mathrm{rbf}}(e_{ij})_k = e^{-\beta_k(\exp(-e_{ij}) - \mu_k)^2}, \tag{20}$$

where $\beta_k, \mu_k$ are coefficients of the $k^{\mathrm{th}}$ basis.

**DisGNN**   In each stage of the DisGNN, we augment its expressiveness and experimental performance by stacking dense layers or residual layers DisGNN's architecture is described in Figure 9.

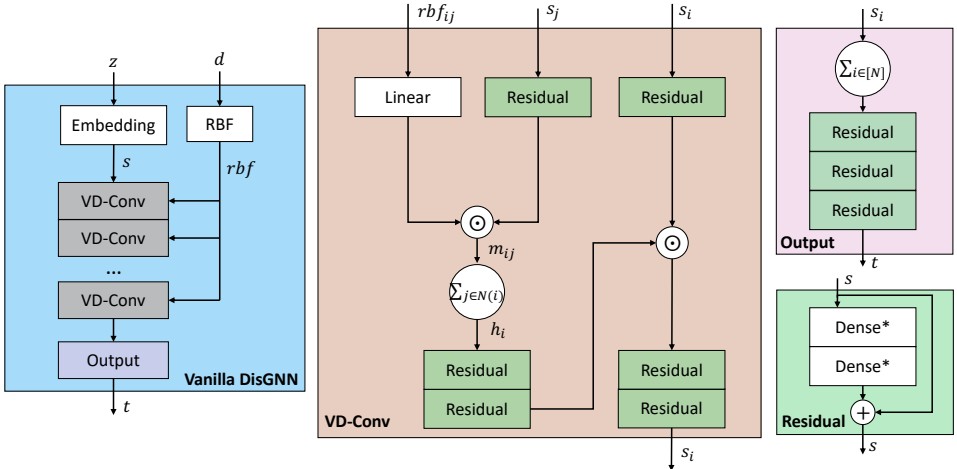

Figure 9: Architecture of DisGNN. $z, d, t$ represent the atomic number, euclidean distance and the target output, respectively. Each Linear block represents an affine transformation with learnable parameters $W, b$, while the Dense* block extends the Linear block by incorporating a non-linear activation function (pre-activation). $\odot$ represents Hadamard product.

We set the number of VD-Conv layers to 7 for rMD17 and 6 for MD22. The hidden dimension is set to 512 and the dimension of radial basis functions (RBF) is set to 16. The message passing cutoff is set to 13Å for rMD17 and 7Å for MD22. We employ the polynomial envelope with $p = 6$, as proposed in Gasteiger et al. (2019), along with the corresponding cutoff.

**NGNN**   GeoNGNN is based on the high-level framework of NGNN proposed in (Zhang & Li, 2021). NGNN learns over topological graphs by nesting a base GNN such as GCN (Kipf & Welling, 2016), GIN (Xu et al., 2018b), or GraphSAGE (Hamilton et al., 2017), as illustrated in Figure 10. For completeness, we include formal descriptions provided in (Zhang & Li, 2021).

Formally, given a topological graph $G = (V, E)$, where $V = \{1, 2, \ldots, n\}$ represents the set of nodes and $E \subseteq V \times V$ represents the set of edges, NGNN first defines $k$-hop ego subgraphs $G_k^v$ for each node $v \in V$. Each $k$-hop ego subgraph $G_k^v$ is induced by the nodes within $k$-hops from $v$ (including $v$ itself) and the edges connecting these nodes.

For each rooted subgraph $G_k^w$, NGNN applies a base GNN to perform $T^{\text{in}}$-rounds of message passing. Let $v$ be any node appearing in $G_k^w$. Denote the hidden state of $v$ at time $t$ in subgraph $G_k^w$ as $h_{v,G_k^w}^t$. The initial hidden state $h_{v,G_k^w}^0$ is typically set to the node's raw features, such as embeddings of atomic numbers. When explicit node marking is adopted, which is the default in GeoNGNN, $h_{v,G_k^w}^0$ is further fused with a unique mark embedding to distinguish the root node within its subgraph.

After $T^{\text{in}}$-rounds of message passing, a subgraph pooling operation aggregates the node embeddings $\{\!\{h_{v,G_k^w}^{T^{\text{in}}} \mid v \in G_k^w\}\!\}$ into a single subgraph representation $h_{G_k^w}$:

$$h_{G_k^w} = R_{\text{subgraph}}\left(\{\!\{h_{v,G_k^w}^{T^{\text{in}}} \mid v \in G_k^w\}\!\}\right),$$

where $R_{\text{subgraph}}$ is the subgraph pooling function that summarizes all node-level information within the subgraph. This subgraph representation $h_{G_k^w}$ serves as the final representation of the root node $w$ in the original graph.

Subsequently, an outer GNN performs $T^{\text{out}}$-rounds of message passing, obtaining the hidden representation $h_v^t$ of node $v$ at step $t$, where the initial representation is $h_v^0 = h_{G_k^v}$. Finally, the graph-level representation $h_G$ is produced:

$$h_G = R_{\text{graph}}\left(\{\!\{h_v^{T^{\text{out}}} \mid v \in V\}\!\}\right),$$

where $R_{\text{graph}}$ is a global pooling function that aggregates node-level embeddings into a comprehensive graph-level representation. This process enables NGNN to capture hierarchical representations of the graph, leveraging both local (subgraph-level) and global (graph-level) features.

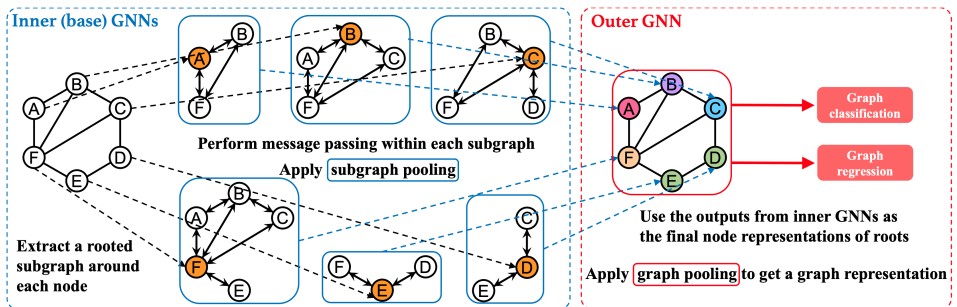

Figure 10: Illustration of NGNN (Zhang & Li, 2021), taken from (Zhang & Li, 2021). NGNN operates by first extracting rooted subgraphs around each node in the original graph. A base GNN with a subgraph pooling layer is then applied independently to each rooted subgraph to compute its representation. The resulting subgraph representation serves as the original feature for the root node in the original graph. Subsequently, an outer GNN is applied to the original graph, leveraging these node features and original graph structures to extract final graph-level representations.

**GeoNGNN** GeoNGNN serves as the geometric counterpart to NGNN, operating on point clouds rather than topological graphs in Figure 10. The high-level architecture of GeoNGNN is described in the main body. The inner DisGNN, which runs on the ego subgraph of each node, is similar to the DisGNN block shown in Figure 9. However, the inner DisGNN additionally performs node marking and distance encoding for each node $j$ in node $i$'s subgraph *before* passing their representations to VD-Conv layers:

$$h_{ij} \leftarrow h_j \odot \text{MLPs}(\text{RBF}(d_{ij})) \odot \text{MLPs}(\text{Emb}(\mathbb{1}_{j=i}))) \tag{21}$$

where $h_{ij}$ represents node $j$'s representation in $i$'s subgraph.

The outer DisGNN additionally fuses the subgraph representation $t_i$ produced by the inner DisGNN for node $i$ with $h_i$ obtained from the embedding layer:

$$h_i \leftarrow \text{MLPs}([h_i, t_i])) \tag{22}$$

The hidden dimension and message passing cutoff are set to the same values as those in DisGNN. The subgraph size is set to 13 Å for rMD17, 5 Å for MD22, 6 Å for 3BPA, 13 Å for QM9 and the

maximal subgraph size is set to 25. We set $(N_{\text{in}}, N_{\text{out}})$ as $(5, 2)$ for rMD17, $(3, 3)$ for MD22, $(6, 2)$ for 3BPA and $(4, 1)$ for QM9. It is worth mentioning that the total number of convolutional layers is the same as that of DisGNN. This ensures that the receptive field of the two models is roughly the same, thereby guaranteeing a relatively fair comparison.

We optimize all models using the Adam optimizer (Kingma & Ba, 2014), incorporating exponential decay and plateau decay learning rate schedulers, as well as a linear learning rate warm-up. To mitigate the risk of overfitting, we employ early stopping based on validation loss and apply exponential moving average (EMA) with a decay rate of 0.99 to the model parameters during the validation and testing phases. The models are trained on Nvidia RTX 4090 and Nvidia A100 (80GB) GPUs. For rMD17, training hours for each molecule range from 20 to 70. For MD22, the training hours ranged from 35 to 150 on Nvidia 4090 for all molecules except Stachyose (due to longer convergence time) and Double-walled nanotube (the largest molecule necessitating an A100), which required about 210 and 70 GPU hours on the Nvidia 4090 and 80GB A100, respectively. For 3BPA, the training hours is around 10 GPU hours on Nvidia 4090. For QM9, the training hour is around 36 to 100 on the Nvidia 4090.

# H  GEONGNN-C: SE(3)-COMPLETE VARIANT OF GEONGNN

## H.1  DEFINITION AND THEORETICAL ANALYSIS OF GEONGNN-C

In certain chemical-related scenarios, it is important to guarantee that the outputs of models exhibit invariance under permutations, transformations, and rotations of point clouds, while **exhibiting differentiation when the point cloud undergoes mere reflection**. This distinction is crucial as pairs of enantiomers, may demonstrate vastly contrasting properties, such as binding affinity. In this section, we propose the SE(3)-complete invariant of GeoNGNN, which is capable of embracing chirality and producing complete representations for downstream tasks.

Generally speaking, GeoNGNN-C is designed by simply replacing the undirected distance in GeoNGNN with the directed distance. The formal definition is:

**Definition H.1.** The fundamental architecture of GeoNGNN remains the same, with the exception that the inner GNN for node $k$'s subgraph performs the subsequent message-passing operation:

$$h_{ki}^{(l+1)} = f_{\text{update}}(h_{ki}^{(l)}, \{\!\!\{ (h_{kj}^{(l)}, d_{k,ij}) \mid j \in N_k(i) \}\!\!\}),$$

where $h_{ki}^{(l)}$ represents node $i$'s representation at layer $l$ in node $k$'s subgraph, and the directed distance, $d_{k,ij}$, is defined as:

$$d_{k,ij} = f_{\text{dist}}(\text{sign}(\vec{r}_{ci} \times \vec{r}_{cj} \cdot \vec{r}_{ck}), d_{ij}),$$

where $c$ is the geometric center of the point cloud.

As can be easily validated, directed distance is invariant under SE(3)-transformation while undergoing a sign reversal after reflecting the point cloud. Consequently, GeoNGNN-C retains SE(3)-invariance while possessing the potential to discriminate enantiomers. Significantly, we show that GeoNGNN-C is SE(3)-complete, able to distinguish all pairs of enantiomers:

**Theorem H.2.** *(SE(3)-Completeness of GeoNGNN-C) When the conditions described in Theorem 5.1 are met, GeoNGNN-C is SE(3)-complete.*

*Proof.* We first note that the proof is highly similar to that in the proof of Theorem 5.1.

Let us consider a subgraph containing a node $i$, where we assume that $i$ is distinct from the geometric center $c$. By imitating the proof of Theorem 5.1, we can ascertain the following:

After three rounds of the Chiral DisGNN (i.e., the DisGNN that incorporates the directed distance as a geometric representation as described in Definition H.1), node representations $h_j^{(3)}$ for any node $j$ can encode the directed distances $d_{ji}$, $d_{jc}$, and $d_{ic}$. This enables the completion of the triangular distance encoding necessary for reconstruction in Lemma B.6.

Next, we employ a modified version of the proof from Lemma B.6 to demonstrate that with an additional two rounds of the Chiral DisGNN, we can derive the *SE(3)-* and permutation-invariant identifier of the point cloud $P$.

In the proof of Lemma B.6, we first search across all the nodes to find two nodes $a$ and $b$ that form the minimal dihedral angles $\theta(aic, bic)$, and then identifies the coordinates of $a, b, i, c$ *up to Euclidean isometry* from the all-pair distances $D = \{(m, n, d_{mn}) \mid m, n \in \{a, b, i, c\}\}$. However, since Chiral DisGNN embeds the geometry using directed distance, we now have $D_i = \{(m, n, d_{i,mn}) \mid m, n \in \{a, b, i, c\}\}$ instead. We are now able to identify the coordinates of $a, b, i, c$ *up to SE(3)-transformation* (i.e., Euclidean isometry without reflection) from $D_i$ in the following way:

1. Fix $x_i = (0, 0, 0)$.

2. Fix $x_c = (d_{ic}, 0, 0)$.

3. Fix $x_a$ in plane $xOy+$ and calculate its coordinates using $d_{ia}, d_{ca}$.

4. Calculate the coordinates of $x_b$ using $d_{ib}, d_{cb}, d_{ab}$, and *determine whether $b$ is on $z+$ side or $z-$ side using the signal embedded in $d_{i,ab}$*.

Note that the orientation relevant to SE(3) transformation is already determined at this stage. We proceed with the remaining steps of the proof outlined in Lemma B.6 and reconstruct the whole geometry in a *deterministic* way based on the known 4-tuple $abic$, and therefore reconstruct the whole geometry up to permutation and SE(3)-transformation.

Similarly, note that such subgraphs $i$, where $i$ is distinct from the geometric center $c$, always exist as GeoNGNN-C traverses all node subgraphs. Hence, we have established the SE(3)-Completeness of GeoNGNN-C.

$\square$

## H.2 EXPERIMENTAL EVALUATION OF GEONGNN-C

**Datasets and tasks** We evaluate the ability of GeoNGNN-C to distinguish enantiomers and learn meaningful chirality-relevant representations through three tasks on two datasets (Adams et al., 2021; Gaiński et al., 2023): 1) Classification of tetrahedral chiral centers as R/S. R/S offers a fundamental measure of molecular chirality, and this classification task serves as an initial evaluation of the model's capacity to discriminate enantiomers. The underlying dataset contains a total of 466K conformers of 78K enantiomers with a single tetrahedral chiral center. 2) Enantiomer ranking task and binding affinity prediction task. The two tasks share the same underlying dataset, while the second task requires models to predict the binding affinity value for each enantiomer directly, and the first requires the model to predict which enantiomer between the corresponding pair exhibits higher such values. Chirality is essential here because enantiomers often exhibit distinct behaviors when docking in a chiral protein pocket, thus leading to differences in binding affinity. The underlying dataset contains 335K conformers from 69K enantiomers carefully selected and labeled by Adams et al. (2021).

**Model architecture and training/evaluation details** In the task of R/S classification, GeoNGNN-C adopts a three-layer outer GNN positioned *before* the inner GNNs. This configuration is designed to enhance the node features first, facilitating the subsequent extraction of chirality-related properties within the inner GNNs. For the other two tasks, GeoNGNN-C consists of five inner layers and one outer layer, with the outer GNN placed after the inner GNN. The model is trained with L1 loss. Specifically, if predicted affinity difference between two enantiomers falls below the threshold of 0.001, it is considered that the model lacks the capability to discriminate between the two enantiomers. Note that all the evaluation criteria are consistent with previous works Adams et al. (2021); Gaiński et al. (2023) for fair comparisons.

**Models to compare with** We compare GeoNGNN-C against previous high-performance models, SphereNet (Liu et al., 2021), ChiRo (Adams et al., 2021), ChiENN (Gaiński et al., 2023), and other baseline models in (Pattanaik et al., 2020; Adams et al., 2021; Gaiński et al., 2023). It is worth noting that, for the last two tasks, the docking scores are labeled at the stereoisomer level, meaning that all different conformers of a given enantiomer are assigned the same label, specifically the best score, by Adams et al. (2021). As a result, 2D Graph Neural Networks (GNNs) such as DMPNN+tags and models that possess **invariance to conformer-level transformations**, such as bond rotations, including ChiRo (Adams et al., 2021), Tetra-DMPNN (Pattanaik et al., 2020) and

Table 7: Results on chiral-sensitive tasks compared to reference models. We **bold** the best result among models and underline the second best ones.

| Model | R/S Accuracy ↑ | Enantiomer ranking R. Accuracy ↑ | Binding affinity MAE ↓ |
|---|---|---|---|
| DMPNN+tags | - | 0.701±0.003 | 0.285±0.001 |
| SphereNet | 0.982±0.002 | 0.686±0.003 | - |
| Tetra-DMPNN | 0.935±0.001 | 0.690±0.006 | 0.324±0.02 |
| ChIRo | 0.968±0.019 | 0.691±0.006 | 0.359±0.009 |
| ChiENN | **0.989±0.000** | **0.760±0.002** | 0.275±0.003 |
| GeoNGNN-C | 0.980±0.002 | 0.751±0.003 | **0.254±0.000** |

ChiENN (Gaiński et al., 2023), inherently exhibit significantly better data efficiency and usually exhibit better performance. GeoNGNN-C and SphereNet do not exhibit such invariance, and are trained on 5 conformers for each enantiomer as data augmentation.

**Results** The experimental results for the three tasks are presented in Table 7. It can be observed that GeoNGNN-C demonstrates a competitive performance compared to SOTA methods on the R/S classification task. Importantly, despite the simple and general model design, GeoNGNN-C outperforms models specifically tailored for the last two tasks, such as ChIRo and ChiENN, which incorporates invariance to bond rotations. It successfully learns the *approximate invariance* from only 5 conformers per enantiomer and achieves the second-best result (very close to the best one) on the enantiomer ranking task and new SOTA results on the binding affinity prediction task. These results reveal the potential of GeoNGNN-C as a simple but theoretically chirality-aware expressive geometric model.

