# OpenReview forum: "On the Completeness of Invariant Geometric Deep Learning Models"
_ICLR.cc/2025/Conference — ICLR 2025 Poster_

### Official Review · Reviewer_PLLb · 2024-11-02

**Soundness:** 2
**Presentation:** 3
**Contribution:** 2
**Rating:** 6
**Confidence:** 2

**Summary:**

This paper studies the expressiveness power of message-passing neural networks incorporating pairwise distance between graph nodes, showing the near E(3)-completeness. Furthermore, the authors study the subgraph graph neural networks, which can achieve E(3)-completeness. Therefore, it is possible to make DimeNet, GemNet, and SphereNet to achieve E(3)-completeness.

**Strengths:**

The paper symmetrically studies the problem of E(3)-completeness geometric graph neural networks.

**Weaknesses:**

The work is based on global connectivity assumption, and this assumption significantly limits this work. Also, the experimental results seem to be quite weak.

**Questions:**

I have two questions.

1. Do you have any insight on achieving E(3)-completeness for frame-based approaches?

2. Can you comment on achieving E(3)-completeness by using node features beyond pairwise distances, e.g., dihedral angles?

---

> ### Author Response · Authors · 2024-11-21
> **Author Response**
>
> Thank you very much for your thoughtful comments and questions! Below, we address each of your points in detail, and we hope our responses provide clarity and insight.
>
> ---
>
> > The work is based on global connectivity assumption, and this assumption significantly limits this work.
> >
>
> **Response:**
>
> We acknowledge your concern regarding the Fully-Connected Condition, and we have explicitly and extensively discussed this limitation in Section 5.4. Here, we would like to further emphasize the *reasonableness* of this assumption in the context of our study:
>
> 1. **Global connectivity is a standard assumption in expressiveness research for *invariant* models.**
>
>     As discussed in Section 5.4, plenty of previous works in the literature adopt this assumption, given the challenges faced by invariant models in preserving local patterns during message passing. Unlike equivariant methods, which can rely on equivariant features to retain local information even in sparse graphs, invariant methods inherently lose such patterns.
>
>
> Additionally, we respectfully refer you to our response to Reviewer **jQxs** (for question “Several studies [1], [2], [3], [4] have explored….”), where we further justify the reasonableness of this assumption for invariant methods.
>
> > Also, the experimental results seem to be quite weak.
> >
>
> **Response:**
>
> We appreciate your concern on the experimental results! Actually, experiments presented in the main text are primarily intended to support the theoretical claims of the paper (e.g., the near-completeness of DisGNN and the completeness of other methods). For additional empirical evidence, we would like to respectfully draw your attention to the **extensive real-world experiments on GeoNGNN provided in Appendix D.**
>
> We demonstrate that GeoNGNN achieves promising performance despite its simple design. For instance, it surpasses advanced methods such as DimeNet, GemNet, and PaiNN on MD17; MACE and Equiformer on MD22; and ComENet and SphereNet on QM9.
>
> Furthermore, we outline potential directions for improving GeoNGNN to further enhance its empirical performance. While we acknowledge that there is room for improvement, we view these experiments as a foundational step for future work, given the theoretical emphasis of the current study.
>
> > Do you have any insight on achieving E(3)-completeness for frame-based approaches?
> >
>
> **Response:**
>
> Thank you for this insightful question! Some frame-based methods (e.g., [1], [2]) indeed face challenges with symmetric structures where global-level frames may degenerate.
>
> A potential way forward is **breaking the overall symmetry through node marking, as demonstrated by GeoNGNN.** By marking a node, frames can be calculated on the symmetry-broken point cloud (e.g., using PCA while ignoring the marked node). To ensure permutation invariance, the results can be averaged across different marked nodes. We believe this strategy could be a promising direction for extending frame-based approaches toward E(3)-completeness.
>
> > Can you comment on achieving E(3)-completeness by using node features beyond pairwise distances, e.g., dihedral angles?
> >
>
> **Response:**
>
> We greatly appreciate the reviewer’s comment on this topic. Actually, complete models we establish like DimeNet and GemNet provide good examples of incorporating angles and dihedral angles, and their completeness is formally established in **Theorem 5.5.** Below, we briefly highlight some key points.
>
> 1. **Achieving completeness using *angle* information.**
>
>     To prove this, we show that models like DimeNet can effectively *implement* GeoNGNN. For example, with angle information $\theta_{aij}$ and distances $(d_{ai}, d_{ij})$, the remaining distance $d_{aj}$ can be calculated. This allows us to express DimeNet equivalently using only distance information, aligning it with our proof framework. Details are provided in Appendix H.5.1.
>
> 2. **Dihedral angles as redundant for completeness.**
>
>     While angle information alone is sufficient for achieving theoretical completeness when properly integrated like DimeNet, dihedral angles (as used in GemNet) are redundant in fully connected scenarios, as noted in **lines 398–399.** However, these features may enhance generalization in practical applications, as demonstrated in empirical results.
>
> ---
>
> We hope these responses could address your concerns and provide clarity on the limitations, insights, and future directions of our work. Thank you again for your thoughtful feedback and the opportunity to further explain our contributions, and we would greatly appreciate it if you could **raise your score** accordingly.
>
> [1] FAENet: Frame Averaging Equivariant GNN for Materials Modeling
>
> [2] Frame Averaging for Invariant and Equivariant Network Design

---

> > ### Comment · Reviewer_PLLb · 2024-11-24
> > **Reply**
> >
> > I thank the authors for providing detailed responses to my questions and provide important insights. However, I think the fundamental weaknesses are not addressed.
> >
> > I rank this paper as a borderline paper and lean to reject it in its current form. Also, I do not feel it is unacceptable if this paper is accepted.

---

> > > ### Author Response · Authors · 2024-11-25
> > > **Author Response**
> > >
> > > Thank you for taking the time to review our paper! Your comments have been invaluable in guiding us to improve our work, and we truly thank you for your effort and engagement.
> > >
> > > We deeply appreciate your insights and acknowledge that some concerns may still require further discussion. We sincerely respect your perspective and **remain open to constructive dialogue to address any unresolved issues**.  We are looking forward to the opportunity for continued discussion!

---

### Official Review · Reviewer_y3SR · 2024-11-02

**Soundness:** 3
**Presentation:** 2
**Contribution:** 3
**Rating:** 6
**Confidence:** 2

**Summary:**

The authors of the paper prove that certain families of models are not only
invariant with respect to the Euclidian group and permutation group, but also
that classes of models distinguish the orbits of $\mathbb{R}^3$ under the
action of $E(3)$. An extended analysis of the expressivity of DisGNN is
provided and it is shown that this network architecture is nearly $E(3)$
complete. As a last contribution an analysis is provided for various families
of neural networks and conditions are provided under which they are
$E(3)$-complete. The   theoretical   results   are   verified   by experiments
on  the QM9  dataset  and  a synthetic  dataset  with designed edge cases.

**Strengths:**

The authors have provided both extensive proofs as well as extensive analysis
to their claims. Overall the presentation and intend is clear and definitions
are well-thought out and the authors provide a good heuristic insight with each
introduced theorem and definition which is nice. The extensive analysis of both
DisGNN and GeoNGNN shows that the work is of good quality and looks to be of
good quality to the reviewer. All theorems come with extensive proofs and with
a intuition which is helpful for the non-mathmatical audience. The quality of
the content, such as originality and potential impact, is harder to asses since
the reviewer is not familiar expressivity research.

**Weaknesses:**

To the reader it seems that some of the definitions are somewhat convolved and
some simplification and clarity in the definitions might improve reading. Some
of the definitions, while they might be customary in the machine learning
literature, are somewhat unfortunately choses from a mathematical perspective.
Completeness of a space in the mathematical sense implies that each Cauchy
sequence has a limit within that space. A second example is the use of the term
isomorphism. While not wrong, a better phrasing is to say that the two point
cloud lie in the same orbit with respect to the action of the Euclidian group
acting on the tensor product of copies of $\mathbb{R}^3$. The current phrasing
might be better if is more in line with terminology used in machine learning.

**Questions:**

- How does this method for expressivity generalize to different types of architectures? To the reviewer it seems that this method for showing is very specific and would be difficult to generalize to other types of equivariant architectures acting on point clouds.

---

> ### Author Response · Authors · 2024-11-21
> **Author Response**
>
> We sincerely thank you for taking the time to review our work and for providing thoughtful feedback and constructive suggestions! Below, we address your comments in detail.
>
> ---
>
> > Some of the definitions, while they might be customary in the machine learning literature, are somewhat unfortunately choses from a mathematical perspective. Completeness of a space in the mathematical sense implies that each Cauchy sequence has a limit within that space
> >
>
> **Response:**
>
> We greatly appreciate your expert remarks on the use of terms like *completeness* and *isomorphism!*
>
> To clarify, we closely adhere to conventions established in prior works in this area ([1-3]). In these contexts, *(in)completeness* typically refers to the (in)ability of geometric models to distinguish point clouds, differing from its mathematical sense related to Cauchy sequences. Similarly, the term *isomorphism* aligns with recent literature ([4]), where the aim has been to bridge the expressiveness research in geometric deep learning with that of traditional graph learning, particularly in relation to the graph isomorphism problem.
>
> That said, we greatly value the reviewer’s perspective on these terms and will carefully consider modifying the terminology if necessary to avoid ambiguity and ensure clarity from both mathematical and machine learning perspectives.
>
> > How does this method for expressivity generalize to different types of architectures? To the reviewer it seems that this method for showing is very specific and would be difficult to generalize to other types of equivariant architectures acting on point clouds.
> >
>
> **Response:**
>
> Thank you for raising this excellent question. Our approach to establishing expressiveness is primarily focused on invariant models working with distance graphs, as supported by the reconstruction proofs (see Appendix H.2 and H.3). However, we offer the following thoughts on generalizing this framework:
>
> 1. **Generalizing to invariant models using other invariant features such as angles:**
>
>     Models such as DimeNet that incorporate higher-order geometric features like angles can still be analyzed within our framework *by transforming these features into distances*. For instance, a function using $(\theta_{AC}, \theta_{AB}, d_{AC})$ as input can be equivalently expressed in terms of $(d_{AC}, d_{AB}, d_{BC})$. This equivalence allows us to evaluate whether the provided geometric information is sufficient for completeness. Indeed, we demonstrate this in Appendix H.5, where we prove the completeness of models like DimeNet, SphereNet, and GemNet under our framework.
>
> 2. **Generalizing to equivariant models:**
>
>     Equivariant models typically learn higher-order geometry using vector or tensor products. These operations often implicitly capture invariant features, for example, by runtime geometry calculation methods in [5]. For a more simple example, the vector product $\vec{r}\_{ij} \cdot \vec{r}\_{ik}$ captures angular information, such as the angle $\angle jki$. A promising direction for extending our proof framework to equivariant models would involve demonstrating that these operations can *extract the necessary invariant features required for geometric reconstruction in our proof*, thus satisfying the completeness criteria established in our work.
>
>
> ---
>
> We hope this response provides clarity and addresses your concerns. Once again, we are very grateful for your thoughtful questions and constructive feedback, which have inspired us to further refine and expand upon our work. We are looking forward to further discussion, and would greatly appreciate it if you could **raise your score** when you feel your concerns are addressed.
>
> [1] Is Distance Matrix Enough for Geometric Deep Learning? NIPS 2023
>
> [2] Complete Neural Networks for Complete Euclidean Graphs. AAAI 2023
>
> [3] Incompleteness of graph neural networks for points clouds in three dimensions.
>
> [4] On the Expressive Power of Geometric Graph Neural Networks.
>
> [5] ViSNet: an equivariant geometry-enhanced graph neural
> network with vector-scalar interactive message passing for
> molecules

---

> > ### Author Response · Authors · 2024-11-30
> > **Looking forward to Your Feedback as the Discussion Deadline Nears**
> >
> > Dear Reviewer y3SR,
> >
> > We greatly appreciate your thoughtful review of our paper and your recognition of its theoretical contributions. Your feedback regarding the presentation weaknesses has been invaluable in helping us refine our work.
> >
> > In response to your comments, we have addressed your concerns by clarifying the terminology choices and elaborating on the extension of our framework to equivariant models. Additionally, during the rebuttal process, we made **revisions to enhance the paper’s clarity and rigor**, which we hope could **further address your concerns regarding the “presentation.”** These revisions include:
> >
> > 1. More precise statements regarding the fully-connected conditions in both the abstract and introduction.
> >
> > 2. Expanded introduction of GeoNGNN in the main text to improve accessibility and understanding.
> >
> > 3. Details about the original NGNN and counterexamples in the Appendix to ensure self-containment and support for readers.
> >
> > We sincerely hope that our responses and updates provide deeper insights into our work and effectively resolve the issues you highlighted. We look forward to further discussions and would be grateful if you might consider **raising your score if you find our improvements satisfactory**.

---

### Official Review · Reviewer_JK57 · 2024-11-04

**Soundness:** 3
**Presentation:** 1
**Contribution:** 3
**Rating:** 6
**Confidence:** 3

**Summary:**

The paper offers the following contributions:
1) Introduces and defines the notion of "Identify" for invariant GNNs, positioned between distinguishability and completeness.
2) Provides a characterization for the incompleteness of DisGNN
3) Proposes GeoNGNN to ensure indentification of the cases where DisGNN is incomplete
4) Demonstrates that several established invariant GNNs are capable of completeness

**Strengths:**

The paper introduces a novel conceptual framework for understanding the efficacy of certain invariant architectures. This is further supported through theoretical analysis and empirical studies. Additionally, it proposes a framework for the development of future architectures extending the impact and significance of the work.

**Weaknesses:**

The paper lacks sufficient empirical evidence to support its theoretical analysis, significantly reducing the overall significance and impact of the work. The selected real world experiments emphasize datasets which lack conformers or nearly isomorphic point clouds. Furthermore, the main text does not provide adequate evidence to demonstrate the advantages of GeoNGNN over the existing complete invariant architectures.

Additionally, the excessive use of bold text and the absence of a clear outline in the introduction make it challenging to follow and clearly understand the contributions of the paper.

**Questions:**

1) The excessive use of bold text and the absence of a clear outline make the paper’s contributions difficult to discern. Could the authors consider restructuring the introduction for better clarity?

2) I find that the statements in the section **Theoretical Characterization vs Practical Use** rely on the example C.2. How increasing the sparsity beyond this simple example is understudied despite the authors strong claims that relaxing the fully-connected condition leads to better expressiveness of GeoNGNN compared to DisGNN.

3) There is no supporting evidence for GeoNGNN over existing architectures in the primary paper. Additionally, there is no comparative analysis involving node feature information generated by a complete invariant function. Could the authors address this gap?

4) A significant portion of the QM9 dataset consists of non-symmetric structures. What are the proportions of indistinguishable data restricted to the subset of QM9 that includes only symmetric structures?

5) In the QM9 noise study, the significant reduction in non-distinguishable point clouds occurs near what appears to be the level of reported error in the QM9 dataset. Given the reported error of 0.1Å, how is this error rescaled based on the applied scaling coefficient?

6) Distinguishing structures on QM9, which lacks conformers, does not seem to be as important as datasets which contain conformers or very nearly isomorphic point clouds.The most compelling analysis appears to come from the study of MD17 but with mixed results. GeoNGNN appears to do particularly well on Benzene which is highly symmetric. How does Benzene behave under the noise tolerance study?

7) Typically, ModelNet40 is sampled to avoid handling large point clouds. It is unclear from the text whether the entire mesh or a sampled version is used. If sampled uniformly, there is no guarantee that the symmetries are preserved. Could the authors clarify this in the text?

8) The selection of ModelNet40 does not seem to rigorously test the theoretical claims of the paper, which focus on nearly isomorphic point clouds. Could the authors provide more rigorous testing on datasets that better align with their theoretical focus?

9) It is unclear from the text and appendix what each structure in the synthetic dataset represents, how these structures were constructed, and why they are significant. Could the authors provide more detailed explanations on the construction and relevance of these synthetic structures?

---

> ### Author Response · Authors · 2024-11-21
> **Author Response [1/3]**
>
> Thank you for taking the time to provide a very detailed and constructive review of our submission! We sincerely appreciate your thoughtful feedback and valuable suggestions, which have helped us identify areas for improvement and refinement in our work.
>
> Below, we address your comments and questions individually, incorporating clarifications and additional insights where relevant.
>
> ---
>
> > Q1: The excessive use of bold text and the absence of a clear outline make the paper’s contributions difficult to discern. Could the authors consider restructuring the introduction for better clarity?
>
> **Response:**
>
> Thank you for your valuable feedback! In the revised version, we have reduced the excessive use of bold text to enhance readability. We are committed to further restructure the introduction to provide a clearer outline of the paper’s contributions in the furture.
>
> > Q2: I find that the statements in the section **Theoretical Characterization vs Practical Use** rely on the example C.2. How increasing the sparsity beyond this simple example is understudied despite the authors strong claims that relaxing the fully-connected condition leads to better expressiveness of GeoNGNN compared to DisGNN.
>
> **Response:**
>
> Thank you for pointing out the reliance on Example C.2 in the discussion of sparsity and expressiveness! Here are further clarifications:
>
> 1. **Broader Examples Supporting Sparse GeoNGNN’s Expressiveness**
>
> Beyond Example C.2, one can easily check that sparse GeoNGNN can distinguish all symmetric point clouds in [1]’s Appendix A that DisGNN cannot. We give some more illustrations:
>
> - **6-node pair in Appendix A.1 of [1]:** Sparse GeoNGNN embeds isosceles triangles with different base lengths into node features, distinguishing the two graphs, even when subgraphs cover only the two nearest neighbors.
> - **The second 10-node pair in Appendix A.2 of [1]:** Sparse GeoNGNN captures regular pentagons in one graph and partial rings in another, again with a two-nearest-neighbor subgraph radius.
>
> 2. **General Rule of Sparse GeoNGNN’s Advancements**
>
> These examples illustrates a key property of GeoNGNN in sparse settings: it can capture local *subgraph* patterns (e.g., triangle structures in Example C.2) in its inner GNN and embed these into node features for the outer GNN. In contrast, DisGNN only captures *subtree* structures, which leads to confusion in symmetric cases (e.g., distinguishing triangles from rings in Example C.2). This aligns with broader findings in graph learning, showing the superiority of *subgraph* patterns over *tree* patterns.
>
> > Q3: There is no supporting evidence for GeoNGNN over existing architectures in the primary paper. Additionally, there is no comparative analysis involving node feature information generated by a complete invariant function. Could the authors address this gap?
>
> **Response:**
>
> We greatly appreciate your observation about the need for comparative evidence. We provide further clarifications:
>
> 1. **Extensive Experimental Evidence for GeoNGNN in Appendix D**
>
> The experiments in the main paper are primarily designed to support our theoretical conclusions. However, to provide a broader perspective and deepen the experimental analysis, we have included real-world evaluations in Appendix D. In these evaluations, GeoNGNN demonstrates competitive performance, surpassing models such as DimeNet, GemNet, and PaiNN (on MD17), as well as MACE, Equiformer (on MD22), and ComENet and SphereNet (on QM9) across multiple targets and on average.
>
> 2. **GeoNGNN as a Promising Model**
>
> Though not achieving SOTA universally, GeoNGNN demonstrates potential as a simple nested extension of DisGNN. We *deliberately keep its design minimal* to focus on theoretical insights, but it can be further tuned for enhanced performance (see lines 1017–1055).
>
> 3. **Node Features via Complete Invariant Functions**
>
> The paper emphasizes global expressiveness. However, as shown in our proof (lines 1816–1817), complete methods generate *powerful node features* capable of solely reconstructing entire geometries, thus ensuring global-level completeness. Node-level *geometric* expressiveness is an intricate topic, which, however, now lacks a commonly adopted formalization like E(3)-completeness in relevant field, thus we defer to future work.

---

> > ### Author Response · Authors · 2024-11-21
> > **Author Response [2/3]**
> >
> > > Q4: A significant portion of the QM9 dataset consists of non-symmetric structures. What are the proportions of indistinguishable data restricted to the subset of QM9 that includes only symmetric structures?
> >
> > **Response:**
> >
> > We appreciate this question, yet we respectfully believe there may be some misunderstandings. We offer the following clarification:
> >
> > **1. Focus on symmetric structures, NOT indistinguishable cases**
> >
> > Theorem 4.2 provides a necessary (but not sufficient) condition for unidentifiable cases, establishing that all unidentifiable cases must be *symmetric*. Consequently, by assessing the proportion of symmetric structures in QM9, we effectively evaluate a superset of the unidentifiable cases. This approach supports our claim that DisGNN is near-complete.
> >
> > Evaluating “the proportion of indistinguishable data within the symmetric subset of QM9” is impossible due to the lack of sufficient and necessary condition of indistinguishable cases, and also not necessary for the claimed theoretical conclusion.
> >
> > > Q5: In the QM9 noise study, the significant reduction in non-distinguishable point clouds occurs near what appears to be the level of reported error in the QM9 dataset. Given the reported error of 0.1Å, how is this error rescaled based on the applied scaling coefficient?
> >
> > We appreciate the reviewer’s insightful observation. However, we clarify that the “Deviation Error ($\epsilon$)” reported in Figure 2 is **dimensionless** --- As described in lines 927–928, we first rescale all point clouds to fit within a unit sphere. This ensures that the Deviation Error is standardized across all point clouds, independent of their original scale. Consequently, the result is not directly tied to the reported error $0.1 \, \text{Å}$ of the QM9 dataset itself.
> >
> > > Q6: Distinguishing structures on QM9, which lacks conformers, does not seem to be as important as datasets which contain conformers or very nearly isomorphic point clouds.The most compelling analysis appears to come from the study of MD17 but with mixed results. GeoNGNN appears to do particularly well on Benzene which is highly symmetric. How does Benzene behave under the noise tolerance study?
> >
> > **Response:**
> >
> > Thank you for this insightful observation! We provide the following clarifications:
> >
> > 1. **Why we evaluate DisGNN’s distinguishing ability on QM9?**
> >
> > The experiments on QM9 are designed to support our conclusion that “DisGNN is near-complete on real-world datasets.” QM9, being an almost exhaustive enumeration of small molecules, serves as a representative dataset in chemical field reflecting natural small molecules' distributions.
> >
> > We also respectfully believe there may be misunderstandings for our experiment purpose regarding “nearly isomorphic point clouds” (as demonstrated by the reviewer in the weakness part), and further response in our response to your Question 8.
> >
> > 2. **DisGNN’s Behavior on Benzene Conformations**
> >
> > It is interesting to find that DisGNN struggles with structures like Benzene, likely due to its high symmetry, affecting DisGNN’s learning behavior in this local manifold area. To investigate further, we evaluated the symmetry proportions of Benzene conformations in MD17, finding approximately 0.043% ($\epsilon =0.1$, $r=2$, 267 symmetric structures out of 627983 conformations) with C-symmetry. On contrast, molecules like aspirin contain 0% C-symmetric structures. These results may indicate that complete models like GeoNGNN can perform particularly well on such symmetric structures where DisGNN may falter.
> >
> > > Q7: Typically, ModelNet40 is sampled to avoid handling large point clouds. It is unclear from the text whether the entire mesh or a sampled version is used. If sampled uniformly, there is no guarantee that the symmetries are preserved. Could the authors clarify this in the text?
> >
> > **Response:**
> >
> > We thank the reviewer for this critical point, which we miss in the paper and will demonstrate in detail in the revised versions.
> >
> > To clarify, **we use farthest point sampling (FPS) to downsample ModelNet40 from 1024 points to 256 points to mitigate noise.** Farthest point sampling is chosen as it effectively preserves the structural integrity of the objects while reducing noise that could be introduced by uniform sampling. This ensures that the symmetries of the original shapes are maintained.

---

> ### Author Response · Authors · 2024-11-21
> **Author response [3/3]**
>
> > Q8: The selection of ModelNet40 does not seem to rigorously test the theoretical claims of the paper, which focus on nearly isomorphic point clouds. Could the authors provide more rigorous testing on datasets that better align with their theoretical focus?
>
> **Response:**
>
> We thank the reviewer for their insightful observation and for raising this important point. Below, we clarify how our evaluation aligns with the theoretical focus of the paper:
>
> 1. **Theoretical claims of DisGNN focus on “Near E(3)-completeness.”**
>
>    Datasets like QM9 and ModelNet40 are selected as real-world examples sampled from natural distributions, providing practical demonstrations of our theoretical claims. The relevance of these datasets has been clarified in the discussion, particularly in **“Why we evaluate DisGNN…on QM9.”**
>
> 2. **Corner cases theoretically unidentifiable by DisGNN are not “Nearly isomorphic point clouds.”**
>
>    While “Nearly isomorphic point clouds” can result in similar graph embeddings (e.g., two molecular conformations differing by a small disturbance in one atom’s position), in theoretical view, they are still distinguishable by DisGNN since precision is assumed large enough. Theoretical works such as [1-3] (including ours), as well as a broader range of studies in traditional graph learning, focus on inherently indistinguishable point cloud (or graph) pairs—those that result in *identical* embeddings even under models with *infinite* precision. These cases represent the intrinsic limitations of the model, rather than practical challenges related to noise or perturbations.
>
>    Consequently, **experiments that align with our theoretical focus should evaluate the distinguishability of models on such challenging pairs**. We address this explicitly in Section 6.2, where our experiments demonstrate the model’s limitations in such cases while showcasing the capabilities of complete models in handling them.
>
> > Q9: It is unclear from the text and appendix what each structure in the synthetic dataset represents, how these structures were constructed, and why they are significant. Could the authors provide more detailed explanations on the construction and relevance of these synthetic structures?
>
> **Response:**
>
> We thank the reviewer for their valuable suggestion. We have now included detailed explanations about these synthetic structures, including their construction and significance, in the appendix, highlighted with **green** text.
>
> ---
>
> Thank you once again for your detailed and thoughtful feedback. We recognize that some of the questions raised may have arisen from misunderstandings due to shortcomings in our initial presentation. We have clarified them in our responses and are continuing to improve them in the next revision.
>
> We sincerely hope that our responses resolve your concerns and offer deeper insights into our work. We are looking forward to further discussion with you. If there are no further technical concerns, we would be delighted if you could consider **raising your score** accordingly.
>
> [1] Is Distance Matrix Enough for Geometric Deep Learning?
>
> [2] Complete Neural Networks for Complete Euclidean Graphs.
>
> [3] Incompleteness of graph neural networks for points clouds in three dimensions

---

> > ### Comment · Reviewer_JK57 · 2024-11-24
> > **Response to Authors**
> >
> > I thank the authors for their detailed response and clarifications. I have raised my score accordingly.

---

> > > ### Author Response · Authors · 2024-11-24
> > > **Author Response**
> > >
> > > We are deeply grateful for your thoughtful engagement and are delighted that our responses addressed your concerns. Your updated score and support mean a great deal to us.
> > >
> > > Please don’t hesitate to reach out if you have any further questions or if there are any remaining concerns—we would be glad to continue the discussion.

---

### Official Review · Reviewer_jQxs · 2024-11-04

**Soundness:** 2
**Presentation:** 2
**Contribution:** 4
**Rating:** 6
**Confidence:** 3

**Summary:**

This paper explores the geometric completeness of a significant class of geometric deep learning models: invariant neural networks. These networks leverage invariant features to impose strong inductive biases on spatial information, yet their theoretical expressive power remains somewhat unclear. This study aims to bridge that gap, enhancing both our theoretical understanding and practical application of these models.

The authors first demonstrate that incorporating distance into message-passing neural networks (like DisGNN) allows for the identification of asymmetric point clouds but struggles with highly symmetric ones. They then investigate geometric extensions of subgraph-based GNNs and prove that these models, specifically GeoNGNN, can successfully distinguish symmetric point clouds, achieving E(3)-completeness.

**Strengths:**

- The paper attempts to address a crucial problem that enhances our understanding of the potential of invariant neural networks and can guide future model design.
- Investigating the geometric counterparts of subgraph GNNs is a novel contribution.
- The results extend beyond specific cases, such as asymmetric point clouds, broadening our understanding of how these models perform on symmetric point clouds as well.

**Weaknesses:**

- The paper lacks clarity and structure in some areas. The detailed explanation of NGNN, which serves as the backbone of their main contribution, the GeoNGNN framework, is left in the appendix. I recommend the authors integrate key aspects of NGNN, such as its core equations or an architectural diagram, into the main text. Additionally, including a comparison with the original DisGNN would be helpful—highlighting the differences and explaining what enables NGNN (intuitively) to overcome the limitations of DisGNN. This would give readers a clearer understanding of how the proposed approach builds on previous work and addresses specific challenges without disrupting the flow. Instead, a brief overview of NGNN, along with key formulas and a comparison with GNN, could benefit the flow of the whole paper.

- The results are primarily constrained to cases with global connectivity, which is often impractical in real-world applications due to the significant computational costs. Several studies [1], [2], [3], [4] have explored scenarios where the graph is not fully connected, underscoring the need to evaluate the performance of invariant neural networks in sparse graph settings. In practice, invariant neural networks tend to perform worse than equivariant ones in these cases. While the authors have left these cases in the future direction, it would greatly strengthen the paper if they could extend their analysis to sparse graphs or at least discuss how their completeness results may vary with different levels of graph sparsity. Providing theoretical bounds on performance degradation as connectivity decreases would also be valuable.

- The experimental results, while showing some improvement, are relatively marginal, which limits the empirical impact of the work. I suspect this might be due to the sparsity of the graphs used in practical applications. The authors should aim to demonstrate the significance of their approach by clarifying in which specific cases their method outperforms existing methods. Providing examples or scenarios where GeoNGNN has a clear advantage would strengthen the empirical contributions.


[1] Wang, L., Liu, Y., Lin, Y., Liu, H., & Ji, S. (2022). ComENet: Towards complete and efficient message passing for 3D molecular graphs. Advances in Neural Information Processing Systems, 35, 650-664.
[2] Joshi, C. K., Bodnar, C., Mathis, S. V., Cohen, T., & Lio, P. (2023, July). On the expressive power of geometric graph neural networks. In International Conference on Machine Learning (pp. 15330-15355). PMLR.
[3] Wang, S. H., Hsu, Y. C., Baker, J., Bertozzi, A. L., Xin, J., & Wang, B. (2024). Rethinking the benefits of steerable features in 3D equivariant graph neural networks. In The Twelfth International Conference on Learning Representations, ICLR
[4] Sverdlov, Y., & Dym, N. (2024). On the Expressive Power of Sparse Geometric MPNNs.

**Questions:**

Please address the concerns I raised in the weaknesses section. Additionally, I recommend revising the introduction to better reflect the paper's contributions. In my view, the primary contribution is the proposal of the geometric counterpart of NGNN and the proof that this approach effectively resolves the limitation of DisGNN in identifying **symmetric point clouds when the graphs are even fully connected**.
The authors should also consider relevant experiments in this direction to emphasize the novelty and significance of this work.

---

> ### Author Response · Authors · 2024-11-21
> **Author Response [1/2]**
>
> We sincerely thank you for taking the time to provide detailed and constructive feedback on our submission. Your comments are highly valuable and have significantly helped us identify areas for improvement in our work. Below, we address your suggestions and concerns.
>
> > I recommend the authors integrate key aspects of NGNN ... into the main text. Additionally, including a comparison with the original DisGNN would be helpful—highlighting the differences and explaining what enables NGNN (intuitively) to overcome the limitations of DisGNN.
> >
>
> **Response:**
>
> Thank you for this insightful suggestion! We appreciate your recommendation to include key aspects of NGNN in the main text for enhanced clarity.
>
> 1. **Further elaboration of NGNN**
>
> While we recognize the value of integrating NGNN’s core equations or an architectural diagram directly into the main text, the constraints imposed by the extensive technical results and page limits make this challenging.
>
> To address this, we have expanded the relevant discussion in the Appendix, highlighting these additions in **green** in the revised version. We hope this approach strikes a balance between clarity and adherence to formatting constraints.
>
> 1. **Intuitive explanation of GeoNGNN’s advancements over DisGNN**
>
> In the main text, we have provided an intuitive explanation for how GeoNGNN resolves previously unidentifiable cases in DisGNN. Specifically, GeoNGNN leverages symmetry-breaking through node marking (see lines 272–278 and 302–308). Since all unidentifiable cases are symmetric (Theorem 4.2), this symmetry-breaking approach suffices to address all these corner cases effectively.
>
> That said, we understand that the dense technical details may still pose challenges to accessibility. We sincerely value your feedback and are committed to further refining our explanations to enhance the paper’s clarity and make it more approachable for a broader audience
>
> > Several studies [1], [2], [3], [4] have explored scenarios where the graph is not fully connected, underscoring the need to evaluate the performance of invariant neural networks in sparse graph settings.
> >
>
> **Response:**
>
> Thank you for raising this important point! We acknowledge that our work primarily focuses on fully connected graphs and that this limitation is discussed in Sections 5.4 and 7 of the paper. Regarding the references you provided, we discuss them as follows:
>
> 1. **ComENet [1] also establish theoretical conclusions assuming strongly connected 3D graphs.**
>
>     We respectfully note that while ComENet [1] establishes important theoretical results, these conclusions are still derived under the assumption of strongly connected graphs, as outlined in Section 3.1 of [1]. Additionally, it does not address cases involving multiple nearest neighbors.
>
> 2. ***Invariant* methods we study cannot trivially extend to sparse settings compared to previous *equivariant* methods [2, 3, 4].**
>
>     We would illustrate this through an example: Consider a point cloud with two clusters connected by a single edge. Such sparsity is common, where the graph has only local connectivity but remains overall connected. In these cases, invariant methods struggle to detect the relevant orientations of the two clusters (e.g., by swinging the edge connecting the two clusters), highlighting a fundamental limitation of invariant models in sparse graphs. We extensively discuss this challenge in lines 406–408, alongside relevant works.
>
>     We appreciate your suggestion to consider sparse graph settings in future work and will strive to address this in subsequent research endeavors.
>
>
> > The authors should aim to demonstrate the significance of their approach by clarifying in which specific cases their method outperforms existing methods. Providing examples or scenarios where GeoNGNN has a clear advantage would strengthen the empirical contributions.
> >
>
> **Response:**
>
> Thank you for this excellent suggestion! We appreciate the opportunity to clarify the empirical contributions of GeoNGNN and would like to direct your attention to Appendix D, where we provide extensive experimental evidence. Below, we highlight some key points:
>
> 1. **GeoNGNN demonstrates strong performance, surpassing advanced geometric models:**
>
> Despite its straightforward design, GeoNGNN achieves very promising results. For instance: It surpasses DimeNet, GemNet, and PaiNN on MD17; It outperforms MACE and Equiformer on MD22; It exceeds ComENet and SphereNet on QM9. These results underscore GeoNGNN’s capability to deliver competitive performance across diverse benchmarks.

---

> ### Author Response · Authors · 2024-11-21
> **Author Response [2/2]**
>
> 2. **GeoNGNN is a promising foundation despite not being fully optimized for SOTA:**
>
> While GeoNGNN does not consistently achieve SOTA results, we believe it represents a highly promising approach. Its current design is *deliberately simple*, serving as a nested version of the basic invariant model, DisGNN, for our theoretical purpose. As discussed in lines 1017–1055, its architecture has significant potential for refinement and extension to enhance performance further. However, given the theoretical focus of this work, we prioritized simplicity to underscore the model’s completeness rather than optimizing it for empirical SOTA performance.
>
> > In my view, the primary contribution is the proposal of the geometric counterpart of NGNN and the proof that this approach effectively resolves the limitation of DisGNN in identifying **symmetric point clouds when the graphs are even fully connected**. The authors should also consider relevant experiments in this direction to emphasize the novelty and significance of this work.
> >
>
> **Response:**
>
> We are grateful for your thoughtful observation regarding the primary contribution of our work.
>
> 1. **Main focus of our work**
>
> We would like to clarify that the central focus of our study is to rigorously **establish the expressiveness of a broad class of models**, including DisGNN and complete models, thereby advancing the theoretical understanding of their capabilities.
>
> While GeoNGNN is introduced as a pivotal proof-of-concept, it is not the exclusive focus of our work. Instead, as the simplest complete model, GeoNGNN provides a foundational framework for demonstrating the completeness of other models. Our experiments are designed to align with this theoretical emphasis rather than highlighting the novelty of GeoNGNN alone.
>
> 2. **Existing Evidence Supporting the Effectiveness of Complete Models**
>
> In Section 6.2, we assess all established complete models on tasks specifically designed to distinguish pairs of point clouds that DisGNN cannot differentiate. The results consistently demonstrate that complete models effectively address these limitations, providing strong evidence of their ability to resolve symmetric structures.
>
> ---
>
> In conclusion, we are deeply thankful for your thoughtful and constructive feedback. Your insights have greatly helped us improve the presentation and focus of our work. We hope our clarification address your concerns and we would greatly appreciate it if you could **raise your score** accordingly.

---

> > ### Comment · Reviewer_jQxs · 2024-11-24
> > **Further Questions**
> >
> > I appreciate the authors' effort in addressing some of my concerns. However, there are still key areas where further clarification is necessary to ensure a clear and coherent understanding of the paper’s contributions and framework.
> >
> > **Primary Contribution and Scope:**
> > My main concern relates to reframing the primary contribution as “the proposal of GeoNGNN and the proof that this approach effectively resolves the limitation of DisGNN in identifying symmetric point clouds when the graphs are fully connected.”
> > Notice that while the results in this work successfully demonstrate the completeness of SphereNet in fully connected graph settings, the paper also implicitly acknowledges that this does not extend to scenarios involving non-fully connected graphs. It is well-known that distinguishing cis- and trans-isomers in sparse graphs often requires additional information, such as torsion angles, to glue local features into a global understanding—something not addressed under the current setup.
> >
> > In my opinion, the claim in the introduction is overly broad. The assertion that “DimeNet, SphereNet, and GemNet are Complete” is valid only under the assumption of fully connected graphs. While it is acceptable for the theoretical results to be restricted to fully connected graphs, the introduction and stated contributions should be revised to explicitly reflect this limitation. As it stands, the way the theoretical results are framed in the introduction risks being misleading or overly generalized, which could detract from the paper’s otherwise significant contributions.  I recommend that the authors rephrase their claims and clarify the scope of their contributions to prevent potential misunderstandings.
> >
> > **Architecture of GeoNGNN:**
> > My concern regarding including GeoNGNN's architecture in the main text stems from its central role as the backbone framework that addresses DisGNN's limitations. Without a clear presentation of this architecture in the main text, I worry that the paper’s flow and ability to convey its key contributions effectively may suffer, especially for the general audience. Providing an accessible and concise explanation of GeoNGNN in the main text would enhance the paper’s clarity and impact.
> >
> > **Empirical Evidence:**
> > While the empirical evidence provided is not sufficiently robust, I acknowledge the novelty of this paper and its proposed frameworks. I understand that the paper's focus is on the ability to distinguish non-isomorphic point clouds rather than on developing architectures that achieve state-of-the-art results. I would like to reconsider the marginal improvement on several benchmarks as not necessarily a weakness but focus on rephrasing the conveying of theoretical contributions.
> >
> > Overall, I believe this is a solid paper with valuable contributions. However, I strongly recommend that the authors address my concerns, especially the one regarding Primary Contribution and Scope. The introduction should explicitly state the assumption of fully connected graphs and emphasize that even under this condition, the question being studied remains unresolved and presents significant challenges for DisGNN. This would underscore the importance of the authors’ work and align the framing with its true contributions. If these concerns are adequately addressed, I would likely raise my score accordingly.

---

> > > ### Author Response · Authors · 2024-11-25
> > > **Author Response**
> > >
> > > Thank you so much for your detailed and constructive feedback! We sincerely appreciate the time and effort you have taken to thoroughly engage with our work and provide us with such thoughtful suggestions for improvement.
> > >
> > > We have carefully considered your concerns and have made preliminary revisions to the paper accordingly. These revisions are marked in $\textcolor{red}{\textbf{red}}$ for clarity. Specifically, we have:
> > >
> > > + **Clarified Theoretical Claims**: Addressed the potentially misleading theoretical claims in the abstract, introduction, and conclusion by explicitly stating the fully connected graph condition. We also emphasized the significance of the problem in the introduction, highlighted the differences from prior work, and directed readers to the relevant sections for further details.
> > >
> > > + **Enhanced GeoNGNN Explanation**: Added a brief intuitive description of GeoNGNN and its relationship with DisGNN in the main text. Additionally, we included a much more detailed formalization of the original NGNN in Appendix G.2 for completeness and self-containment.
> > >
> > > Due to page limitations, some of these revisions are located in the appendix, and we acknowledge that further refinements may still be needed. We remain committed to improving both the **rigor** and **readability** of the paper, particularly for a general audience, by revisiting and rephrasing the statements more broadly where possible.
> > >
> > > Please do not hesitate to reach out if the revisions do not fully address your concerns. We are always eager to refine and enhance the paper further based on your valuable input. Thank you again for your guidance and support!

---

> > > > ### Comment · Reviewer_jQxs · 2024-11-26
> > > > **Response to Authors**
> > > >
> > > > The revisions reformulating the theoretical claims have significantly clarified the contributions of this paper, resolving my initial concerns. As a result, I have raised my rating to Borderline Accept.
> > > >
> > > > However, I believe there is considerable room for improvement in the paper’s flow, particularly regarding the presentation of GeoNGNN. While the authors direct readers to the appendix for details about the architecture, it is unconventional to relegate the explicit description of a new proposed theoretical framework to the appendix. This approach disrupts the flow of the main text and makes it harder for readers to fully grasp the core ideas without constant back-and-forth references. I strongly recommend integrating a concise yet complete explanation of GeoNGNN within the main body of the paper to enhance its coherence and accessibility.

---

> ### Author Response · Authors · 2024-11-27
> **Author Response**
>
> Again, we deeply appreciate your continued engagement and valuable feedback on improving our paper! We deeply acknowledge your efforts, and we remain committed to refining and updating the manuscript.
>
> We recognize that the presentation of GeoNGNN may have caused some confusion, particularly for broader audiences who are less familiar with subgraph GNNs. To address this, we have prepared a preliminary revision that **provides both an intuitive and formal explanation of GeoNGNN in the main text**. Additionally, we have **highlighted its key advantages over DisGNN**, as discussed with you (e.g., subgraph versus subtree). These modifications are highlighted with **$\textcolor{orange}{yellow}$ in Section 5.1**. We believe these updates will help align readers’ understanding with our intended message.
>
> We hope this revision addresses your concerns to some extent. We are eager to engage in further discussions with you to continue improving the paper. Once again, thank you so much for your valuable guidance and support!

---

### Meta-Review · Area_Chair_22pe · 2024-12-20

**Metareview:**

This paper analyzes the theoretical expressive power of invariant geometric deep learning models, focusing on their ability to represent point cloud geometry under fully-connected conditions. It establishes that DisGNN (message-passing neural networks incorporating distance) is limited only by highly symmetric point clouds and demonstrates that GeoNGNN ( the geometric counterpart of one of the simplest subgraph graph neural networks) achieves E(3)-completeness, resolving these cases. The work further shows that other models like DimeNet, GemNet, and SphereNet can also achieve E(3)-completeness. These results enhance the understanding of invariant models' capabilities, bridging the gap between theoretical expressiveness and practical applications.

One of the main criticism, shared by reviewers, was targeted toward the clarity of exposition of this work. Authors provided revisions of their text, that was in general approved and acknowledged by reviewers. The scores are consistently given as borderline accept. After reading the paper, I believe that the contributions are above the threshold for a publication at ICLR, and I recommend an accept. Nevertheless, I strongly encourage authors to thoroughly take into account the reviewers remarks regarding the final version of their paper.

**Additional Comments On Reviewer Discussion:**

Two reviewers raised their scores to 6 after the revision provided by authors.

---

### Decision · Program_Chairs · 2025-01-22

Accept (Poster)